# LEARNING FOR HIGHLY FAITHFUL EXPLAINABILITY

**Yuhan Guo, Lizhong Ding**[*]**, Shihao Jia, Yanyu Ren, Pengqi Li, Jiarun Fu, Changsheng Li, Ye Yuan, Guoren Wang**
School of Computer Science and Technology
Beijing Institute of Technology
No. 5 Zhongguancun South Street, Haidian District, Beijing, China
`3220231188@bit.edu.cn, lizhong.ding@outlook.com,`
`{1120221561, renyanyu, 3220242275, 3120235197, lcs}@bit.edu.cn,`
`yuan-ye@bit.edu.cn, wanggrbit@126.com`

## ABSTRACT

*Learning to Explain* is a forward-looking paradigm recently proposed in the field of explainable AI, which envisions training explainers capable of producing high-quality explanations for target models efficiently. Although existing studies have made attempts through self-supervised optimization or learning from prior explanation methods, the *Learning to Explain* paradigm still faces three critical challenges: 1) self-supervised objectives rely on assumptions about the target model or task, restricting their generalizability; 2) methods driven by prior explanations struggle to guarantee the quality of the supervisory signals; and 3) depending exclusively on either approach leads to poor convergence or limited explanation quality. To address these challenges, we propose a *faithfulness*-guided amortized explainer that 1) theoretically derives a self-supervised objective free from assumptions about the target model or task, 2) practically generates high-quality supervisory signals by deduplicating and filtering prior explanations, and 3) jointly optimizes both objectives via a dynamic weighting strategy, enabling the amortized explainer to produce more faithful explanations for complex, high-dimensional models. We re-formalize multiple well-validated faithfulness evaluation metrics within a unified notation system and theoretically prove that an explanation mapping can simultaneously achieve optimality across all these metrics. We aggregate prior explanation methods to generate high-quality supervised signals through deduplicating and faithfulness-based filtering. Our amortized explainer leverages dynamic weighting to guide optimization, initially emphasizing pattern consistency with the supervised signals for rapid convergence, and subsequently refining explanation quality by approximating the most faithful explanation mapping. Extensive experiments across various target models and image, text, and tabular tasks demonstrate that the proposed explainer consistently outperforms all prior explanation methods across all faithfulness metrics, highlighting its effectiveness and its potential to offer a systematic solution to the fundamental challenges of the *Learning to Explain* paradigm.

## 1 INTRODUCTION

The widespread deployment of deep models has raised substantial concerns regarding their explainability (Li et al., 2025; Fu et al., 2026; Zhang et al., 2025), motivating the community to propose a variety of explanation methods across different eXplainable AI (XAI) paradigms (Jethani et al., 2021). Among these, the *Learning to Explain* paradigm (Madsen et al., 2024) puts forward an appealing future: training a neural network as an explainer for the target model by optimizing a learning objective, thereby generating high-quality explanations in inference through a single forward pass (Yoon et al., 2019; Bhalla et al., 2023). This paradigm is often referred to as *amortized explanation methods*, as it shifts the computational burden of interacting with the target model to the training of the explainer, thereby reducing the cost of generating explanations at inference time.

---

[*]Corresponding author.

Explanations cannot be learned from scratch without guidance; they necessarily rely on prior theoretical assumptions or supervisory signals. *Self-supervised optimization-based methods* (Bhalla et al., 2023; Schwab & Karlen, 2019) allow the explainer to interact with the target model during training, optimizing a self-supervised loss function designed to measure explanation quality. *Prior-explanation-driven methods* (Jethani et al., 2022; Covert et al., 2023; Barkan et al., 2023) employ existing XAI algorithms to generate a set of attributions for the target model, and then train the explainer to model the mapping from inputs to explanations. However, the *Learning to Explain* paradigm has not been widely adopted, primarily because three major challenges in improving the performance of amortized explainers: 1) The objective functions of self-supervised optimization-based methods are limited by ideal assumptions about the target model or the task setting (Chen et al., 2018). 2) Prior-explanation-driven methods face difficulties to ensure the quality of the supervisory signals (Barkan et al., 2023). 3) Explainers trained solely on self-supervised objectives struggle with convergence and scale poorly to high-dimensional or complex models (Schwab & Karlen, 2019), whereas those relying only on prior explanations cannot surpass the quality of the labels they imitate (Situ et al., 2021).

To address these challenges, we introduce the critical concept of *faithfulness* (Bhatt et al., 2020; Dasgupta & Moshkovitz, 2022) from XAI into the *Learning to Explain* paradigm and propose Deep architecture-based Faithful amortized explainer (**DeepFaith**). Faithfulness serves as a quantitative metric that assesses whether an explanation accurately reflects the model's true decision-making process (Li et al., 2023), thereby directly indicating explanation quality. From the perspective of faithfulness, we theoretically construct a self-supervised optimization objective that requires no assumptions about the target model or task, practically design a methodology for generating high-quality supervisory signals by filtering prior explanations through faithfulness criteria, and develop a dynamic joint optimization strategy that integrates the self-supervised objective with faithfulness-filtered supervisory signals. Together, these novel designs enable our amortized explainers to generate explanations that are more faithful than prior methods, even for complex models and high-dimensional tasks.

Our technical approach and contributions to the XAI field can be summarized in three aspects:

(1) We are the first to introduce a self-supervised objective that directly optimizes explanation faithfulness within the *Learning to Explain* paradigm. By re-formalizing ten widely used and well-validated faithfulness metrics, we propose and theoretically prove the existence of an optimal faithful explanation mapping that simultaneously achieves optimality across all these metrics.

(2) We are the first to provide a systematic solution for enhancing the quality of supervisory signals in the *Learning to Explain* paradigm. We aggregate explanations generated by multiple methods for the target model, increase the diversity of supervisory signals through similarity-based deduplicating, and improve their faithfulness via $p$-quantile filtering across multiple metrics.

(3) We are the first to complementarily integrate the two main approaches within the *Learning to Explain* paradigm. We design a dynamic weighting-based joint optimization strategy: during the early stage of explainer training, fitting the supervisory labels enables rapid convergence to a basic explanatory capability, followed by approximating the theoretically optimal mapping to further enhance explanation quality.

To validate the effectiveness of these contributions, we evaluate **DeepFaith** on 12 explanation tasks spanning image, text, and tabular modalities, as well as across diverse models to be explained. Comparative experiments demonstrate that **DeepFaith** consistently outperforms baseline methods in terms of faithfulness. Through ablation studies, we further validate the effectiveness of each component in our proposed method. In addition, the visualizations provided in the appendix illustrate that **DeepFaith** produces clear and intuitive explanations, along with a comparison of runtime.

## 2 RELATED WORK

**Self-Supervised Optimization-based Methods** Methods that train explainers by optimizing self-supervised objectives rely on assumptions about the target model or task setting, constructing a self-supervised loss function to measure explanation quality. During training, the explainer interacts with the target model, improving explanations by minimizing this loss. For example, VerT (Bhalla et al., 2023) assumes features strictly divide into signal and noise, training the explainer to learn the spars-

est separating mask. However, since deep models encode higher-order interactions, this assumption rarely holds in practice. Similarly, L2X (Chen et al., 2018) is grounded in an information-theoretic assumption that tasks involve clearly separable features; it trains the explainer by maximizing the mutual information between feature subsets and predictions, facing the same limitation as VerT. CXPlain (Schwab & Karlen, 2019) assumes input features capture all potential factors influencing the prediction, treating feature removal effects as causality and training the explainer to learn the causal importance of features for the output. Yet, real-world confounders are highly unobservable, and removing correlated features can easily produce spurious causal effects. Moreover, because the self-supervised objective nests the target model, its highly non-convex nature makes explainers difficult to converge when dealing with complex models or high-dimensional tasks.

**Prior-Explanation-Driven Methods** These approaches leverage existing explanation methods as prior references by generating explanations of the target model under different inputs before training and using them as supervisory signals. The explainer is then trained by fitting the mapping from inputs to explanations. L2E (Situ et al., 2021) neuralizes multiple XAI algorithms via knowledge distillation, enabling explainers to produce fast and stable explanations. However, prior explanation methods cannot guarantee high-quality explanations for all samples, and low-quality supervisory signals negatively affect explainer training. FastSHAP (Jethani et al., 2022) and ViT Shapley (Covert et al., 2023) adopt the Shapley Value (Lundberg & Allen, 2017) as a high-quality prior and train explainers to approximate the true Shapley Values. Nevertheless, Shapley Values fail to attribute non-independent features and nonlinear interactions, making them unreliable as the sole prior source. Moreover, since fitting supervisory explanation signals essentially mimics existing XAI methods, the explainer's performance is inherently bounded by the quality of the explanations in the dataset, hindering further improvement.

## 3 THEORETICALLY OPTIMAL FAITHFULNESS OBJECTIVE

In this section, building on deep insights into the theory of *faithfulness*, we propose a model- and task-agnostic self-supervised objective within the *Learning to Explain* paradigm. Let $f : \mathcal{X} \to \mathcal{Y}$ denote the model to be explained, where the input space $\mathcal{X} \subseteq \mathbb{R}^{n \times d}$ consists of instance $x = (x_1, x_2, \ldots, x_n)$ with each element $x_i \in \mathbb{R}^d$. In our experiments: for vision, $x$ is an image of $n$ patches, each $x_i$ representing the $d$-dimensional pixels in a patch; for NLP, $x$ is a sequence of $n$ tokens with $x_i$ as the $d$-dimensional embedding of the $i$-th token; for tabular data, $x$ is a row with $n$ scalar features ($d = 1$). The model output $f(x)$ aims to approximate $y \in \mathcal{Y} \subseteq \mathbb{R}$, e.g., the predicted probability for the target class in classification. We use $[n]$ to denote the set $\{1, 2, \ldots, n\}$, and use $(i)_{i=1}^n$ to denote the vector $(1, 2, \ldots, n)$.

We begin with the observation that current faithfulness metrics follow two distinct views: one evaluates the accuracy of attribution values from a *saliency* perspective (Bhatt et al., 2020; Alvarez Melis & Jaakkola, 2018), while the other assesses the relative importance of input elements from a *permutation* perspective (Samek et al., 2015; Rieger & Hansen, 2020). Thus, it is essential to distinguish between explanations under these two perspectives before conducting the analysis.

**Definition 1** (Saliency Explanation). *A saliency explanation method is defined as a mapping $S_f : \mathcal{X} \to [0, 1]^n$ that, given an input $x$ and model $f$, outputs a saliency vector $s = (s_1, s_2, \ldots, s_n) \in [0, 1]^n$, where each $s_i$ quantifies the contribution of $x_i$ (e.g., a patch, token, or scalar feature) to the prediction $\hat{y} = f(x)$.*

**Definition 2** (Permutation Explanation). *A permutation explanation method is defined as a mapping $\Pi_f : \mathcal{X} \to \mathfrak{S}_n$, where $\mathfrak{S}_n = \{(\pi(i))_{i=1}^n | \{\pi(1), \pi(2), \cdots, \pi(n)\} = [n]\}$[1] denotes all permutations of $[n]$. Given $x$ and model $f$, $\Pi_f$ outputs $\pi \in \mathfrak{S}_n$, indicating that $x_{\pi(i)}$ contributes no less to the model's prediction than $x_{\pi(i+1)}$.*

*Remark.* Two types of explanations can be interconverted via simple functions: $\mathfrak{P}(s) = \text{argsort}_\downarrow \{s_1, s_2, \ldots, s_n\}$ represents the descending-order index of $s$, mapping a saliency explanation to a permutation explanation; while $\Sigma(\pi)$ converts a permutation explanation into a saliency explanation, where $\Sigma(\pi)_i = (n - \pi(i) + 1)/n$. Since a saliency explanation assigns a specific importance score to each $x_i$, while a permutation explanation does not, $\mathfrak{P}(s)$ cannot be recovered back to $s$ through $\Sigma$.

---

[1] For clarity, we use $\pi(i)$ to denote the $i$-th element in vector $\pi$.

Table 1: We re-formalize ten widely-used and well-validated faithfulness metrics under our notation system, including FC (Bhatt et al., 2020), FE (Alvarez Melis & Jaakkola, 2018), INF (Yeh et al., 2019), and MC for saliency explanations, as well as DEL and INS (Petsiuk & Saenko, 2018), NEG and POS (Barkan et al., 2023), RP, and IROF (Rieger & Hansen, 2020) for permutation explanations. ↑ indicates that higher values correspond to greater explanation faithfulness, while ↓ indicates that lower values correspond to greater faithfulness.

| Faithfulness Metric | Input | Formula | Output |
|---|---|---|---|
| Faithfulness Correlation (FC) ↑ | $s; x, f$ | $\tau\left[\left(\sum_{i \in \mathcal{I}} s_i\right)_{\mathcal{I} \subseteq [n]}, \left(\Delta\left[f(x), f(x \setminus \mathcal{I})\right]\right)_{\mathcal{I} \subseteq [n]}\right]$ | $[-1, 1]$ |
| Faithfulness Estimate (FE) ↑ | $S_f; \{x^{(i)}, \mathcal{I}_i\}_{i=1}^N, f$ | $\tau\left[\left(\sum_{j \in \mathcal{I}_i} S_f(x^{(i)})_j\right)_{i=1}^N, \left(\Delta\left[f(x^{(i)}), f(x^{(i)} \setminus \mathcal{I}_i)\right]\right)_{i=1}^N\right]$ | $[-1, 1]$ |
| Monotonicity Correlation (MC) ↑ | $s; x, \{\mathcal{I}_i\}_{i=1}^N, f$ | $\tau[(\sum_{j \in \mathcal{I}_i} s_j)_{i=1}^N, (\Delta\left[f(x), f(x \setminus \mathcal{I}_i)\right])_{i=1}^N]$ | $[-1, 1]$ |
| Infidelity (INF) ↑ | $s; x, \{\mathcal{I}_i \sim \mathcal{P}([n])\}_{i=1}^N, f$ | $\tau[(\sum_{j \in \mathcal{I}_i} s_j)_{i=1}^N, (\Delta\left[f(x), f(x \setminus \mathcal{I}_i)\right])_{i=1}^N]$ | $[-1, 1]$ |
| Deletion Score (DEL) ↓ | $\pi; x, f$ | $\frac{1}{n} \int_{i=0^+}^{n} \Delta^-\left[f(x), f(x \setminus \bigcup_{j=1}^{[i]} \pi(j))\right] \mathrm{d}i$ | $[0, 1]$ |
| Insertion Score (INS) ↑ | $\pi; x, f$ | $\frac{1}{n} \int_{i=0^+}^{n} \Delta^-\left[f(x), f(x^\circ \cup \bigcup_{j=1}^{[i]} \pi(j))\right] \mathrm{d}i$ | $[0, 1]$ |
| Negative Perturbation (NEG) ↑ | $\pi; x, f$ | $\frac{1}{t} \int_{i=0^+}^{t} \Delta^-\left[f(x), f(x \setminus \bigcup_{j=1}^{[i]} \overleftarrow{\pi}(j))\right] \mathrm{d}i$ | $[0, 1]$ |
| Positive Perturbation (POS) ↓ | $\pi; x, f$ | $\frac{1}{t} \int_{i=0^+}^{t} \Delta^-\left[f(x), f(x \setminus \bigcup_{j=1}^{[i]} \pi(j))\right] \mathrm{d}i$ | $[0, 1]$ |
| Region Perturbation (RP) ↑ | $\Pi_f; \{x^{(i)}\}_{i=1}^N, f$ | $\frac{1}{N} \sum_{i=1}^N \left(\frac{1}{n+1} \sum_{j=0}^n \Delta\left[f(x^{(i)}), f(x^{(i)} \setminus \bigcup_{k=1}^j \Pi_f(x^{(i)})(k))\right]\right)$ | $[0, 1]$ |
| Iterative Removal of Features (IROF) ↑ | $\Pi_f; \{x^{(i)}\}_{i=1}^N, f$ | $\frac{1}{Nn} \sum_{i=1}^N \int_{j=0^+}^n 1 - \Delta^-\left[f(x^{(i)}), f(x^{(i)} \setminus \bigcup_{k=1}^{[j]} \Pi_f(x^{(i)})(k))\right] \mathrm{d}j$ | $[0, 1]$ |

As shown in Table 1, we select ten widely used and well-validated faithfulness evaluation metrics, uncover and extract their shared functional components, and abstract them to enable a unified theoretical analysis. The functional components we define are as follows:

**Input Perturbation** $x \setminus \mathcal{I}$ ($\mathcal{I} \subseteq [n]$) denotes input $x$ with sub-elements $\{x_i | i \in \mathcal{I}\}$ removed (via noise substitution (Rong et al., 2022), baseline replacement (Bhatt et al., 2020; Bach et al., 2015), or linear interpolation (Rieger & Hansen, 2020)). And $x^\circ \cup \mathcal{I}$ denotes inserting $x_i \mid i \in \mathcal{I}$ into the baseline input $x^\circ$ (e.g., blurred input, noise, or zero vector).

**Perturbation Effect** $\Delta : \mathcal{Y} \times \mathcal{Y} \to [0, 1]$ (e.g., $|y^{(1)} - y^{(2)}|$ or $\frac{1}{2}(y^{(1)} - y^{(2)})^2$ (Yeh et al., 2019)) measures the discrepancy between model predictions before and after perturbation.

**Preservation Effect** $\Delta^- : \mathcal{Y} \times \mathcal{Y} \to [0, 1]$ (e.g., $\left|y^{(1)}/y^{(2)}\right|$ (Rieger & Hansen, 2020) or target class confidence) measures the degree to which the original prediction is preserved, which is strictly decreasing with respect to $\Delta$.

**Correlation** $\tau : \mathbb{R}^m \times \mathbb{R}^m \to [-1, 1]$ measures correlations between $m$-dimensional vectors, such as Pearson or Spearman coefficients (Alvarez Melis & Jaakkola, 2018; Nguyen & Martínez, 2020).

Please refer to Appendix D for more specific examples. We re-formalize four *saliency* perspective faithfulness metrics under our notation system in Table 1. Specifically, FC enumerates all subsets of $[n]$ as perturbation index sets $\mathcal{I}$ (computed using Monte Carlo sampling to ensure feasibility); FE evaluates $N$ samples, each with a specific $\mathcal{I}$; MC defines a fixed perturbation sequence $\{\mathcal{I}_i\}_{i=1}^N$ on one sample; and INF samples $N$ index sets from a distribution $\mathcal{P}$, which we instantiate as $\mathcal{P}([n]) = \mathrm{Uniform}(2^{[n]})$, a discretized version of the original INF.

By uncovering that the core idea behind FC, FE, MC, and INF is to evaluate the correlation between the local sum of saliency explanations over perturbed indices and the corresponding perturbation effects, we propose a saliency explanation mapping with optimal faithfulness across these four.

**Proposition 1.** *Given a model $f$ being explained and its input space $\mathcal{X}$, for a fixed correlation measure $\tau$ and perturbation effect $\Delta$, suppose there exists a saliency explanation mapping $S_f^*$ such that $\forall x \in \mathcal{X}$ and $\forall \{\mathcal{I}_i \subseteq [n]\}_{i=1}^N$,*

$$S_f^* = \underset{S_f}{\mathrm{argmax}}\, \tau\left[\left(\sum_{j \in \mathcal{I}_i} S_f(x)_j\right)_{i=1}^N, \left(\Delta[f(x), f(x \setminus \mathcal{I}_i)]\right)_{i=1}^N\right], \tag{1}$$

*then the saliency explanations generated by $S_f^*$ always achieve optimal faithfulness under the* FC*,* FE*,* INF*, and* MC *evaluation metrics.*

*Proof.* Please refer to Appendix A for detailed proof. □

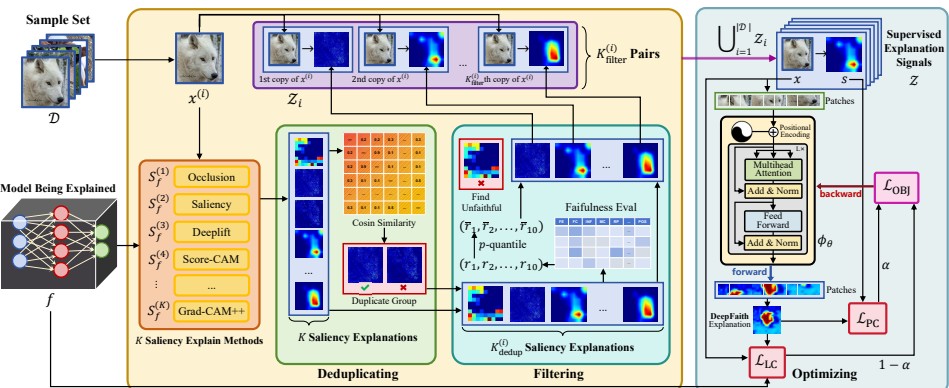

Figure 1: **DeepFaith** learning framework. We meticulously design a high-quality supervised explanation signal generation workflow that leverages $K$ existing explanation methods with deduplicating and filtering. We further introduce a training pipeline for a deep neural explainer (an $L$-layer Transformer encoder in the figure) that optimizes $\mathcal{L}_{\text{LC}}$ (Eq. 2) theoretically grounded by Theorem 1 and $\mathcal{L}_{\text{PC}}$ (Eq. 3) empirically guided by the supervised signals. Image modality is shown as an example.

For *permutation* perspective metrics, we formalize four empirical metrics (DEL, INS, NEG, POS) and re-formulate two existing ones (RP, IROF) in Table 1. DEL and INS respectively remove features from the original input $x$ or insert features into a baseline input $x^{\circ}$, using the area under the curve (AUC) of preservation effects as faithfulness scores; NEG and POS remove features in ascending order $\overleftarrow{\pi}$ or descending order $\pi$ until $t$-th removal leading to prediction changes significantly (e.g., class flips), with AUC used to quantify the over-all preservation effect; RP perturbs features in descending order of importance and averages the prediction drop; IROF uses the same order as RP and computes the mean area over the curve (AOC) of preservation effects across $N$ samples.

Although RP, IROF, DEL, INS, NEG, and POS evaluate permutation explanations in ways that differ substantially from FC, FE, INF, and MC, we theoretically show that they share an underlying consistency, and prove that $S_f^*$ in Proposition 1 can induce an optimal permutation explanation mapping on all six permutation-based faithfulness metrics.

**Theorem 1.** *Under the conditions of Proposition 1, given a fixed preservation effect $\Delta^-$ that is strictly decreasing with $\Delta$, let $\Pi_f^*(\cdot) = \mathfrak{P}[S_f^*(\cdot)]$ denote the permutation explanation mapping induced by $S_f^*$, then for any sample $x$, $\Pi_f^*(x)$ always achieve optimal faithfulness under the DEL, INS, NEG, POS, RP and IROF evaluation metrics.*

*Proof.* Please refer to Appendix A for the proof. $\qquad\square$

*Remark.* Through Proposition 1 and Theorem 1, we find that the saliency explanation mapping $S_f^*$ achieves optimal faithfulness under both types of explanation evaluation, indicating that diverse faithfulness metrics share a *common underlying mathematical essence*. Therefore, Equation (1) can be regarded as the consistent objective function of the *most faithful explanation mapping*.

We can construct a self-supervised optimization objective for the *Learning to Explain* paradigm by employing a Monte Carlo approximation of the ideal $S_f^*$. Let $\phi_\theta : \mathcal{X} \rightarrow [0,1]^n$ denote a neural network parameterized by $\theta$. Given a sample set $\mathcal{D} = \{x^{(i)}\}_{i=1}^{|\mathcal{D}|} \subseteq \mathcal{X}$, **DeepFaith** approximates the ideal most faithful explanation mapping $S_f^*$ by optimizing the $\underline{\text{L}}$ocal $\underline{\text{C}}$orrelation loss $\mathcal{L}_{\text{LC}}$:

$$\mathcal{L}_{\text{LC}}(\phi_\theta; \mathcal{D}, f) = \frac{1}{2} - \frac{1}{2|\mathcal{D}|} \sum_{x \in \mathcal{D}} \tau \left[ \left( \sum_{i \in \mathcal{I}_j} \phi_\theta(x)_i \right)_{j=1}^k, \left( \Delta[f(x), f(x \setminus \mathcal{I}_j)] \right)_{j=1}^k \right], \quad (2)$$

where $\Delta$, $\tau$ and $k$ are user-defined, and $\mathcal{I}_j \sim \mathcal{P}([n])$. Since we make no assumptions about the target model or task but instead start from explanation faithfulness itself, Equation (1) imposes strict requirements on $S_f^*$, making it an ideally defined yet practically intractable explanation mapping. However, our self-supervised objective (2) *does not attempt to solve $S_f^*$ directly*; rather, it provides a clear optimization direction for **DeepFaith** by approximating $S_f^*$.

---

**Algorithm 1** Dynamic Joint Optimization Strategy for **DeepFaith**

---

1: **Input:** Training dataset $\mathcal{D}$, target model $f$, supervised signals $\mathcal{Z}$, initial explainer parameters $\phi_\theta^{(0)}$, variance threshold $\epsilon$, monitoring window $e$, scaling factor $C \geq 1$, and learning rate $\eta$
2: **Initialize:** dynamic weight $\alpha^{(0)} \leftarrow 1$, convergence flag $CF = 0$, and set $t_0 = 0$
3: **while** not converged **do**
4:      $\mathcal{L}_{\mathrm{OBJ}}^{(t)} \leftarrow \alpha^{(t)} \mathcal{L}_{\mathrm{PC}}^{(t)} + (1 - \alpha^{(t)}) \mathcal{L}_{\mathrm{LC}}^{(t)}$     ▷ Compute total loss
5:      $\phi_\theta^{(t+1)} \leftarrow \phi_\theta^{(t)} - \eta \nabla_{\phi_\theta} \mathcal{L}_{\mathrm{OBJ}}^{(t)}$     ▷ Update explainer parameters
6:      $\sigma_{\mathrm{PC}}^2 \leftarrow \mathrm{Var}\big(\mathcal{L}_{\mathrm{PC}}^{(t-e+1)}, \ldots, \mathcal{L}_{\mathrm{PC}}^{(t)}\big)$     ▷ Compute variance of $\mathcal{L}_{\mathrm{PC}}$ over last $e$ iterations
7:      **if** $CF = 0$ **then**
8:          **if** $\sigma_{\mathrm{PC}}^2 < \epsilon$ **then**
9:              $CF \leftarrow 1, t_0 \leftarrow t$     ▷ Set Convergence Flag
10:          **end if**
11:      **end if**
12:      **if** $\sigma_{\mathrm{PC}}^2 > C\epsilon$ **then**
13:          $CF \leftarrow 0, \alpha^{(t+1)} \leftarrow 1$     ▷ Reset Convergence Flag
14:      **end if**
15:      **if** $CF = 1$ **then**
16:          $\alpha^{(t+1)} \leftarrow 1 - \frac{1}{1+\exp(-\frac{t-t_0}{C})}$     ▷ Gradually decrease $\alpha$
17:      **end if**
18: **end while**

---

# 4   HIGH-FAITHFULNESS SUPERVISED EXPLANATION SIGNAL

In this section, we propose, for the first time, the generation of high-quality supervised explanation signals for the *Learning to Explain* paradigm, based on curation by explanation faithfulness.

As illustrated in Figure 1, to increase the diversity of supervisory signals and collect more faithful explanations for each sample, we first aggregate multiple prior explanation methods to generate a set of *input-saliency explanation pairs* as supervised explanation signals. Given a sample set $\mathcal{D}$ and $K$ saliency explanation methods $\{S_f^{(i)}\}_{i=1}^K$ (e.g., Occlusion (Matthew D. Zeiler, 2013), Saliency (Simonyan et al., 2014), DeepLIFT (Shrikumar et al., 2017), Score-CAM (Wang et al., 2020) and Grad-CAM++ (Chattopadhyay et al., 2018)), we generate $K$ saliency explanation $\{S_f^{(j)}(x^{(i)})\}_{j=1}^K$ for each sample $x^{(i)}$. These explanations are then processed via deduplicating and filtering:

- Deduplicating: We compute the pairwise cosine similarity between $K$ saliency explanations of a given sample $x^{(i)}$ and identify duplicate groups based on a similarity threshold $\delta$. The first explanation in each group is retained, while the others are removed. After deduplicating, the number of distinct saliency explanations is denoted as $K_{\mathrm{dedup}}^{(i)} \leq K$. This step aims to prevent highly similar explanations from introducing bias into the training of the explainer and to enhance the diversity of the signals.

- Filtering: For each of the $K_{\mathrm{dedup}}^{(i)}$ retained explanations, we use all ten faithfulness metrics (the faithfulness of a saliency explanation can be evaluated from permutation perspective via $\mathfrak{P}$) to get their evaluation scores $(r_1, r_2, ..., r_{10})$. We determine a filtering threshold $(\bar{r}_1, \bar{r}_2, ..., \bar{r}_{10})$ by computing the $p$-quantile (or the $(1-p)$-quantile for metrics where lower is better) of all $K_{\mathrm{dedup}}^{(i)}$ scores under each metric. Finally, we retain $K_{\mathrm{filter}}^{(i)} \leq K_{\mathrm{dedup}}^{(i)}$ explanations satisfying $\forall j \in [10], r_j \geq \bar{r}_j$ (or $r_j \leq \bar{r}_j$ for metrics where lower is better).

After our explanation processing steps, the remained ones can be regarded as *high-quality supervised explanation signals*. For each input $x^{(i)}$, we replicate it $K_{\mathrm{filter}}^{(i)}$ times and pair each copy with its corresponding saliency explanation to construct the input–saliency explanation pair set

$$\mathcal{Z} = \left\{ \left( x^{(i)}, S_f^{(j)}(x^{(i)}) \right) \mid i \leq |\mathcal{D}|, j \in \left[ K_{\mathrm{filter}}^{(i)} \right] \right\}.$$

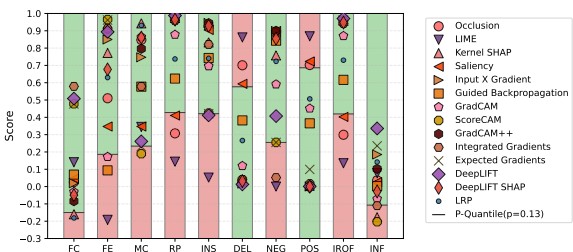 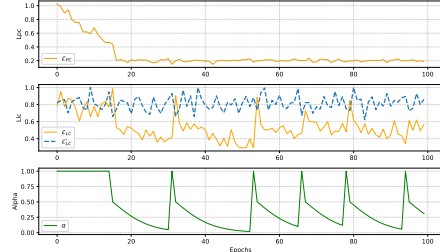

Figure 2: In the ImageNet+DeiT explanation task, we compute 10 faithfulness metrics for each explanation method on a single sample. Red and green regions denote the filtered-out range and retained range.

Figure 3: In the ImageNet+DeiT explanation task, the values of $\mathcal{L}_{PC}$, $\mathcal{L}_{LC}$, and $\alpha$ over 100 training epochs of **DeepFaith**. We also report the training dynamic of optimizing $\mathcal{L}_{LC}$ alone as $\mathcal{L}'_{LC}$.

**DeepFaith** optimizes the proximity between the explanations generated by $\phi_\theta$ and the high-quality saliency explanation through the Pattern Consistency loss $\mathcal{L}_{PC}$:

$$\mathcal{L}_{PC}(\phi_\theta; \mathcal{Z}) = \frac{1}{|\mathcal{Z}|} \sum_{(x,s) \in \mathcal{Z}} \left( 1 - \tau \left[ \phi_\theta(x), s \right] \right), \tag{3}$$

where $\tau$ can be any correlation measure and is not necessarily the same as the one used in Eq. (2).

## 5 FAITHFUL AMORTIZED EXPLAINER TRAINING STRATEGY

Many studies in the current *Learning to Explain* paradigm have shown that solely optimizing a self-supervised objective or merely fitting supervisory signals derived from prior explanations each has inherent limitations. However, jointly optimizing both objectives is not straightforward. This is because the local and global optima of the two objectives differ, leading to inconsistent gradient directions: optimizing one can easily degrade the other. To address this, we adopt a dynamic weighting-based strategy for explainer optimization, as illustrated in Algorithm 1. Let the dynamic weight be $\alpha \in [0, 1]$, the overall optimization objective of **DeepFaith** can be expressed as:

$$\mathcal{L}_{OBJ}(\phi_\theta; \mathcal{D}, f, \mathcal{Z}) = \alpha \mathcal{L}_{PC}(\phi_\theta; \mathcal{Z}) + (1 - \alpha) \mathcal{L}_{LC}(\phi_\theta; \mathcal{D}, f). \tag{4}$$

At the beginning of training, we set $\alpha = 1$ so that the explainer initially prioritizes acquiring basic explanatory capability by fitting the supervised signals. During training, we continuously monitor the value of $\mathcal{L}_{PC}$. If the variance of $\mathcal{L}_{PC}$ over $e$ consecutive iterations falls below a threshold $\epsilon$, we consider $\mathcal{L}_{PC}$ to have converged and gradually decrease $\alpha$ so that $\mathcal{L}_{LC}$ dominates the optimization, further improving explanation quality. To ensure that optimizing $\mathcal{L}_{LC}$ does not converge to a poor local minimum, the variance of $\mathcal{L}_{PC}$ continues to be monitored. If it exceeds $C\epsilon$, the optimization direction of $\mathcal{L}_{LC}$ is deemed to have deviated, and $\alpha = 1$ is reset to 1, allowing $\mathcal{L}_{PC}$ to again dominate the optimization.

## 6 EXPERIMENTS

In this section, we report the observations during the generation of the supervised explanation signals, as well as the performance of **DeepFaith** across various explanation tasks. We also provide ablation experiments to verify the necessity of combining $\mathcal{L}_{PC}$ and $\mathcal{L}_{LC}$, as well as the deduplicating and filtering of generating supervision signals. Visualizations of **DeepFaith** explanations and comparisons of runtime are provided in Appendices H and I. Our code is available at this Github Repository.

**Experimental Setting** To validate its capabilities across multiple target models and tasks, **DeepFaith** is tested on image, text, and tabular modalities using various model architectures. These dataset-model combinations yield diverse settings with varying complexity, forming a *comprehensive and challenging* benchmark for explanation quality. Dataset details are in Appendix B. All experiments were conducted on Ubuntu 22.04 with eight NVIDIA RTX A6000 GPUs.

Table 2: Comparison of average faithfulness between **DeepFaith** and other baseline methods across 12 explanation tasks. We report the average rank of each method under 10 faithfulness evaluation metrics, where **Red** denotes the optimal.

| Method | OCT | | | ImageNet | | | IMDb | | AGNews | | NAP | WCD |
|---|---|---|---|---|---|---|---|---|---|---|---|---|
| | DeiT | EfficientNet | ResNet | DeiT | EfficientNet | ResNet | LSTM | Transformer | LSTM | Transformer | MLP | MLP |
| DeepFaith (ours) | **3.4** | **2.9** | **4.1** | **4.4** | **4.4** | **3.3** | **2.3** | **2.1** | **2.9** | **2.7** | **1.8** | **1.8** |
| Integrated Grads | 7.8 | 7.6 | 4.8 | 6.4 | 7.0 | 5.4 | 3.3 | 5.6 | 4.9 | 5.9 | 2.8 | 5.2 |
| Gradient SHAP | N/A | N/A | N/A | N/A | N/A | N/A | 4.4 | 4.0 | **2.9** | 4.2 | 4.7 | 7.3 |
| DeepLIFT | 5.8 | 7.8 | 8.1 | 7.0 | 6.9 | 8.4 | 6.1 | 6.4 | 7.9 | 5.9 | 4.4 | 2.3 |
| Saliency | 13.2 | 11.0 | 12.8 | 10.7 | 11.1 | 10.6 | 5.2 | 5.9 | 4.7 | 5.8 | 2.8 | 4.9 |
| Occlusion | 8.5 | 6.5 | 8.4 | 8.9 | 9.6 | 10.9 | 4.6 | 3.6 | **2.9** | **2.7** | 3.3 | 5.9 |
| Feature Ablation | N/A | N/A | N/A | N/A | N/A | N/A | 6.4 | 5.1 | 6.6 | 8.5 | 3.5 | 4.5 |
| LIME | 12.3 | 8.1 | 9.9 | 10.7 | 6.6 | 8.5 | 7.7 | 6.8 | 4.6 | 4.5 | 4.7 | 2.7 |
| Kernel SHAP | 4.2 | 10.9 | 12.1 | 7.0 | 5.9 | 8.9 | 5.0 | 5.5 | 6.4 | 3.9 | 3.9 | 8.9 |
| Input × Gradient | 5.7 | 12.3 | 12.2 | 5.3 | 12.9 | 10.7 | N/A | N/A | N/A | N/A | N/A | N/A |
| Guided Backprop | 12.3 | 6.5 | 7.6 | 11.4 | 10.3 | 10.4 | N/A | N/A | N/A | N/A | N/A | N/A |
| Grad-CAM | 8.6 | 8.2 | 7.6 | 11.9 | 6.6 | 7.0 | N/A | N/A | N/A | N/A | N/A | N/A |
| Score-CAM | 7.0 | 7.9 | 6.0 | 5.8 | 7.2 | 7.1 | N/A | N/A | N/A | N/A | N/A | N/A |
| Grad-CAM++ | 5.0 | 10.0 | 7.4 | 4.9 | 7.8 | 8.3 | N/A | N/A | N/A | N/A | N/A | N/A |
| Expected Grads | 6.9 | 8.0 | 7.1 | 7.1 | 9.5 | 6.4 | N/A | N/A | N/A | N/A | N/A | N/A |
| DeepLIFT SHAP | 6.9 | 7.3 | 5.6 | 7.5 | 8.7 | 9.3 | N/A | N/A | N/A | N/A | N/A | N/A |
| LRP | 12.0 | 4.5 | 5.4 | 10.2 | 5.0 | 4.5 | N/A | N/A | N/A | N/A | N/A | N/A |

- **Image modality:** Following Latec (Klein et al., 2024), we use ImageNet (Deng et al., 2009) and UCSD OCT Retina (OCT) (Kermany et al., 2018), explaining ResNet50 (He et al., 2016), EfficientNetb0 (Tan & Le, 2019), and DeiT (Touvron et al., 2024).
- **Text modality:** IMDb Movie Review (IMDb) (Maas et al., 2011) and AGNews (Zhang et al., 2016) are used with LSTM and vanilla Transformer (Vaswani et al., 2017).
- **Tabular modality:** We use NHANES Age Prediction (NAP) (National Center for Health Statistics, 2019) and Wholesale Customers Data (WCD) (Cardoso, 2013) from UCI, with MLP-based predictors.

For the image modality, we generate supervised signals from and compare against the following baseline methods: Occlusion, LIME (Ribeiro et al., 2016), Kernel SHAP and DeepLIFT SHAP (Lundberg & Allen, 2017), Saliency, Input × Gradient (Shrikumar et al., 2017), Guided Backprop (Springenberg et al., 2015), Grad-CAM (Selvaraju et al., 2017), Score-CAM, Grad-CAM++, Integrated Grads (Sundararajan et al., 2017), Expected Grads (Erion et al., 2021), DeepLIFT, and LRP (Binder et al., 2016). For the text and tabular modalities, we adopt Integrated Grads, Gradient SHAP (Lundberg & Allen, 2017), DeepLIFT, Saliency, Occlusion, Feature Ablation (Kokhlikyan et al., 2020), LIME, and Kernel SHAP. Parameter settings are listed in Appendix C.

**Generating Supervised Explanation Signals** Given a specific dataset and target model, **DeepFaith** generates high-quality input-saliency explanation pairs before training. Taking the task of explaining DeiT's predictions on ImageNet as an example, we use 14 widely adopted explanation methods from Captum (Kokhlikyan et al., 2020) to generate patch-level explanations for 20,000 validation samples. Each explanation is evaluated using 10 faithfulness metrics (detailed in Appendix D).

Figure 2 illustrates the faithfulness-based filtering process of the supervised explanation signals for one sample. For each evaluation metric, we compute the $p$-quantile and remove explanations deemed unfaithful by any of the metrics. Detailed processes for all explanation tasks are provided in Appendix E.

**Training Faithful Saliency Explainer** We use a multi-layer Transformer Encoder as the explainer for its strength in processing sequential inputs. It encodes patch-based images, tokenized text, or tabular rows, followed by a normalized linear layer projecting to an $n$-dimensional saliency explanation. Task-specific configurations and hyperparameters of **DeepFaith** are in Appendix F.

We split the supervised explanation signals into training (80%) and test (20%) sets and train the explainer, with the training loss curves shown in Figure 3. During the early stage of training, the explainer optimizes $\mathcal{L}_{PC}$ to fit the prior explanation methods, and subsequently maintains a low $\mathcal{L}_{PC}$ throughout training under the constraints of the training strategy described in Algorithm 1. Once $\mathcal{L}_{PC}$ is confirmed to have converged, the explainer gradually optimizes $\mathcal{L}_{LC}$, intermittently setting $\alpha = 1$ to adjust the optimization direction. Despite oscillations, $\mathcal{L}_{LC}$ steadily converges over this process, whereas optimizing the local-correlation loss alone, $\mathcal{L}'_{LC}$ exhibits pronounced oscillations and fails to converge.

Table 3: **DeepFaith** is compared against other *Learning to Explain* methods across four different settings. The table reports the average test set scores on ten faithfulness evaluation metrics, as well as the average rank of each explainer across all metrics under each setting.

| Setting | Explainer | FC↑ | FE↑ | MC↑ | RP↑ | INS↑ | DEL↓ | NEG↑ | POS↓ | IROF↑ | INF↑ | Mean Rank |
|---|---|---|---|---|---|---|---|---|---|---|---|---|
| NAP+MLP | DeepFaith | **0.788** | **0.763** | **0.952** | **0.957** | 0.844 | **0.031** | 0.770 | **0.031** | **0.844** | **0.238** | **1.3** |
| | L2X | 0.129 | 0.141 | 0.899 | 0.346 | 0.832 | 0.749 | 0.819 | 0.748 | 0.302 | 0.142 | 3.5 |
| | CXPlain | 0.017 | 0.308 | 0.501 | 0.290 | 0.153 | 0.647 | 0.722 | 0.301 | 0.445 | 0.143 | 3.7 |
| | FastSHAP | 0.071 | 0.097 | 0.800 | 0.149 | **0.849** | 0.714 | **0.837** | 0.710 | 0.126 | 0.041 | 3.7 |
| | VerT | 0.772 | 0.484 | 0.560 | 0.454 | 0.564 | 0.467 | 0.518 | 0.551 | 0.603 | 0.160 | 2.8 |
| AgNews+LSTM | DeepFaith | 0.363 | **0.597** | **0.629** | 0.648 | **0.919** | 0.197 | **0.906** | 0.256 | 0.650 | **0.275** | **1.6** |
| | L2X | 0.012 | 0.303 | 0.336 | 0.367 | 0.765 | 0.621 | 0.740 | 0.618 | 0.320 | 0.071 | 4.2 |
| | CXPlain | 0.097 | 0.258 | 0.402 | 0.328 | 0.589 | **0.189** | 0.613 | **0.189** | 0.371 | 0.030 | 3.6 |
| | FastSHAP | **0.419** | 0.251 | 0.500 | 0.505 | 0.800 | 0.546 | 0.780 | 0.551 | 0.395 | 0.062 | 3.2 |
| | VerT | 0.077 | 0.519 | 0.526 | **0.710** | 0.838 | 0.362 | 0.423 | 0.204 | **0.762** | 0.092 | 2.4 |
| OCT+ResNet | DeepFaith | **0.135** | **0.534** | **0.942** | **0.744** | **0.863** | **0.248** | 0.655 | 0.242 | **0.742** | 0.015 | **1.4** |
| | L2X | 0.015 | 0.121 | 0.393 | 0.501 | 0.642 | 0.505 | 0.644 | 0.433 | 0.482 | 0.009 | 3.5 |
| | CXPlain | 0.004 | 0.239 | 0.611 | 0.356 | 0.554 | 0.525 | 0.508 | 0.513 | 0.349 | 0.008 | 4.2 |
| | FastSHAP | 0.013 | 0.119 | 0.211 | 0.302 | 0.808 | 0.698 | 0.605 | 0.656 | 0.297 | 0.022 | 4.1 |
| | VerT | 0.015 | 0.397 | 0.783 | 0.540 | 0.763 | 0.466 | **0.707** | **0.226** | 0.528 | **0.028** | 1.8 |
| ImageNet+DeiT | DeepFaith | **0.026** | **0.447** | **0.884** | **0.486** | **0.568** | **0.127** | 0.417 | 0.295 | **0.672** | 0.014 | **1.6** |
| | L2X | 0.004 | 0.101 | 0.267 | 0.198 | 0.486 | 0.526 | **0.520** | 0.481 | 0.385 | 0.014 | 3.9 |
| | CXPlain | 0.003 | 0.023 | 0.341 | 0.237 | 0.186 | 0.172 | 0.211 | **0.192** | 0.431 | 0.017 | 3.6 |
| | FastSHAP | 0.006 | 0.012 | 0.227 | 0.358 | 0.451 | 0.439 | 0.485 | 0.457 | 0.461 | **0.027** | 3.2 |
| | VerT | 0.005 | 0.131 | 0.505 | 0.414 | 0.363 | 0.323 | 0.365 | 0.367 | 0.588 | 0.019 | 2.7 |

For each explanation task, we compare the faithfulness of **DeepFaith** against other baseline explanation methods. Each explanation is scored using all ten faithfulness metrics and averaged across all test samples (see Appendix G for full results). To concisely summarize the overall explanation quality of each method, we rank all explanation methods under each metric and report their average rankings. Table 2 presents the evaluation results across all explanation tasks. **DeepFaith** consistently achieves the highest faithfulness across various modalities, demonstrating that our method enables the explainer to *surpass the faithfulness of the baseline methods used* during training and remains effective for complex models and high-dimensional feature tasks.

To further validate the advantages of **DeepFaith** over prior *Learning to Explain* methods, we conducted comparisons on four explanation tasks (NAP+MLP, AGNews+LSTM, OCT+ResNet, and ImageNet+DeiT) evaluating DeepFaith alongside four explainer baselines applicable across multiple modalities: VerT (Bhalla et al., 2023), L2X (Chen et al., 2018), CXPlain (Schwab & Karlen, 2019), and FastSHAP (Jethani et al., 2022). Table 3 reports the averaged test-set scores across ten faithfulness metrics.

In every task, **DeepFaith** achieves the highest faithfulness on multiple metrics and consistently obtains the best average ranking. These results demonstrate that our method provides a clear improvement over existing *Learning to Explain* approaches.

**Ablation Study** We selected four representative explanation tasks to organize ablation experiments, aiming to evaluate the individual contributions of each loss in **DeepFaith** as well as the effects of deduplicating and filtering when generating supervised explanation signals. For each setting, the explainer is trained for an equal number of epochs (100) under five ablation configurations: using the full loss function ($\mathcal{L}_{\text{OBJ}}$), only the pattern consistency loss ($\mathcal{L}_{\text{PC}}$), only the local consistency loss ($\mathcal{L}_{\text{LC}}$), using both losses with deduplicating only ($\mathcal{L}_{\text{OBJ}}^{\text{d}}$), and with filtering only ($\mathcal{L}_{\text{OBJ}}^{\text{f}}$).

Table 4 presents the results of the ablation experiments, revealing a consistent pattern across all four settings: **DeepFaith** trained with $\mathcal{L}_{\text{OBJ}}$ achieves the highest faithfulness across all metrics. When only $\mathcal{L}_{\text{PC}}$ or $\mathcal{L}_{\text{LC}}$ is used, **DeepFaith** degrades into either a prior-explanation-driven or a self-supervised *Learning to Explain* method, with the latter exhibiting poor faithfulness due to optimization convergence difficulties. This aligns with the design rationale of our dynamic joint optimization strategy. In the ablations concerning the generation of supervised explanation signals, applying only deduplicating or only filtering leads to reduced explainer performance, with the former showing particularly lower faithfulness due to the lack of faithfulness-based selection, thereby validating the effectiveness of our proposed approach.

**Sensitivity Study of Hyperparameters** We introduces several hyperparameters in different components of **DeepFaith**: the subset sampling size $k$ in the computation of $\mathcal{L}_{\text{LC}}$; the similarity threshold

Table 4: Ablation study of **DeepFaith** on explanation tasks across four different settings. The table reports the average scores over test set on ten faithfulness evaluation metrics, where $\mathcal{L}_{\text{OBJ}}$ denotes the explainer trained with both loss terms.

| Setting | Ablation | FC ↑ | FE ↑ | MC ↑ | RP ↑ | INS ↑ | DEL ↓ | NEG ↑ | POS ↓ | IROF ↑ | INF ↑ |
|---|---|---|---|---|---|---|---|---|---|---|---|
| OCT+DeiT | $\mathcal{L}_{\text{OBJ}}$ | **0.217** | **0.475** | **0.897** | **0.643** | **0.944** | **0.356** | **0.917** | **0.368** | **0.638** | **0.089** |
| | $\mathcal{L}_{\text{PC}}$ | 0.032 | 0.231 | 0.655 | 0.540 | 0.913 | 0.463 | 0.904 | 0.521 | 0.534 | 0.031 |
| | $\mathcal{L}_{\text{LC}}$ | 0.101 | 0.104 | 0.240 | 0.169 | 0.763 | 0.830 | 0.809 | 0.813 | 0.162 | 0.023 |
| | $\mathcal{L}_{\text{OBJ}}^{\text{d}}$ | 0.097 | 0.341 | 0.669 | 0.585 | 0.923 | 0.447 | 0.906 | 0.476 | 0.552 | 0.059 |
| | $\mathcal{L}_{\text{OBJ}}^{\text{f}}$ | 0.156 | 0.392 | 0.711 | 0.602 | 0.931 | 0.421 | 0.910 | 0.444 | 0.576 | 0.067 |
| ImageNet+DeiT | $\mathcal{L}_{\text{OBJ}}$ | **0.026** | **0.447** | **0.884** | **0.486** | **0.568** | **0.127** | **0.417** | **0.295** | **0.672** | **0.014** |
| | $\mathcal{L}_{\text{PC}}$ | 0.022 | 0.364 | 0.823 | 0.456 | 0.501 | 0.185 | 0.406 | 0.366 | 0.638 | 0.008 |
| | $\mathcal{L}_{\text{LC}}$ | -0.047 | -0.051 | 0.033 | 0.373 | 0.552 | 0.380 | 0.397 | 0.414 | 0.493 | -0.037 |
| | $\mathcal{L}_{\text{OBJ}}^{\text{d}}$ | 0.023 | 0.390 | 0.839 | 0.464 | 0.523 | 0.170 | 0.409 | 0.348 | 0.647 | 0.010 |
| | $\mathcal{L}_{\text{OBJ}}^{\text{f}}$ | 0.024 | 0.406 | 0.856 | 0.472 | 0.545 | 0.156 | 0.412 | 0.331 | 0.655 | 0.011 |
| IMDb+Transformer | $\mathcal{L}_{\text{OBJ}}$ | **0.162** | **0.495** | **0.203** | **0.759** | **0.806** | **0.189** | **0.799** | **0.205** | **0.742** | **0.047** |
| | $\mathcal{L}_{\text{PC}}$ | 0.058 | 0.358 | 0.195 | 0.718 | 0.784 | 0.192 | 0.775 | 0.344 | 0.655 | 0.038 |
| | $\mathcal{L}_{\text{LC}}$ | 0.023 | 0.235 | 0.167 | 0.316 | 0.667 | 0.708 | 0.738 | 0.652 | 0.223 | 0.013 |
| | $\mathcal{L}_{\text{OBJ}}^{\text{d}}$ | 0.089 | 0.402 | 0.196 | 0.733 | 0.791 | 0.191 | 0.783 | 0.310 | 0.678 | 0.040 |
| | $\mathcal{L}_{\text{OBJ}}^{\text{f}}$ | 0.115 | 0.427 | 0.198 | 0.746 | 0.796 | 0.190 | 0.789 | 0.276 | 0.701 | 0.042 |
| NAP+MLP | $\mathcal{L}_{\text{OBJ}}$ | **0.788** | **0.763** | **0.952** | **0.957** | **0.844** | **0.031** | **0.770** | **0.031** | **0.844** | **0.238** |
| | $\mathcal{L}_{\text{PC}}$ | 0.674 | 0.671 | 0.558 | 0.424 | 0.358 | 0.514 | 0.227 | 0.541 | 0.361 | 0.025 |
| | $\mathcal{L}_{\text{LC}}$ | 0.748 | 0.515 | 0.535 | 0.426 | 0.360 | 0.442 | 0.512 | 0.124 | 0.638 | 0.135 |
| | $\mathcal{L}_{\text{OBJ}}^{\text{d}}$ | 0.703 | 0.694 | 0.623 | 0.552 | 0.479 | 0.393 | 0.363 | 0.414 | 0.482 | 0.079 |
| | $\mathcal{L}_{\text{OBJ}}^{\text{f}}$ | 0.731 | 0.717 | 0.755 | 0.690 | 0.601 | 0.273 | 0.498 | 0.286 | 0.603 | 0.132 |

$\delta$ and filtering quantile $p$ when constructing the supervised explanation set; and the hyperparameters $\epsilon$, $e$, and $C$ in Algorithm 1 for explainer training. Studying the sensitivity of these hyperparameters is essential for understanding whether **DeepFaith** can generalize stably under different parameter settings, thereby reducing reliance on manual tuning.

Using the OCT+ResNet explanation task as an example, we conducted a comprehensive hyperparameter sensitivity analysis. The detailed experimental configurations and results are provided in Appendix J. The findings show that varying these hyperparameters leads to only minor changes in performance, and in practice it suffices to keep them within reasonable ranges. This demonstrates that **DeepFaith** exhibits low sensitivity to its hyperparameters and maintains stable generalization.

## 7 CONCLUSION

*Learning to Explain* paradigm has charted a promising yet challenging path for XAI researchers, and **DeepFaith** takes a crucial step along this trajectory. From the perspective of explanation faithfulness, we propose improvements to the two main existing approaches within the *Learning to Explain* paradigm and introduce a strategy to integrate them, bridging inherent limitations and offering a potential systematic solution. The baseline explanation methods used by **DeepFaith** to generate supervised signals can be replaced with any newly proposed techniques, and the explainer architecture can comprise any deep neural network capable of handling sequential inputs. This flexibility implies that **DeepFaith** can continuously evolve alongside advancements in XAI.

## ACKNOWLEDGMENTS

This work was supported by the Joint Funds of the National Natural Science Foundation of China under (Grant No. U22A2099), the General Program of the National Natural Science Foundation of China under (Grant No. 62376028), the Excellent Young Scientists Fund (Overseas) of the National Natural Science Foundation of China, the National Key Scientific Instruments and Equipment Development Project under (Grant No. 62427808), and the National Key Research and Development Program of China under (Grant No. 2022YFB2703100).

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

# A  PROOF OF PROPOSITION 1 AND THEOREM 1

## A.1  PROOF OF PROPOSITION 1

*Proof.* By definition (Eq. 1), $S_f^*$ maximizes the correlation

$$\tau\left[\left(\sum_{j\in\mathcal{I}_i} S_f(x)_j\right)_{i=1}^N, \left(\Delta[f(x), f(x\setminus\mathcal{I}_i)]\right)_{i=1}^N\right]$$

over all valid saliency mappings $S_f$, for every choice of sample $x\in\mathcal{X}$ and every collection of index sets $\{\mathcal{I}_i\}_{i=1}^N$. We show optimality for each metric in turn.

Faithfulness Correlation (FC) for a given $s$ is precisely the correlation $\tau$ evaluated over the full family of index sets $\mathcal{I}\subseteq[n]$. Since $S_f^*$ maximizes $\tau$ for every such collection, no other saliency mapping can achieve a strictly larger FC value. Hence $S_f^*$ attains the optimal FC.

Faithfulness Estimate (FE) is the empirical correlation $\tau$ computed across a dataset $\{x^{(i)}, \mathcal{I}_i\}_{i=1}^N$ using the mapping $S_f$. By Eq. (1), $S_f^*$ maximizes $\tau$ for every possible choice of $\{x^{(i)}, \mathcal{I}_i\}_{i=1}^N$; therefore, $S_f^*$ yields the largest possible FE on any finite sample set and thus is FE-optimal.

Monotonicity Correlation (MC) and Infidelity (INF) share the same functional form as FC/FE: they are correlations $\tau$ computed over particular collections of index sets (MC typically over ordered or monotone collections, INF over random draws $\mathcal{I}_i\sim\mathcal{P}([n])$). Since $S_f^*$ maximizes $\tau$ for all choices of collections (including those specific to MC and INF), it must also maximize MC and INF. $\square$

## A.2  PROOF OF THEOREM 1

*Proof.* $\forall\Pi_f$, given an input sample $x\in\mathcal{X}$, let $\pi=\Pi_f(x)$ and $\pi^*=\Pi_f^*(x)$ denote the permutation explanations generated by different mappings, and let $s^*=S_f^*(x)$ be the corresponding optimal saliency explanation. In addition, we denote $\Delta_{f,x}(\mathcal{I})=\Delta[f(x), f(x\setminus\mathcal{I})]$ for simplicity.

Given any two index sets $\mathcal{I}_a, \mathcal{I}_b$ satisfying $\sum_{j\in\mathcal{I}_a} s_j^* \geq \sum_{j\in\mathcal{I}_b} s_j^*$ (i.e., the total saliency assigned to $\mathcal{I}_a$ is no less than that of $\mathcal{I}_b$), suppose *for contradiction* that $\Delta_{f,x}(\mathcal{I}_a) < \Delta_{f,x}(\mathcal{I}_b)$, meaning that removing $\mathcal{I}_b$ causes a larger change in the model output despite receiving a lower total saliency score.

Then, by the definition of the correlation measure $\tau$, there must exist another saliency assignment $s$ such that $\sum_{j\in\mathcal{I}_a} s_j < \sum_{j\in\mathcal{I}_b} s_j$ while achieving a higher correlation between the saliency ordering and the corresponding perturbation effects, i.e.,

$$\tau\left[\begin{pmatrix}\sum_{j\in\mathcal{I}_a} s_j\\ \sum_{j\in\mathcal{I}_b} s_j\end{pmatrix}, \begin{pmatrix}\Delta_{f,x}(\mathcal{I}_a)\\ \Delta_{f,x}(\mathcal{I}_b)\end{pmatrix}\right] > \tau\left[\begin{pmatrix}\sum_{j\in\mathcal{I}_a} s_j^*\\ \sum_{j\in\mathcal{I}_b} s_j^*\end{pmatrix}, \begin{pmatrix}\Delta_{f,x}(\mathcal{I}_a)\\ \Delta_{f,x}(\mathcal{I}_b)\end{pmatrix}\right],$$

which directly contradicts the optimality of $S_f^*$ as defined in Eq. (1), since $S_f^*$ is the maximizer of the same correlation measure over all valid saliency mappings. Therefore, the assumption $\Delta_{f,x}(\mathcal{I}_a) < \Delta f, x(\mathcal{I}_b)$ must be false, and we can conclude that

$$\forall\,\mathcal{I}_a, \mathcal{I}_b, \sum_{j\in\mathcal{I}_a} s_j^* \geq \sum_{j\in\mathcal{I}_b} s_j^* \Rightarrow \Delta_{f,x}(\mathcal{I}_a) \geq \Delta_{f,x}(\mathcal{I}_b).$$

Now consider the index sets $\bigcup_{j=1}^i \pi^*(j)$ and $\bigcup_{j=1}^i \pi(j)$, which correspond to the top-$i$ features selected by $\pi^*$ and $\pi$, respectively. Since the permutation explanation $\pi^*=\mathfrak{P}(s^*)$ is obtained by sorting features in descending order of their saliency scores, it follows that for all $i\leq n$, we have $\sum_{j=1}^i s_{\pi^*(j)}^* \geq \sum_{j=1}^i s_{\pi(j)}^*$, i.e., the cumulative saliency of the top-$i$ features selected by $\pi^*$ is always at least as large as that selected by any other permutation $\pi$. By the monotonic relationship just proven, this implies

$$\Delta_{f,x}\left(\bigcup_{j=1}^i \pi^*(j)\right) \geq \Delta_{f,x}\left(\bigcup_{j=1}^i \pi(j)\right).$$

By aggregating this result over samples $\{x^{(i)}\}_{i=1}^N$, we can directly get $\mathrm{RP}(\Pi_f^*; \{x^{(i)}\}_{i=1}^N, f) \geq \mathrm{RP}(\Pi_f; \{x^{(i)}\}_{i=1}^N, f)$. Since $\Delta^-$ is strictly decreasing with respect to $\Delta$, i.e., a larger perturbation

effect corresponds to a smaller $\Delta^-$ value, we have

$$\forall \, \mathcal{I}_a, \mathcal{I}_b, \sum_{j \in \mathcal{I}_a} s_j^* \geq \sum_{j \in \mathcal{I}_b} s_j^* \Rightarrow \Delta_{f,x}^-(\mathcal{I}_a) \leq \Delta_{f,x}^-(\mathcal{I}_b),$$

from which it follows immediately that $\mathrm{DEL}(\pi^*; x, f) \leq \mathrm{DEL}(\pi; x, f)$ and $\mathrm{POS}(\pi^*; x, f) \leq \mathrm{POS}(\pi; x, f)$; by the same way, one can derive $\mathrm{NEG}(\pi^*; x, f) \geq \mathrm{NEG}(\pi; x, f)$ and $\mathrm{IROF}(\Pi_f^*; \{x^{(i)}\}_{i=1}^N, f) \geq \mathrm{IROF}(\Pi_f; \{x^{(i)}\}_{i=1}^N, f)$.

Finally, given a baseline input $x^\circ$ representing the uninformative state, we have the identity

$$x^\circ \cup \bigcup_{j=1}^{i} \pi(j) = x \setminus \bigcup_{j=1}^{n-i} \overleftarrow{\pi},$$

which establishes the equivalence between the insertion and deletion processes. Together with the above ordering property, this leads to $\mathrm{INS}(\pi^*; x, f) \geq \mathrm{INS}(\pi; x, f)$. □

## B  DATASETS INFORMATION

Table 5: **DeepFaith** performs explanation tasks on datasets from three modalities. In NAP, the model predicts *age group* from body measurements; in WCD, it forecasts *Channel* using product sales data.

| Dataset | Sample Num | Description |
|---------|-----------|-------------|
| ImageNet | 20,000 | ImageNet is a large-scale visual dataset encompassing real-world concepts such as animals, objects, and scenes, serving as a foundational dataset in the era of deep learning. |
| OCT | 1,000 | The OCT dataset comprises layered structural images of tissues such as the retina or cornea, used for medical image analysis and disease diagnosis, typically including high-resolution cross-sectional or 3D volumetric scan data. |
| AGNews | 127,600 | AGNews is a news article classification dataset containing English news headlines and content, covering four major categories (World, Sports, Business, Science), commonly used for text classification tasks. |
| IMDb | 50,000 | IMDb is a movie review sentiment analysis dataset containing English reviews labeled with binary sentiment (positive/negative). |
| NAP | 2,278 | The National Health and Nutrition Examination Survey (NHANES), administered by the Centers for Disease Control and Prevention, collects extensive health and nutritional information from a diverse U.S. population. |
| WCD | 440 | The data set refers to clients of a wholesale distributor. It includes the annual spending in monetary units (m.u.) on diverse product categories. |

Table 5 presents the explanation tasks of **DeepFaith** across three distinct data modalities, including ImageNet (Deng et al., 2009) and OCT (Kermany et al., 2018) for image-based tasks, AGNews (Zhang et al., 2016) and IMDb (Maas et al., 2011) for text classification and sentiment analysis, as well as NAP (National Center for Health Statistics, 2019) and WCD (Cardoso, 2013) for tabular prediction problems related to health and consumer behavior.

Specifically, ImageNet is employed for large-scale image recognition and object classification, while OCT is used for medical image analysis and disease-related pattern identification. AGNews and IMDb respectively support topic classification and sentiment analysis in the NLP domain. Meanwhile, NAP and WCD capture structured information in public health and retail contexts, enabling the evaluation of explanation quality on tabular data. Together, these datasets form a diverse and representative benchmark suite, allowing us to systematically assess the generality, robustness, and cross-modality applicability of explanation methods such as **DeepFaith**.

## C  Baseline Explanation Methods

Table 6: Parameters of baseline explanation methods for experiments on text and tabular modality, as well as whether each method is model-agnostic.

| Method | Parameters | Model Agnostic |
| --- | --- | --- |
| Integrated Grads | num steps=20, baseline=0 | True |
| Gradient SHAP | num samples=5 | True |
| DeepLIFT | baselines=0, eps=1e-9 | True |
| Saliency | None | True |
| Occlusion | strides=1, sliding window=1 | True |
| Feature Ablation | perturbations per eval=1 | True |
| LIME | num samples=100, perturbations per eval=1 | True |
| Kernel SHAP | num samples=100, perturbations per eval=1 | True |

Table 7: Parameters of baseline explanation methods for experiments on the image modality, as well as whether each method is model-agnostic.

| Method | Parameters | Model Agnostic |
| --- | --- | --- |
| Occlusion | strides=25, sliding window=50 | True |
| LIME | num samples=1000, perturbations per eval=5 | True |
| Kernel SHAP | num samples=1000, perturbations per eval=5 | True |
| Saliency | None | True |
| Input × Gradient | None | True |
| Guided Backprop | None | True |
| Grad-CAM | None | False |
| Score-CAM | None | False |
| Grad-CAM++ | None | False |
| Integrated Grads | num steps=30, baselines=0 | True |
| Expected Grads | num samples=40, stdevs=0.001 | True |
| DeepLIFT | baselines=0, eps=1e-9 | True |
| DeepLIFT SHAP | None | True |
| LRP | eps=1e-4, gamma=0.25 | False |

Tables 7 and 6 detail the parameters for each baseline explanation method used across the three modalities. For Occlusion  (Matthew D. Zeiler, 2013), the *sliding window* and *strides* must be specified. LIME  (Ribeiro et al., 2016) and Kernel SHAP require setting the *num samples* and the number of perturbed features per evaluation. For Integrated Grads  (Sundararajan et al., 2017), *num steps* sets the discretization granularity along the integration path from the baseline to the input, while *baselines* defines the reference input from which integration starts. Expected Grads  (Erion et al., 2021) requires both the *num samples* and *stdevs*, the standard deviation of Gaussian noise added to each sampled baseline. DeepLIFT  (Shrikumar et al., 2017) attributes contributions relative to *baselines* input, with *eps* preventing division-by-zero. In LRP  (Binder et al., 2016), *eps* serves the same numerical stability purpose, and *gamma* adjusts the amplification of positive contributions. Gradient SHAP  (Lundberg & Allen, 2017) only requires specifying *num samples*, while Feature Ablation  (Kokhlikyan et al., 2020) requires defining the number of perturbed features. For Saliency (Simonyan et al., 2014), Input × Gradient  (Shrikumar et al., 2017), Guided Backprop  (Springenberg et al., 2015), Grad-CAM  (Selvaraju et al., 2017), Score-CAM  (Wang et al., 2020), Grad-CAM++ (Chattopadhyay et al., 2018), and DeepLIFT SHAP  (Lundberg & Allen, 2017), we adopt the default parameters provided by Captum  (Kokhlikyan et al., 2020).

# D   FAITHFULNESS EVALUATION METRICS

Table 8: Variants of the perturbation effect $\Delta$, presevation effect $\Delta^-$, and correlation $\tau$, along with their corresponding formulations and descriptions.

| Function | Variant | Formula | Description |
|---|---|---|---|
| $\Delta[f(x), f(x \setminus \mathcal{I})]$ | $\Delta_{\mathrm{minus}}$ | $f(x) - f(x \setminus \mathcal{I})$ | Difference in model prediction before and after perturbation. |
| | $\Delta_{\mathrm{variance}}$ | $\mathrm{Var}\left[(f(x) - f(x \setminus \mathcal{I}_i))_{i=1}^N\right]$ | Variance of the predicted class score under perturbations. |
| $\Delta^-[f(x), f(x \setminus \mathcal{I})]$ | $\Delta^-_{\mathrm{target}}$ | $f(x \setminus \mathcal{I})$ | Prediction value retained after perturbation. |
| | $\Delta^-_{\mathrm{ratio}}$ | $f(x \setminus \mathcal{I})/f(x)$ | Ratio of prediction values before and after perturbation. |
| $\tau[(a_i)_{i=1}^N, (b_i)_{i=1}^N]$ | $\tau_{\mathrm{pearson}}$ | $(a - \bar{a})^\top (b - \bar{b})/\|a - \bar{a}\|_2\|b - \bar{b}\|_2$ | Classical Pearson correlation coefficient. |
| | $\tau_{\mathrm{spearman}}$ | $\tau_{\mathrm{pearson}}[\mathfrak{P}(a), \mathfrak{P}(b)]$ | Classical Spearman correlation. coefficient |
| | $\tau_{\mathrm{mse}}$ | $1 - \frac{1}{2N}\sum_{i=1}^N (a_i - b_i)^2$ | Classical mean squared error. |

Table 8 reports several specific forms of the perturbation effect $\Delta$, preservation effect $\Delta^-$, and correlation $\tau$ used in our faithfulness evaluation methods. Note that we assume perturbing the originally predicted features will lead to a decrease in the model output; therefore, $\Delta^-_{\mathrm{ratio}} \in [0, 1]$.

Table 9: Parameters of ten faithfulness metrics for three explanation tasks on ImageNet.

| Metric | $\Delta$ | $\Delta^-$ | $\tau$ | $|\mathcal{I}_i|$ | $N$ |
|---|---|---|---|---|---|
| FC | $\Delta_{\mathrm{minus}}$ | – | $\tau_{\mathrm{pearson}}$ | 3136 | 50 |
| FE | $\Delta_{\mathrm{minus}}$ | – | $\tau_{\mathrm{pearson}}$ | – | – |
| MC | $\Delta_{\mathrm{variance}}$ | – | $\tau_{\mathrm{spearman}}$ | – | – |
| RP | $\Delta_{\mathrm{minus}}$ | – | – | – | – |
| INS | – | $\Delta^-_{\mathrm{target}}$ | – | 3136 | – |
| DEL | – | $\Delta^-_{\mathrm{target}}$ | – | 3136 | – |
| NEG | – | $\Delta^-_{\mathrm{target}}$ | – | 3136 | – |
| POS | – | $\Delta^-_{\mathrm{target}}$ | – | 3136 | – |
| IROF | – | $\Delta^-_{\mathrm{ratio}}$ | – | 3136 | – |
| INF | $\Delta_{\mathrm{minus}}$ | – | $\tau_{\mathrm{pearson}}$ | – | 60 |

Table 10: Parameters of ten faithfulness metrics for three explanation tasks on OCT.

| Metric | $\Delta$ | $\Delta^-$ | $\tau$ | $|\mathcal{I}_i|$ | $N$ |
|---|---|---|---|---|---|
| FC | $\Delta_{\mathrm{minus}}$ | – | $\tau_{\mathrm{pearson}}$ | 3136 | 100 |
| FE | $\Delta_{\mathrm{minus}}$ | – | $\tau_{\mathrm{pearson}}$ | – | – |
| MC | $\Delta_{\mathrm{variance}}$ | – | $\tau_{\mathrm{spearman}}$ | – | – |
| RP | $\Delta_{\mathrm{minus}}$ | – | – | – | – |
| INS | – | $\Delta^-_{\mathrm{target}}$ | – | 3136 | – |
| DEL | – | $\Delta^-_{\mathrm{target}}$ | – | 3136 | – |
| NEG | – | $\Delta^-_{\mathrm{target}}$ | – | 3136 | – |
| POS | – | $\Delta^-_{\mathrm{target}}$ | – | 3136 | – |
| IROF | – | $\Delta^-_{\mathrm{ratio}}$ | – | 3136 | – |
| INF | $\Delta_{\mathrm{minus}}$ | – | $\tau_{\mathrm{pearson}}$ | – | 100 |

Table 11: Parameters of ten faithfulness metrics for four explanation tasks on text modality.

| Metric | $\Delta$ | $\Delta^-$ | $\tau$ | $|\mathcal{I}_i|$ | $N$ |
|---|---|---|---|---|---|
| FC | $\Delta_{\mathrm{minus}}$ | – | $\tau_{\mathrm{pearson}}$ | 10 | 100 |
| FE | $\Delta_{\mathrm{minus}}$ | – | $\tau_{\mathrm{pearson}}$ | – | – |
| MC | $\Delta_{\mathrm{variance}}$ | – | $\tau_{\mathrm{spearman}}$ | – | – |
| RP | $\Delta_{\mathrm{minus}}$ | – | – | – | – |
| INS | – | $\Delta^-_{\mathrm{target}}$ | – | 10 | – |
| DEL | – | $\Delta^-_{\mathrm{target}}$ | – | 10 | – |
| NEG | – | $\Delta^-_{\mathrm{target}}$ | – | 10 | – |
| POS | – | $\Delta^-_{\mathrm{target}}$ | – | 10 | – |
| IROF | – | $\Delta^-_{\mathrm{ratio}}$ | – | 10 | – |
| INF | $\Delta_{\mathrm{minus}}$ | – | $\tau_{\mathrm{pearson}}$ | – | 100 |

Table 12: Parameters of ten faithfulness metrics for two explanation tasks on tabular modality.

| Metric | $\Delta$ | $\Delta^-$ | $\tau$ | $|\mathcal{I}_i|$ | $N$ |
|---|---|---|---|---|---|
| FC | $\Delta_{\mathrm{minus}}$ | – | $\tau_{\mathrm{pearson}}$ | 1 | 30 |
| FE | $\Delta_{\mathrm{minus}}$ | – | $\tau_{\mathrm{pearson}}$ | – | – |
| MC | $\Delta_{\mathrm{variance}}$ | – | $\tau_{\mathrm{spearman}}$ | – | – |
| RP | $\Delta_{\mathrm{minus}}$ | – | – | – | – |
| INS | – | $\Delta^-_{\mathrm{target}}$ | – | 1 | – |
| DEL | – | $\Delta^-_{\mathrm{target}}$ | – | 1 | – |
| NEG | – | $\Delta^-_{\mathrm{target}}$ | – | 1 | – |
| POS | – | $\Delta^-_{\mathrm{target}}$ | – | 1 | – |
| IROF | – | $\Delta^-_{\mathrm{ratio}}$ | – | 1 | – |
| INF | $\Delta_{\mathrm{minus}}$ | – | $\tau_{\mathrm{pearson}}$ | – | 30 |

As shown in Table 9, we use the same evaluation metric parameters for the three explanation tasks on ImageNet, with those for the three tasks on OCT given in Table 10. For the four explanation tasks in the text modality, we use the parameters in Table 11, while those for the two tasks in the tabular modality are listed in Table 12. We use the same perturbation scheme—baseline-value replacement—with identical baseline values across all experiments.

# E  GENERATING SUPERVISED EXPLANATION SIGNALS

Table 13: Similarity threshold and $p$-quantile used by **DeepFaith** for deduplicating and filtering supervised explanation signals.

| Threshold | OCT | | | ImageNet | | | AGNews | | IMDb | | NAP | WCD |
| | DeiT | EfficientNet | ResNet | DeiT | EfficientNet | ResNet | LSTM | Transformer | LSTM | Transformer | MLP | MLP |
|---|---|---|---|---|---|---|---|---|---|---|---|---|
| Similarity Threshold | 0.90 | 0.90 | 0.88 | 0.90 | 0.85 | 0.90 | 0.60 | 0.65 | 0.60 | 0.60 | 0.90 | 0.93 |
| P-Quantile | 0.13 | 0.13 | 0.14 | 0.13 | 0.14 | 0.13 | 0.20 | 0.22 | 0.21 | 0.19 | 0.15 | 0.15 |

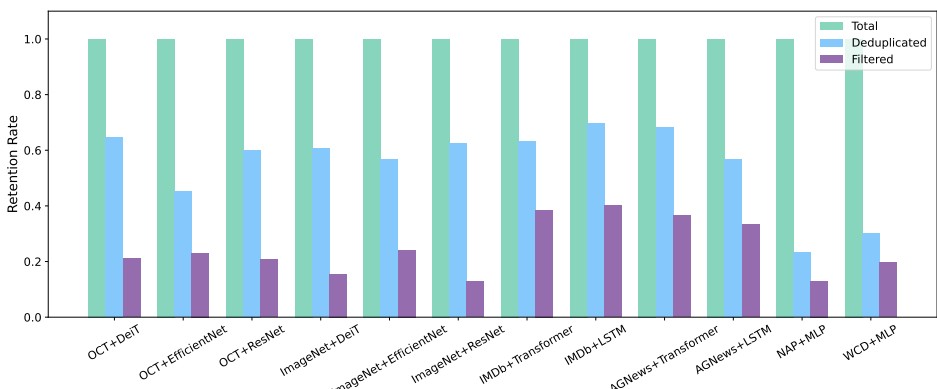

Figure 4: For the 12 explanation tasks we selected, **DeepFaith** performs deduplicating and filtering on the explanations generated by all baseline methods. *Total* denotes the total number of explanations, *Deduplicated* denotes the proportion remaining after deduplicating, and *Filtered* denotes the proportion remaining after further quality-based filtering on top of deduplicating.

Table 13 reports the similarity thresholds (for deduplicating) and $p$-quantiles (for filtering) used by **DeepFaith** when generating supervised explanation signals for 12 explanation tasks. Figure 4 illustrates, for each task, the proportion of explanations retained after deduplicating and filtering, relative to all generated explanations.

# F  EXPLAINER TRAINING CONFIGURATIONS

Table 14: The hyperparameters and training parameters of the explainer model (a multi-layer Transformer encoder) are set with varying complexities for different explanation tasks to ensure generalization. *nr runs* denotes the number of samples used for our local consistency loss.

| Config | OCT | | | ImageNet | | | AGNews | | IMDb | | NAP | WCD |
| | DeiT | EfficientNet | ResNet | DeiT | EfficientNet | ResNet | LSTM | Transformer | LSTM | Transformer | MLP | MLP |
|---|---|---|---|---|---|---|---|---|---|---|---|---|
| seq len | 196 | 196 | 196 | 196 | 196 | 196 | 300 | 128 | 500 | 200 | 8 | 7 |
| embed dim | 768 | 768 | 768 | 768 | 768 | 768 | 256 | 512 | 512 | 512 | 64 | 64 |
| n head | 12 | 8 | 8 | 12 | 8 | 8 | 8 | 8 | 8 | 8 | 8 | 8 |
| max len | 1000 | 1000 | 1000 | 1000 | 1000 | 1000 | 1000 | 1000 | 1000 | 1000 | 100 | 100 |
| ffn hidden | 1024 | 1024 | 1024 | 2048 | 1024 | 1024 | 1024 | 1024 | 1024 | 1024 | 128 | 128 |
| n layers | 8 | 6 | 6 | 12 | 6 | 6 | 8 | 8 | 8 | 8 | 4 | 4 |
| drop prob | 0.01 | 0.5 | 0.01 | 0.01 | 0.05 | 0.05 | 0.01 | 0.01 | 0.01 | 0.01 | 0.01 | 0.01 |
| epochs | 100 | 100 | 100 | 100 | 100 | 100 | 50 | 50 | 50 | 50 | 20 | 20 |
| learning rate | 1e-5 | 1e-5 | 1e-5 | 1e-5 | 1e-5 | 1e-5 | 1e-5 | 1e-5 | 1e-5 | 1e-5 | 1e-5 | 1e-5 |
| weight decay | 1e-5 | 1e-5 | 1e-5 | 1e-5 | 1e-5 | 1e-5 | 1e-5 | 1e-5 | 1e-5 | 1e-5 | 1e-5 | 1e-5 |

We set the hyperparameters in Algorithm 1 for all explanation tasks as follows: $\epsilon = 1e - 5$, $e = 5$, $C = 10$, and $k = 100$ for $\mathcal{L}_{\mathrm{LC}}$ in Equation (2).

## G FAITHFULNESS COMPARISON

In this section, we report the faithfulness comparison of **DeepFaith** and other baseline explanation methods across 12 explanation tasks, including the average over all test samples for each faithfulness evaluation metric and the average ranking across all metrics, including FC (Bhatt et al., 2020), FE (Alvarez Melis & Jaakkola, 2018), INF (Yeh et al., 2019), and MC for saliency explanations, as

Table 15: Faithfulness comparison of **DeepFaith** and baseline methods on OCT+DeiT task.

| Method | FC | FE | MC | RP | INS | DEL | NEG | POS | IROF | INF | Mean Rank |
|---|---|---|---|---|---|---|---|---|---|---|---|
| DeepFaith | 0.217 | 0.475 | 0.897 | 0.643 | 0.944 | 0.356 | 0.917 | 0.368 | 0.638 | 0.089 | 3.4 |
| Occlusion | 0.034 | 0.602 | 0.850 | 0.520 | 0.903 | 0.479 | 0.924 | 0.486 | 0.517 | 0.009 | 8.5 |
| LIME | -0.015 | 0.132 | 0.394 | 0.531 | 0.689 | 0.466 | 0.650 | 0.469 | 0.527 | 0.024 | 12.3 |
| Kernel SHAP | 0.018 | 0.649 | 0.928 | 0.683 | 0.888 | 0.314 | 0.853 | 0.300 | 0.678 | 0.038 | 4.2 |
| Saliency | 0.006 | 0.105 | 0.287 | 0.314 | 0.740 | 0.685 | 0.764 | 0.691 | 0.310 | 0.004 | 13.2 |
| Input x Gradient | -0.002 | 0.575 | 0.759 | 0.604 | 0.920 | 0.395 | 0.887 | 0.312 | 0.598 | 0.030 | 5.7 |
| Guided Backprop | 0.009 | 0.075 | 0.424 | 0.339 | 0.690 | 0.660 | 0.774 | 0.682 | 0.336 | -0.026 | 12.3 |
| Grad-CAM | 0.030 | 0.464 | 0.784 | 0.582 | 0.877 | 0.414 | 0.869 | 0.374 | 0.577 | -0.008 | 8.6 |
| Score-CAM | 0.479 | 0.663 | 0.154 | 0.597 | 0.769 | 0.399 | 0.709 | 0.390 | 0.591 | 0.079 | 7.0 |
| Grad-CAM++ | 0.047 | 0.571 | 0.822 | 0.599 | 0.921 | 0.397 | 0.874 | 0.358 | 0.594 | 0.058 | 5.0 |
| Integrated Grads | 0.272 | 0.788 | 0.438 | 0.547 | 0.858 | 0.449 | 0.664 | 0.429 | 0.542 | 0.114 | 7.8 |
| Expected Grads | 0.439 | 0.724 | 0.214 | 0.597 | 0.761 | 0.399 | 0.701 | 0.377 | 0.591 | 0.060 | 6.9 |
| DeepLIFT | 0.322 | 0.696 | 0.182 | 0.610 | 0.784 | 0.384 | 0.762 | 0.391 | 0.605 | 0.165 | 5.8 |
| DeepLIFT SHAP | 0.060 | 0.530 | 0.810 | 0.584 | 0.908 | 0.413 | 0.866 | 0.390 | 0.579 | 0.043 | 6.9 |
| LRP | -0.010 | 0.012 | 0.944 | 0.410 | 0.707 | 0.589 | 0.696 | 0.617 | 0.408 | 0.019 | 12.0 |

Table 16: Faithfulness comparison of **DeepFaith** and baseline methods on OCT+EfficientNet task.

| Method | FC | FE | MC | RP | INS | DEL | NEG | POS | IROF | INF | Mean Rank |
|---|---|---|---|---|---|---|---|---|---|---|---|
| DeepFaith | 0.060 | 0.784 | 0.959 | 0.759 | 0.572 | 0.240 | 0.339 | 0.161 | 0.747 | 0.029 | 2.9 |
| Occlusion | 0.106 | 0.501 | 0.525 | 0.639 | 0.845 | 0.360 | 0.825 | 0.312 | 0.633 | 0.022 | 6.5 |
| LIME | 0.098 | 0.463 | 0.624 | 0.636 | 0.841 | 0.364 | 0.867 | 0.391 | 0.631 | -0.001 | 8.1 |
| Kernel SHAP | -0.031 | 0.054 | -0.044 | 0.691 | 0.403 | 0.303 | 0.252 | 0.272 | 0.685 | -0.007 | 10.9 |
| Saliency | -0.007 | 0.021 | 0.106 | 0.564 | 0.528 | 0.434 | 0.392 | 0.289 | 0.559 | 0.021 | 11.0 |
| Input x Gradient | 0.011 | -0.181 | -0.618 | 0.452 | 0.531 | 0.551 | 0.397 | 0.479 | 0.448 | -0.008 | 12.3 |
| Guided Backprop | 0.107 | 0.542 | 0.710 | 0.634 | 0.853 | 0.365 | 0.915 | 0.382 | 0.629 | 0.007 | 6.5 |
| Grad-CAM | 0.041 | 0.498 | 0.248 | 0.731 | 0.353 | 0.270 | 0.287 | 0.196 | 0.72 | -0.033 | 8.2 |
| Score-CAM | 0.015 | 0.516 | 0.627 | 0.728 | 0.406 | 0.271 | 0.162 | 0.196 | 0.718 | -0.018 | 7.9 |
| Grad-CAM++ | 0.007 | 0.099 | 0.363 | 0.710 | 0.350 | 0.295 | 0.287 | 0.224 | 0.702 | -0.019 | 10.0 |
| Integrated Grads | 0.074 | 0.522 | 0.116 | 0.735 | 0.347 | 0.259 | 0.282 | 0.176 | 0.725 | -0.051 | 7.6 |
| Expected Grads | -0.004 | 0.390 | 0.056 | 0.737 | 0.379 | 0.262 | 0.245 | 0.175 | 0.726 | 0.001 | 8.0 |
| DeepLIFT | 0.092 | 0.471 | 0.599 | 0.638 | 0.847 | 0.362 | 0.858 | 0.373 | 0.632 | -0.008 | 7.8 |
| DeepLIFT SHAP | 0.050 | 0.495 | 0.228 | 0.751 | 0.351 | 0.260 | 0.303 | 0.187 | 0.740 | -0.026 | 7.3 |
| LRP | 0.065 | 0.726 | 0.981 | 0.727 | 0.727 | 0.270 | 0.490 | 0.171 | 0.717 | 0.001 | 4.5 |

Table 17: Faithfulness comparison of **DeepFaith** and baseline methods on OCT+ResNet task.

| Method | FC | FE | MC | RP | INS | DEL | NEG | POS | IROF | INF | Mean Rank |
|---|---|---|---|---|---|---|---|---|---|---|---|
| DeepFaith | 0.135 | 0.534 | 0.942 | 0.744 | 0.863 | 0.248 | 0.655 | 0.242 | 0.742 | 0.015 | 4.1 |
| Occlusion | 0.072 | 0.523 | 0.766 | 0.677 | 0.930 | 0.316 | 0.888 | 0.372 | 0.677 | 0.028 | 8.4 |
| LIME | 0.062 | 0.413 | 0.738 | 0.672 | 0.931 | 0.320 | 0.908 | 0.364 | 0.672 | -0.010 | 9.9 |
| Kernel SHAP | -0.029 | 0.318 | 0.900 | 0.651 | 0.734 | 0.343 | 0.490 | 0.409 | 0.645 | -0.029 | 12.1 |
| Saliency | 0.011 | 0.111 | 0.327 | 0.551 | 0.675 | 0.443 | 0.576 | 0.368 | 0.549 | -0.003 | 12.8 |
| Input x Gradient | 0.010 | 0.013 | 0.818 | 0.502 | 0.673 | 0.490 | 0.573 | 0.438 | 0.500 | 0.012 | 12.2 |
| Guided Backprop | 0.039 | 0.452 | 0.769 | 0.685 | 0.932 | 0.307 | 0.922 | 0.333 | 0.686 | 0.008 | 7.6 |
| Grad-CAM | 0.139 | 0.638 | 0.375 | 0.730 | 0.667 | 0.261 | 0.457 | 0.148 | 0.723 | 0.007 | 7.6 |
| Score-CAM | 0.108 | 0.697 | 0.810 | 0.717 | 0.939 | 0.275 | 0.659 | 0.236 | 0.710 | -0.001 | 6.0 |
| Grad-CAM++ | 0.100 | 0.619 | 0.256 | 0.728 | 0.898 | 0.266 | 0.544 | 0.184 | 0.720 | 0.015 | 7.4 |
| Integrated Grads | 0.155 | 0.730 | 0.358 | 0.741 | 0.692 | 0.250 | 0.509 | 0.136 | 0.733 | 0.024 | 4.8 |
| Expected Grads | 0.158 | 0.661 | 0.368 | 0.731 | 0.668 | 0.260 | 0.465 | 0.135 | 0.724 | -0.010 | 7.1 |
| DeepLIFT | 0.075 | 0.425 | 0.771 | 0.683 | 0.932 | 0.308 | 0.908 | 0.357 | 0.683 | -0.002 | 8.1 |
| DeepLIFT SHAP | 0.103 | 0.696 | 0.351 | 0.737 | 0.808 | 0.253 | 0.582 | 0.152 | 0.732 | 0.015 | 5.6 |
| LRP | 0.109 | 0.684 | 0.852 | 0.724 | 0.932 | 0.268 | 0.727 | 0.242 | 0.718 | 0.011 | 5.4 |

well as DEL and INS (Petsiuk & Saenko, 2018), NEG and POS (Barkan et al., 2023), RP, and IROF (Rieger & Hansen, 2020) for permutation explanations.

Table 18: Faithfulness comparison of **DeepFaith** and baseline methods on ImageNet+DeiT task.

| Method | FC | FE | MC | RP | INS | DEL | NEG | POS | IROF | INF | Mean Rank |
|---|---|---|---|---|---|---|---|---|---|---|---|
| DeepFaith | 0.060 | 0.784 | 0.959 | 0.759 | 0.572 | 0.240 | 0.339 | 0.161 | 0.747 | 0.029 | 2.9 |
| Occlusion | 0.106 | 0.501 | 0.525 | 0.639 | 0.845 | 0.360 | 0.825 | 0.312 | 0.633 | 0.022 | 6.5 |
| LIME | 0.098 | 0.463 | 0.624 | 0.636 | 0.841 | 0.364 | 0.867 | 0.391 | 0.631 | -0.001 | 8.1 |
| Kernel SHAP | -0.031 | 0.054 | -0.044 | 0.691 | 0.403 | 0.303 | 0.252 | 0.272 | 0.685 | -0.007 | 10.9 |
| Saliency | -0.007 | 0.021 | 0.106 | 0.564 | 0.528 | 0.434 | 0.392 | 0.289 | 0.559 | 0.021 | 11.0 |
| Input x Gradient | 0.011 | -0.181 | -0.618 | 0.452 | 0.531 | 0.551 | 0.397 | 0.479 | 0.448 | -0.008 | 12.3 |
| Guided Backprop | 0.107 | 0.542 | 0.710 | 0.634 | 0.853 | 0.365 | 0.915 | 0.382 | 0.629 | 0.007 | 6.5 |
| Grad-CAM | 0.041 | 0.498 | 0.248 | 0.731 | 0.353 | 0.270 | 0.287 | 0.196 | 0.720 | -0.033 | 8.2 |
| Score-CAM | 0.015 | 0.516 | 0.627 | 0.728 | 0.406 | 0.271 | 0.162 | 0.196 | 0.718 | -0.018 | 7.9 |
| Grad-CAM++ | 0.007 | 0.099 | 0.363 | 0.710 | 0.350 | 0.295 | 0.287 | 0.224 | 0.702 | -0.019 | 10.0 |
| Integrated Grads | 0.074 | 0.522 | 0.116 | 0.735 | 0.347 | 0.259 | 0.282 | 0.176 | 0.725 | -0.051 | 7.6 |
| Expected Grads | -0.004 | 0.390 | 0.056 | 0.737 | 0.379 | 0.262 | 0.245 | 0.175 | 0.726 | 0.001 | 8.0 |
| DeepLIFT | 0.092 | 0.471 | 0.599 | 0.638 | 0.847 | 0.362 | 0.858 | 0.373 | 0.632 | -0.008 | 7.8 |
| DeepLIFT SHAP | 0.050 | 0.495 | 0.228 | 0.751 | 0.351 | 0.260 | 0.303 | 0.187 | 0.740 | -0.026 | 7.3 |
| LRP | 0.065 | 0.726 | 0.981 | 0.727 | 0.727 | 0.270 | 0.490 | 0.171 | 0.717 | 0.001 | 4.5 |

Table 19: Faithfulness comparison of **DeepFaith** and baseline methods on ImageNet+EfficientNet task.

| Method | FC | FE | MC | RP | INS | DEL | NEG | POS | IROF | INF | Mean Rank |
|---|---|---|---|---|---|---|---|---|---|---|---|
| DeepFaith | 0.021 | 0.217 | 0.835 | 0.591 | 0.525 | 0.174 | 0.405 | 0.156 | 0.749 | 0.004 | 4.4 |
| Occlusion | 0.007 | -0.017 | -0.525 | 0.333 | 0.499 | 0.360 | 0.445 | 0.415 | 0.476 | 0.028 | 9.6 |
| LIME | -0.008 | 0.112 | 0.921 | 0.461 | 0.593 | 0.231 | 0.542 | 0.306 | 0.657 | 0.012 | 6.6 |
| Kernel SHAP | 0.003 | 0.102 | 0.928 | 0.535 | 0.554 | 0.158 | 0.400 | 0.245 | 0.772 | 0.019 | 5.9 |
| Saliency | -0.016 | -0.022 | -0.161 | 0.410 | 0.423 | 0.283 | 0.350 | 0.338 | 0.559 | 0.011 | 11.1 |
| Input × Gradient | -0.053 | -0.073 | -0.802 | 0.344 | 0.423 | 0.349 | 0.380 | 0.393 | 0.446 | -0.015 | 12.9 |
| Guided Backprop | -0.002 | -0.012 | 0.279 | 0.321 | 0.535 | 0.372 | 0.496 | 0.430 | 0.469 | 0.003 | 10.3 |
| Grad-CAM | -0.009 | 0.122 | 0.282 | 0.595 | 0.404 | 0.099 | 0.255 | 0.182 | 0.852 | 0.000 | 6.6 |
| Score-CAM | -0.005 | 0.103 | 0.389 | 0.559 | 0.466 | 0.135 | 0.291 | 0.233 | 0.808 | -0.005 | 7.2 |
| Grad-CAM++ | 0.004 | 0.087 | 0.234 | 0.538 | 0.457 | 0.156 | 0.295 | 0.241 | 0.782 | 0.014 | 7.8 |
| Integrated Grads | -0.016 | 0.192 | 0.287 | 0.590 | 0.397 | 0.103 | 0.274 | 0.186 | 0.848 | -0.030 | 7.0 |
| Expected Grads | -0.037 | 0.137 | 0.263 | 0.492 | 0.460 | 0.201 | 0.334 | 0.286 | 0.717 | -0.014 | 9.5 |
| DeepLIFT | -0.050 | 0.164 | 0.890 | 0.450 | 0.575 | 0.243 | 0.537 | 0.304 | 0.639 | 0.027 | 6.9 |
| DeepLIFT SHAP | -0.066 | 0.112 | 0.287 | 0.588 | 0.397 | 0.106 | 0.248 | 0.195 | 0.843 | -0.035 | 8.7 |
| LRP | 0.005 | 0.139 | 0.964 | 0.555 | 0.506 | 0.138 | 0.363 | 0.231 | 0.794 | 0.010 | 5.0 |

Table 20: Faithfulness comparison of **DeepFaith** and baseline methods on ImageNet+ResNet task.

| Method | FC | FE | MC | RP | INS | DEL | NEG | POS | IROF | INF | Mean Rank |
|---|---|---|---|---|---|---|---|---|---|---|---|
| DeepFaith | 0.031 | 0.254 | 0.938 | 0.677 | 0.577 | 0.106 | 0.471 | 0.182 | 0.871 | 0.019 | 3.3 |
| Occlusion | 0.023 | 0.053 | 0.125 | 0.457 | 0.507 | 0.285 | 0.393 | 0.366 | 0.633 | 0.015 | 10.9 |
| LIME | -0.017 | 0.202 | 0.942 | 0.573 | 0.562 | 0.169 | 0.498 | 0.280 | 0.776 | -0.004 | 8.5 |
| Kernel SHAP | -0.015 | 0.224 | 0.761 | 0.579 | 0.435 | 0.164 | 0.344 | 0.255 | 0.775 | 0.021 | 8.9 |
| Saliency | 0.006 | 0.030 | 0.005 | 0.627 | 0.365 | 0.115 | 0.268 | 0.220 | 0.840 | 0.003 | 10.6 |
| Input × Gradient | -0.041 | -0.012 | -0.337 | 0.640 | 0.373 | 0.101 | 0.257 | 0.204 | 0.856 | -0.041 | 10.7 |
| Guided Backprop | -0.012 | 0.066 | 0.414 | 0.492 | 0.513 | 0.250 | 0.407 | 0.340 | 0.668 | 0.028 | 10.4 |
| Grad-CAM | 0.042 | 0.199 | 0.569 | 0.655 | 0.367 | 0.088 | 0.271 | 0.199 | 0.870 | -0.026 | 7.0 |
| Score-CAM | 0.041 | 0.268 | 0.697 | 0.628 | 0.380 | 0.116 | 0.282 | 0.230 | 0.846 | 0.009 | 7.1 |
| Grad-CAM++ | 0.057 | 0.104 | 0.626 | 0.627 | 0.449 | 0.116 | 0.284 | 0.212 | 0.835 | -0.017 | 8.3 |
| Integrated Grads | 0.023 | 0.244 | 0.570 | 0.666 | 0.368 | 0.078 | 0.256 | 0.162 | 0.882 | 0.035 | 5.4 |
| Expected Grads | 0.012 | 0.253 | 0.560 | 0.654 | 0.352 | 0.090 | 0.259 | 0.180 | 0.872 | 0.045 | 6.4 |
| DeepLIFT | -0.001 | 0.181 | 0.910 | 0.510 | 0.526 | 0.232 | 0.456 | 0.327 | 0.693 | 0.040 | 8.4 |
| DeepLIFT SHAP | 0.019 | 0.178 | 0.604 | 0.629 | 0.362 | 0.114 | 0.267 | 0.207 | 0.839 | -0.003 | 9.3 |
| LRP | 0.036 | 0.345 | 0.946 | 0.648 | 0.450 | 0.096 | 0.374 | 0.186 | 0.868 | 0.004 | 4.5 |

Tables 15, 16, and 17 present the experimental results for the three explanation tasks on the OCT dataset, while Tables 18, 19, and 20 report those on the ImageNet dataset.

In all cases, **DeepFaith** achieves the optimal average ranking, indicating that its generated explanations exhibit strong generalization from a faithfulness perspective across test set samples.

Table 21: Faithfulness comparison of **DeepFaith** and baseline methods on IMDb+LSTM task.

| Method | FC | FE | MC | RP | INS | DEL | NEG | POS | IROF | INF | Mean Rank |
|---|---|---|---|---|---|---|---|---|---|---|---|
| DeepFaith | 0.172 | 0.486 | 0.360 | 0.812 | 0.872 | 0.151 | 0.869 | 0.182 | 0.813 | 0.038 | 2.3 |
| Integrated Grads | 0.201 | 0.492 | 0.048 | 0.798 | 0.832 | 0.167 | 0.827 | 0.195 | 0.798 | 0.324 | 3.3 |
| Gradient SHAP | 0.081 | 0.277 | 0.054 | 0.796 | 0.832 | 0.168 | 0.828 | 0.207 | 0.797 | 0.278 | 4.4 |
| DeepLIFT | 0.031 | 0.088 | 0.154 | 0.517 | 0.840 | 0.442 | 0.846 | 0.501 | 0.521 | 0.081 | 6.1 |
| Saliency | 0.093 | 0.256 | 0.050 | 0.706 | 0.835 | 0.255 | 0.840 | 0.328 | 0.709 | 0.161 | 5.2 |
| Occlusion | 0.087 | 0.311 | 0.049 | 0.794 | 0.832 | 0.169 | 0.829 | 0.209 | 0.795 | 0.313 | 4.6 |
| Feature Ablation | 0.042 | 0.082 | 0.018 | 0.765 | 0.832 | 0.199 | 0.829 | 0.233 | 0.766 | 0.082 | 6.4 |
| LIME | 0.020 | 0.056 | -0.844 | 0.408 | 0.890 | 0.551 | 0.864 | 0.530 | 0.410 | 0.036 | 7.7 |
| Kernel SHAP | 0.094 | 0.250 | 0.994 | 0.452 | 0.913 | 0.508 | 0.900 | 0.411 | 0.453 | 0.060 | 5.0 |

Table 22: Faithfulness comparison of **DeepFaith** and baseline methods on IMDb+Transformer task.

| Method | FC | FE | MC | RP | INS | DEL | NEG | POS | IROF | INF | Mean Rank |
|---|---|---|---|---|---|---|---|---|---|---|---|
| DeepFaith | 0.162 | 0.495 | 0.203 | 0.759 | 0.806 | 0.189 | 0.799 | 0.205 | 0.742 | 0.047 | 2.1 |
| Integrated Grads | 0.061 | 0.208 | 0.037 | 0.634 | 0.766 | 0.307 | 0.762 | 0.355 | 0.625 | 0.128 | 5.6 |
| Gradient SHAP | 0.117 | 0.347 | 0.066 | 0.739 | 0.760 | 0.210 | 0.747 | 0.219 | 0.722 | 0.301 | 4.0 |
| DeepLIFT | 0.019 | 0.062 | 0.073 | 0.338 | 0.800 | 0.599 | 0.774 | 0.577 | 0.332 | 0.058 | 6.4 |
| Saliency | 0.050 | 0.167 | 0.060 | 0.577 | 0.770 | 0.363 | 0.768 | 0.406 | 0.570 | 0.107 | 5.9 |
| Occlusion | 0.119 | 0.352 | 0.071 | 0.740 | 0.760 | 0.210 | 0.747 | 0.218 | 0.723 | 0.308 | 3.6 |
| Feature Ablation | 0.110 | 0.324 | 0.060 | 0.729 | 0.760 | 0.220 | 0.748 | 0.236 | 0.712 | 0.258 | 5.1 |
| LIME | 0.021 | 0.064 | 0.075 | 0.299 | 0.817 | 0.638 | 0.767 | 0.592 | 0.294 | 0.040 | 6.8 |
| Kernel SHAP | 0.102 | 0.178 | 0.979 | 0.283 | 0.819 | 0.655 | 0.803 | 0.566 | 0.276 | 0.052 | 5.5 |

Table 23: Faithfulness comparison of **DeepFaith** and baseline methods on AGNews+LSTM task.

| Method | FC | FE | MC | RP | INS | DEL | NEG | POS | IROF | INF | Mean Rank |
|---|---|---|---|---|---|---|---|---|---|---|---|
| DeepFaith | 0.363 | 0.597 | 0.629 | 0.648 | 0.919 | 0.197 | 0.906 | 0.256 | 0.650 | 0.275 | 2.9 |
| Integrated Grads | 0.492 | 0.597 | 0.052 | 0.571 | 0.838 | 0.375 | 0.825 | 0.383 | 0.573 | 0.313 | 4.9 |
| Gradient SHAP | 0.302 | 0.623 | 0.019 | 0.787 | 0.879 | 0.161 | 0.868 | 0.176 | 0.786 | 0.434 | 2.9 |
| DeepLIFT | 0.032 | 0.095 | 0.077 | 0.373 | 0.846 | 0.568 | 0.825 | 0.564 | 0.377 | 0.074 | 7.9 |
| Saliency | 0.497 | 0.597 | 0.052 | 0.571 | 0.838 | 0.375 | 0.825 | 0.383 | 0.573 | 0.316 | 4.7 |
| Occlusion | 0.313 | 0.671 | -0.001 | 0.727 | 0.882 | 0.122 | 0.871 | 0.214 | 0.726 | 0.362 | 2.9 |
| Feature Ablation | 0.226 | 0.311 | 0.028 | 0.539 | 0.844 | 0.407 | 0.831 | 0.416 | 0.541 | 0.217 | 6.6 |
| LIME | 0.061 | 0.202 | 0.269 | 0.657 | 0.908 | 0.284 | 0.890 | 0.296 | 0.663 | 0.067 | 4.6 |
| Kernel SHAP | 0.112 | 0.245 | 0.952 | 0.445 | 0.853 | 0.496 | 0.838 | 0.498 | 0.449 | 0.111 | 6.4 |

Table 24: Faithfulness comparison of **DeepFaith** and baseline methods on AGNews+Transformer task.

| Method | FC | FE | MC | RP | INS | DEL | NEG | POS | IROF | INF | Mean Rank |
|---|---|---|---|---|---|---|---|---|---|---|---|
| DeepFaith | 0.111 | 0.318 | 0.464 | 0.663 | 0.901 | 0.277 | 0.864 | 0.194 | 0.651 | 0.082 | 2.7 |
| Integrated Grads | 0.119 | 0.290 | 0.015 | 0.400 | 0.808 | 0.545 | 0.776 | 0.529 | 0.370 | 0.154 | 5.9 |
| Gradient SHAP | 0.091 | 0.402 | 0.065 | 0.627 | 0.811 | 0.321 | 0.780 | 0.319 | 0.601 | 0.249 | 4.2 |
| DeepLIFT | 0.036 | 0.095 | 0.222 | 0.464 | 0.813 | 0.474 | 0.785 | 0.477 | 0.444 | 0.110 | 5.9 |
| Saliency | 0.121 | 0.290 | 0.015 | 0.400 | 0.808 | 0.545 | 0.776 | 0.529 | 0.370 | 0.154 | 5.8 |
| Occlusion | 0.099 | 0.500 | 0.028 | 0.787 | 0.811 | 0.163 | 0.782 | 0.163 | 0.755 | 0.290 | 2.7 |
| Feature Ablation | 0.027 | 0.204 | 0.019 | 0.391 | 0.775 | 0.554 | 0.739 | 0.557 | 0.363 | 0.099 | 8.5 |
| LIME | 0.081 | 0.231 | -0.038 | 0.655 | 0.816 | 0.285 | 0.789 | 0.282 | 0.633 | 0.175 | 4.5 |
| Kernel SHAP | 0.102 | 0.236 | 0.993 | 0.609 | 0.897 | 0.328 | 0.858 | 0.271 | 0.599 | 0.123 | 3.9 |

Tables 21 and 22 present the experimental results for the two explanation tasks on the IMDb dataset, while Tables 23 and 24 report those on the AGNews dataset. **DeepFaith** outperforms all base-

line explanation methods on the two IMDb tasks and matches the overall faithfulness of the best-performing baseline on AGNews. These results demonstrate that **DeepFaith** also provides highly faithful explanations for data in the text modality.

Table 25: Faithfulness comparison of **DeepFaith** and baseline methods on NAP+MLP task.

| Method | FC | FE | MC | RP | INS | DEL | NEG | POS | IROF | INF | Mean Rank |
|---|---|---|---|---|---|---|---|---|---|---|---|
| DeepFaith | 0.788 | 0.763 | 0.952 | 0.957 | 0.844 | 0.031 | 0.770 | 0.031 | 0.844 | 0.238 | 1.8 |
| Integrated Grads | -0.173 | -0.186 | 0.077 | -0.052 | 0.821 | 0.875 | 0.823 | 0.875 | -0.064 | -0.154 | 2.8 |
| Gradient SHAP | -0.178 | -0.189 | 0.043 | -0.052 | 0.817 | 0.875 | 0.819 | 0.875 | -0.064 | -0.153 | 4.7 |
| DeepLift | -0.175 | -0.187 | -0.033 | -0.052 | 0.814 | 0.875 | 0.817 | 0.875 | -0.064 | -0.146 | 4.4 |
| Saliency | -0.173 | -0.186 | 0.077 | -0.052 | 0.821 | 0.875 | 0.823 | 0.875 | -0.064 | -0.154 | 2.8 |
| Occlusion | -0.171 | -0.188 | 0.132 | -0.052 | 0.816 | 0.875 | 0.819 | 0.875 | -0.064 | -0.151 | 3.3 |
| Feature Ablation | -0.173 | -0.188 | 0.098 | -0.052 | 0.816 | 0.875 | 0.819 | 0.875 | -0.064 | -0.151 | 3.5 |
| LIME | -0.175 | -0.189 | 0.052 | -0.052 | 0.814 | 0.875 | 0.817 | 0.875 | -0.064 | -0.148 | 4.7 |
| Kernel SHAP | -0.180 | -0.187 | 0.531 | -0.052 | 0.820 | 0.875 | 0.822 | 0.875 | -0.064 | -0.158 | 3.9 |

Table 26: Faithfulness comparison of **DeepFaith** and baseline methods on WCD+MLP task.

| Method | FC | FE | MC | RP | INS | DEL | NEG | POS | IROF | INF | Mean Rank |
|---|---|---|---|---|---|---|---|---|---|---|---|
| DeepFaith | 0.961 | 0.961 | 0.679 | 0.364 | 0.723 | 0.551 | 0.575 | 0.455 | 0.306 | 0.929 | 1.8 |
| Integrated Grads | 0.142 | 0.132 | 0.534 | 0.074 | 0.536 | 0.492 | 0.526 | 0.477 | 0.066 | 0.197 | 5.2 |
| Gradient SHAP | 0.169 | 0.130 | 0.523 | 0.070 | 0.534 | 0.496 | 0.522 | 0.488 | 0.063 | 0.202 | 7.3 |
| DeepLIFT | 0.184 | 0.152 | 0.666 | 0.089 | 0.548 | 0.478 | 0.539 | 0.467 | 0.081 | 0.218 | 2.3 |
| Saliency | 0.160 | 0.132 | 0.534 | 0.074 | 0.536 | 0.492 | 0.526 | 0.477 | 0.066 | 0.200 | 4.9 |
| Occlusion | 0.168 | 0.136 | 0.672 | 0.072 | 0.536 | 0.495 | 0.524 | 0.486 | 0.064 | 0.198 | 5.9 |
| Feature Ablation | 0.175 | 0.141 | 0.654 | 0.073 | 0.538 | 0.493 | 0.526 | 0.484 | 0.066 | 0.211 | 4.5 |
| LIME | 0.186 | 0.159 | 0.551 | 0.089 | 0.547 | 0.479 | 0.537 | 0.468 | 0.080 | 0.220 | 2.7 |
| Kernel SHAP | 0.071 | 0.049 | -0.056 | 0.048 | 0.515 | 0.516 | 0.500 | 0.512 | 0.042 | 0.181 | 8.9 |

Tables 25 and 26 present the experimental results of explaining MLPs on the UCI dataset. Many baseline explanation methods score poorly on correlation-based faithfulness metrics such as FC and FE, as their perturbation procedures are overly complex relative to the simplicity of the explained model. In contrast,**DeepFaith** directly improves correlation-based faithfulness by optimizing the local correlation loss, achieving comprehensive superiority over the baseline explanation methods.

# H VISUALIZATIONS OF EXPLANATIONS

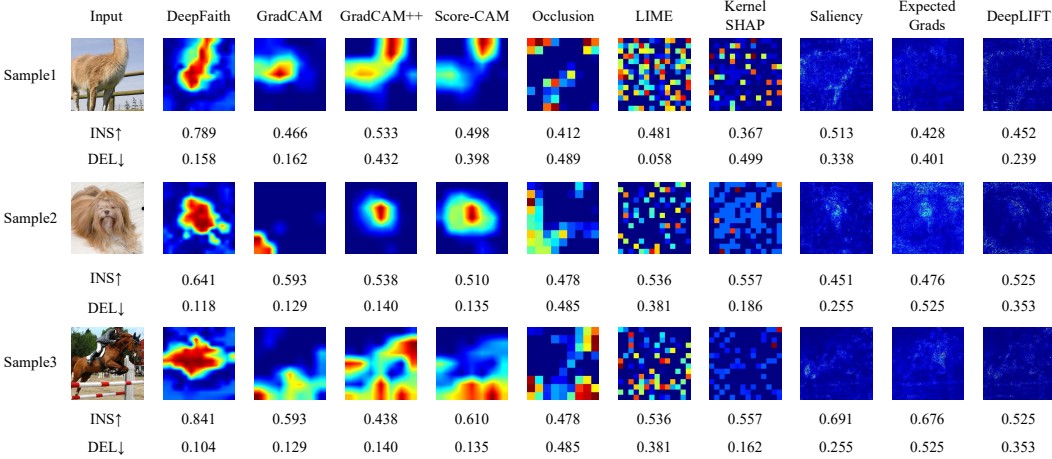

Figure 5: Saliency explanations generated by **DeepFaith** for DeiT on the ImageNet dataset, along with those produced by baseline methods for the same inputs.

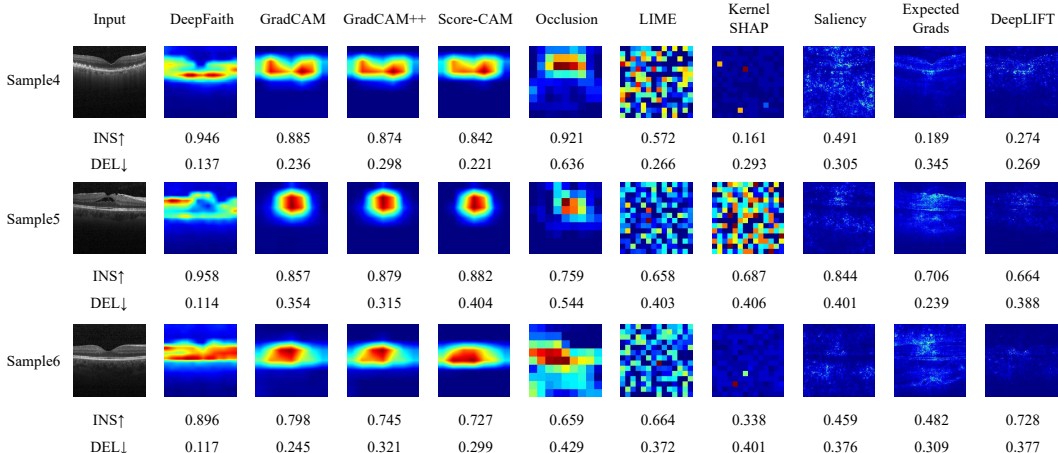

Figure 6: Saliency explanations generated by **DeepFaith** for EfficientNet on the OCT dataset, along with those produced by baseline methods for the same inputs.

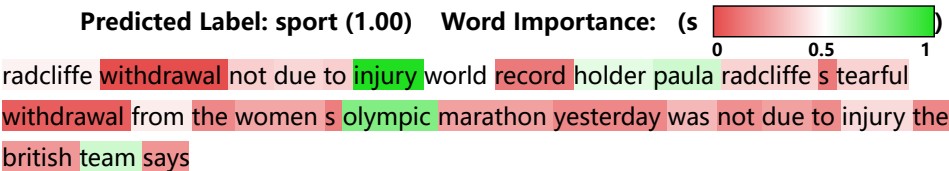

Figure 7: Attribution by **DeepFaith** for an LSTM predicting *sport* news on the AGNews dataset.

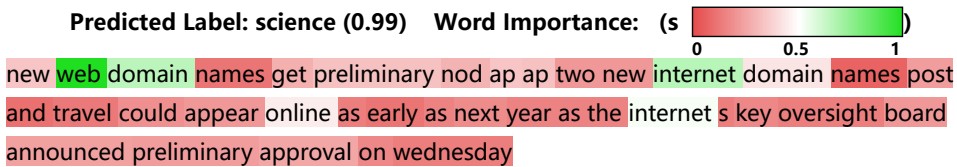

Figure 8: Attribution by **DeepFaith** for an LSTM predicting *science* news on the AGNews dataset.

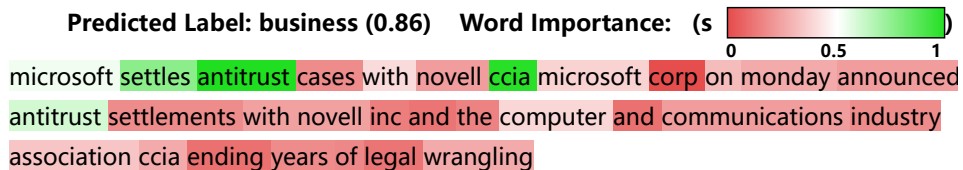

Figure 9: Attribution by **DeepFaith** for an LSTM predicting *business* news on the AGNews dataset.

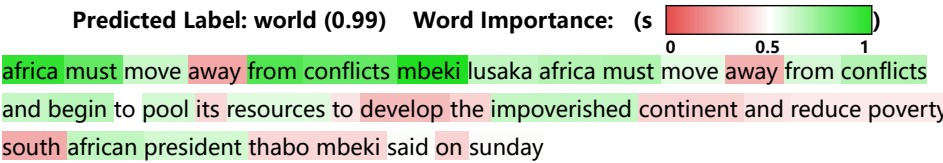

Figure 10: Attribution by **DeepFaith** for an LSTM predicting *world* news on the AGNews dataset.

**Predicted Label: positive (0.99)   Word Importance: (s**

0   0.5   1

whenever ida lupino appeared or directed a film in the s s and s you were guaranteed great entertainment even if the picture was black and white ida was able to capture audiences and keep them spellbound until the very end of her pictures in this film as mrs helen gordon high sierra along with robert ryan howard wilton golden gloves she keeps you guessing just how the relationship is going to turn out and just how poor mrs gordon will be able to have a normal and happy marriage with love and real affection if you liked ida lupino who could play the roles as a criminal in a woman s prison and prison warden who was hated this is the film for you to enjoy i truly believe that ida lupino was not given the true credit she deserved for her great talents in the movie industry

Figure 11: Attribution by **DeepFaith** for a Transformer predicting a review as the *positive* category on the IMDb dataset.

**Predicted Label: negative (0.99)   Word Importance: (s**

0   0.5   1

when you make a film with a killer kids premise there are two effective ways to approach it you can either make it as realistic as possible creating believable characters and situations or you can make it as fun as possible by playing it for laughs something which the makers of silent night deadly night did for example on an equally controversial subject a killer santa the people who made bloody birthday however do neither of those things they simply rely on the shock value of the image of a kid with a gun or a knife or a noose or an arrow in his her hand the result is both offensive and stupid the whole film looks like a bad idea that was rushed through production and then kept from release for several years it s redeemed a tiny bit by good performances from the kids but it s very sloppily made

Figure 12: Attribution by **DeepFaith** for a Transformer predicting a review as the *negative* category on the IMDb dataset.

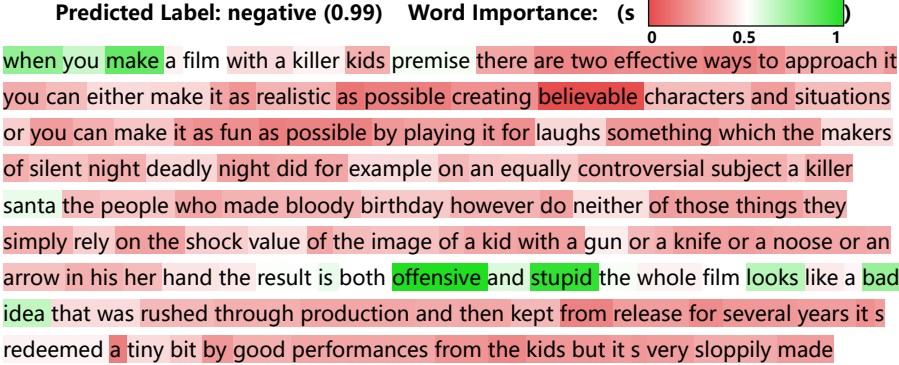

Figure 13: **DeepFaith** attributing four feature vectors predicted as different classes on the NAP (left) and WCD (right) datasets, respectively.

Figures 5 and 6 visualize the saliency maps produced by **DeepFaith** and baseline explanation methods on two image-modality datasets, together with their faithfulness scores INS and DEL. **Deep-Faith** not only achieves superior faithfulness metrics but also provides explanations that are more aligned with human interpretability.

For example, on the OCT dataset shown in Figure 6, **DeepFaith** explanations fully cover the relevant feature regions while highlighting finer-grained critical features. For samples 4 and 5—each corresponding to different ocular diseases—the highlighted areas produced by **DeepFaith** concentrate precisely on the abnormal lesion regions. For sample 6, which is labeled as healthy, **DeepFaith** highlights are evenly distributed across the feature region. The model decision rationale revealed by the explainer aligns closely with human understanding.

# I  RUNTIME COMPARISONS

Table 27: Average runtime (in ms) of **DeepFaith** and baseline methods for explaining a single sample in the image modality.

| Method | OCT | | | ImageNet | | |
|---|---|---|---|---|---|---|
| | DeiT | EfficientNet | ResNet | DeiT | EfficientNet | ResNet |
| DeepFaith | **0.609** | **2.406** | **2.103** | **3.103** | **1.207** | **2.403** |
| Integrated Grads | 78.425 | 48.764 | 103.721 | 95.132 | 52.537 | 110.638 |
| DeepLIFT | 3.618 | 14.649 | 15.003 | 14.918 | 14.417 | 15.428 |
| Saliency | 3.874 | 6.526 | 8.548 | 11.264 | 6.543 | 9.254 |
| Occlusion | 96.147 | 67.728 | 170.348 | 115.435 | 72.366 | 185.721 |
| LIME | 145.517 | 135.273 | 93.352 | 121.143 | 100.769 | 137.326 |
| Kernel SHAP | 69.891 | 65.862 | 63.114 | 68.946 | 71.627 | 69.642 |
| Input × Gradient | 3.747 | 2.584 | 3.134 | 3.839 | 1.964 | 3.378 |
| Guided Backprop | 3.438 | 2.561 | 3.372 | 4.849 | 1.978 | 3.673 |
| Grad-CAM | 4.139 | 9.163 | 9.617 | 13.756 | 8.739 | 10.539 |
| Score-CAM | 553.973 | 1037.456 | 3628.819 | 3609.633 | 1266.827 | 4203.115 |
| Grad-CAM++ | 4.389 | 3.957 | 3.769 | 6.048 | 3.652 | 4.548 |
| Expected Grads | 57.642 | 65.432 | 122.261 | 124.935 | 65.934 | 127.967 |
| DeepLIFT SHAP | 14.293 | 29.317 | 86.179 | 57.848 | 34.158 | 98.453 |
| LRP | 7.519 | 6.867 | 8.539 | 17.548 | 7.716 | 8.134 |

Table 28: Average runtime (in ms) of **DeepFaith** and baseline methods for explaining a single sample in text and tabular modality.

| Method | IMDb | | AGNews | | NAP | WCD |
|---|---|---|---|---|---|---|
| | LSTM | Transformer | LSTM | Transformer | MLP | MLP |
| DeepFaith | 1.217 | 0.563 | **1.137** | **0.433** | **0.117** | 0.173 |
| Integrated Grads | 3.125 | 21.926 | 53.941 | 58.473 | 2.839 | 0.219 |
| Gradient SHAP | **0.781** | 1.302 | 2.938 | 3.983 | 0.331 | **0.135** |
| DeepLIFT | 2.446 | 0.539 | 3.101 | 0.849 | 0.272 | 0.193 |
| Saliency | 1.278 | **0.238** | 5.894 | 0.682 | 0.254 | 0.137 |
| Occlusion | 10.872 | 0.587 | 61.725 | 25.734 | 0.563 | 0.293 |
| Feature Ablation | 21.921 | 0.253 | 60.987 | 25.768 | 0.337 | 0.293 |
| LIME | 59.067 | 15.954 | 79.311 | 112.438 | 16.125 | 22.443 |
| Kernel SHAP | 152.933 | 20.539 | 79.645 | 106.965 | 37.575 | 49.441 |

**DeepFaith** differs from conventional post-hoc attribution methods by incurring a one-time cost for generating supervised explanation signals and training its explainer. Once trained, however, it delivers explanations with low latency, making it well-suited for time-critical applications.

Across image, text, and tabular modalities, **DeepFaith** consistently produces explanations faster than sampling-based methods such as LIME, Kernel SHAP, and Occlusion, which require repeated perturbations and evaluations. It also outperforms most gradient-based approaches, including Integrated Grads, Grad-CAM, and Grad-CAM++, by avoiding repeated backpropagation through the explained model. This advantage becomes more pronounced for complex architectures, where the runtime of many baselines scales with model size, while **DeepFaith** remains largely unaffected due to its decoupling from the architecture of the target model.

The results demonstrate that **DeepFaith**'s inference speed is not only competitive but often superior across different backbones and modalities. Moreover, its consistent efficiency ensures that the same explainer design can be deployed in varied domains without sacrificing latency, enabling real-time interpretability in settings where both speed and explanation quality are critical.

## J   SENSITIVITY STUDY OF HYPERPARAMETRES

Table 29: Sensitivity study of **DeepFaith**. The baseline (first row) uses the original recommended hyperparameters $(\delta, p, e, C, k)$. $\uparrow/\downarrow$ after each hyperparameter indicates increasing/decreasing that hyperparameter while keeping others fixed.

| Method | FC↑ | FE↑ | MC↑ | RP↑ | INS↑ | DEL↓ | NEG↑ | POS↓ | IROF↑ | INF↑ | $\|\Delta F\|$ |
|---|---|---|---|---|---|---|---|---|---|---|---|
| DeepFaith$_{\delta,p,e,C,k}$ | 0.135 | 0.534 | 0.942 | 0.744 | 0.863 | 0.248 | 0.655 | 0.242 | 0.742 | 0.015 | 0 |
| DeepFaith$_{\delta\uparrow,p,e,C,k}$ | 0.140 | 0.497 | 0.919 | 0.706 | 0.783 | 0.266 | 0.628 | 0.266 | 0.717 | 0.016 | 3.513% |
| DeepFaith$_{\delta\downarrow,p,e,C,k}$ | 0.123 | 0.563 | 0.870 | 0.753 | 0.940 | 0.225 | 0.698 | 0.223 | 0.774 | 0.014 | 1.246% |
| DeepFaith$_{\delta,p\uparrow,e,C,k}$ | 0.147 | 0.511 | 0.954 | 0.679 | 0.863 | 0.272 | 0.619 | 0.258 | 0.693 | 0.016 | 2.083% |
| DeepFaith$_{\delta,p\downarrow,e,C,k}$ | 0.122 | 0.575 | 0.891 | 0.817 | 0.797 | 0.235 | 0.705 | 0.218 | 0.799 | 0.014 | 1.013% |
| DeepFaith$_{\delta,p,e\uparrow,C,k}$ | 0.118 | 0.588 | 0.924 | 0.709 | 0.884 | 0.276 | 0.672 | 0.270 | 0.664 | 0.036 | 0.410% |
| DeepFaith$_{\delta,p,e\downarrow,C,k}$ | 0.135 | 0.552 | 0.913 | 0.744 | 0.864 | 0.247 | 0.654 | 0.241 | 0.743 | 0.016 | 0.214% |
| DeepFaith$_{\delta,p,e,C\uparrow,k}$ | 0.136 | 0.557 | 0.905 | 0.743 | 0.863 | 0.248 | 0.654 | 0.243 | 0.740 | 0.016 | 0.292% |
| DeepFaith$_{\delta,p,e,C\downarrow,k}$ | 0.117 | 0.589 | 0.930 | 0.710 | 0.885 | 0.276 | 0.673 | 0.269 | 0.665 | 0.037 | 0.605% |
| DeepFaith$_{\delta,p,e,C,k\downarrow}$ | 0.143 | 0.522 | 0.913 | 0.727 | 0.847 | 0.245 | 0.626 | 0.236 | 0.721 | 0.005 | 2.636% |
| DeepFaith$_{\delta,p,e,C,k\uparrow}$ | 0.260 | 0.674 | 0.929 | 0.720 | 0.910 | 0.270 | 0.697 | 0.264 | 0.688 | 0.008 | 5.859% |

To empirically validate the sensitivity of **DeepFaith** to hyperparameters, we conducted a study on the OCT+ResNet explanation task, testing the impact of varying the similarity threshold $\delta$, quantile $p$, monitoring window size $e$, and scaling factor $C$ on **DeepFaith** (with $\epsilon$ set to a small positive value close to 0, using $\epsilon = 10^{-5}$ as in Appendix F). The configurations are as follows:

- DeepFaith$_{\delta,p,e,C}$: the original configuration from the paper, $\delta = 0.88$, $p = 0.14$, $e = 5$, $C = 10$ and $k = 100$.

- DeepFaith$_{\delta\uparrow,p,e,C,k}$: similarity threshold increased to $\delta = 0.95$, representing a stricter deduplication policy.

- DeepFaith$_{\delta\downarrow,p,e,C,k}$: similarity threshold decreased to $\delta = 0.70$, representing a more permissive deduplication policy.

- DeepFaith$_{\delta,p\uparrow,e,C,k}$: filtering quantile increased to $p = 0.2$, representing a stricter filtering policy.

- DeepFaith$_{\delta,p\downarrow,e,C,k}$: filtering quantile decreased to $p = 0.05$, representing a more permissive filtering policy.

- DeepFaith$_{\delta,p,e\uparrow,C,k}$: window size increased to $e = 10$, making the algorithm less sensitive to variance changes in $\mathcal{L}_{\text{PC}}$.

- DeepFaith$_{\delta,p,e\downarrow,C,k}$: window size reduced to $e = 3$, making the algorithm more sensitive to variance changes in $\mathcal{L}_{\text{PC}}$.

- DeepFaith$_{\delta,p,e,C\uparrow,k}$: scaling factor increased to $C = 20$, increasing the magnitude of $\alpha$-updates.

- DeepFaith$_{\delta,p,e,C\downarrow,k}$: scaling factor reduced to $C = 5$, decreasing the magnitude of $\alpha$-updates.

- DeepFaith$_{\delta,p,e,C,k\downarrow}$: sampling frequency $k = 10$, resulting in faster training but lower sampling precision.

- DeepFaith$_{\delta,p,e,C,k\uparrow}$: sampling frequency $k = 200$, leading to slower training but higher sampling precision.

We report the performance of these variants of **DeepFaith** across 10 faithfulness evaluation metrics and compute the overall faithfulness deviation $|\Delta F| = \left| \frac{\sum_{i \in [10]} F_i - \sum_{i \in [10]} F_i^{\text{ori}}}{\sum_{i \in [10]} F_i^{\text{ori}}} \right|$, where $F_i$ and $F_i^{\text{ori}}$ denote the score of the $i$-th faithfulness metric for the current variant and for DeepFaith$_{\delta,p,e,C,k}$ respectively. The experimental results are reported in Table 29.

The experimental results in the table show that **DeepFaith** exhibits similar explanation faithfulness across different hyperparameter settings for $\delta$, $p$, $e$, $C$ and $k$, with an overall faithfulness deviation

that is relatively small ($< 6\%$). This demonstrates that **DeepFaith** is sufficiently robust to the introduced hyperparameters. In practice, we recommend keeping the parameter ranges within a reasonable scope, without requiring excessive manual tuning.

## K  REPRODUCIBILITY STATEMENT

We have made every effort to ensure that the results presented in this paper are reproducible. All code have been made publicly available in a github repository to facilitate replication and verification. The experimental setup, including training steps, model configurations, and hardware details, is described in detail in the paper.

Additionally, public datasets used in the paper, such as ImageNet (Deng et al., 2009), OCT (Kermany et al., 2018), AGNews (Zhang et al., 2016), IMDb (Maas et al., 2011), NAP (National Center for Health Statistics, 2019) and WCD (Cardoso, 2013), are publicly available, ensuring consistent and reproducible evaluation results.

## L  LLM USAGE STATEMENT

Large Language Models (LLMs) were used to aid in the writing and polishing of the manuscript. Specifically, we used an LLM to assist in refining the language, improving readability, and ensuring clarity in various sections of the paper. The model helped with tasks such as sentence rephrasing, grammar checking, and enhancing the overall flow of the text.

It is important to note that the LLM was not involved in the ideation, research methodology, or experimental design. All research concepts, ideas, and analyses were developed and conducted by the authors. The contributions of the LLM were solely focused on improving the linguistic quality of the paper, with no involvement in the scientific content or data analysis.

The authors take full responsibility for the content of the manuscript, including any text generated or polished by the LLM. We have ensured that the LLM-generated text adheres to ethical guidelines and does not contribute to plagiarism or scientific misconduct.

