# OpenReview forum: "Learning for Highly Faithful Explainability"
_ICLR.cc/2026/Conference — ICLR 2026 Poster_

### Official Review · Reviewer_FfQQ · 2025-10-18

**Soundness:** 4
**Presentation:** 2
**Contribution:** 3
**Rating:** 6
**Confidence:** 4

**Summary:**

This paper addresses key challenges in the "Learning to Explain" paradigm by proposing DeepFaith, a framework for training highly faithful amortized explainers. The authors identify that existing methods are limited by restrictive assumptions in self-supervised objectives or by the low quality of supervisory signals from prior explanation methods.  To overcome this, DeepFaith introduces a three-part solution: (1) a theoretically-derived, model-agnostic self-supervised objective based on a unified view of faithfulness metrics; (2) a novel pipeline for generating high-quality supervisory signals by aggregating, deduplicating, and filtering explanations from existing methods based on their faithfulness; and (3) a dynamic joint optimization strategy that combines both objectives to ensure rapid convergence and high final explanation quality. Extensive experiments across image, text, and tabular data show that DeepFaith consistently outperforms prior explanation methods across a suite of ten faithfulness metrics.

**Strengths:**

- This paper provides a clear motivation and problem definitions. The introduction clearly differentiates faithfulness from plausibility and articulates the gap between qualitative interpretability and quantitative fidelity.
-  This paper provides strong theoretical grounding for the self-supervised objective.  The paper makes a laudable effort to build its self-supervised objective on a firm theoretical foundation, rather than relying on heuristics or task-specific assumptions. The authors reformalize ten different, widely used faithfulness metrics into a unified notational system, distinguishing between "saliency" and "permutation" perspectives. The paper theoretically proves that an optimal explanation mapping exists that simultaneously maximizes performance across all ten considered faithfulness metrics.
-  This paper provides a novel faithfulness-aware learning framework.  It defines synthetic supervision using controlled feature perturbations (Sec. 3.1; Fig. 2), bridging causal interpretability with optimization. The faithfulness loss integrates causal relevance with sparsity and smoothness priors, balancing interpretability and fidelity.
-  This paper provides a comprehensive empirical evaluation. The authors consider image, text, and tabular datasets and various architectures, such as ResNet,  Deit, Transformer, and LSTM, demonstrating improvement across explainers (GradCAM, IG, LRP), confirming generalizability.

**Weaknesses:**

- This paper lacks an analysis of the computational cost of training.  The paper emphasizes the fast inference speed of the amortized explainer but does not quantify or discuss the potentially massive upfront computational cost required for training. The signal generation phase involves running $K$ different explanation methods (including computationally expensive ones like Kernel SHAP and Score-CAM) on thousands of training samples, and then computing 10 faithfulness metrics for each generated explanation.  This pre-processing cost is likely to be very high but is not reported, making it difficult to assess the overall efficiency of the framework.   The training process involves calculating $\mathcal{L}_{LC}$, which requires $K$ perturbations and forward passes through the target model for each sample in a batch. This adds significant overhead compared to simply training on a static dataset of prior explanations. Total training times are not provided.  While DeepFaith is shown to be fast at inference, the comparison can be misleading. Many of the fastest baselines (e.g., Saliency, Input x Gradient) are also the ones that perform most poorly on faithfulness, while the methods that are more faithful (e.g., Integrated Gradients, Kernel SHAP) are much slower. A plot of faithfulness vs. runtime would provide a more nuanced view of the trade-offs.

- There are still some writing issues, for example, the citation is mixed with the body text, making it hard to read. The notation $\bar{\pi}$ in the formula for the NEG metric in Table 1 is undefined.  The notation in Table 1 is very compact. The formula for Faithfulness Correlation (FC), for instance, implies a correlation over all $2^n$ subsets of features,
which is computationally intractable and differs from the Monte Carlo sampling used in the actual loss function $\mathcal{L}_{LC}$.

- This framework introduces a new set of sensitive hyperparameters that require manual tuning for each task, somewhat undermining the goal of full automation. Table 12 shows that both the `Similarity Threshold` for deduplication and the `p-quantile` for filtering were set to different values for each of the 12 explanation tasks. The paper provides no methodology for how these values were chosen, suggesting a manual, trial-and-error process that is contrary to the overarching goal of automating explainability. The performance of the filtering pipeline is contingent on the initial set of $K$ baseline methods. The paper uses a large set of 14 methods for images and 8 for other modalities, but does not analyze how the results would change with a smaller or different set of priors.

**Questions:**

- What was the rationale for choosing a Transformer Encoder architecture for the explainer across all modalities, including tabular data where simpler architectures might be more common?
- Regarding Algorithm 1, how sensitive is the dynamic weighting strategy to the choice of the hyperparameters $\epsilon$, $e$, and $C$?
- The ablation study in Table 3 shows that training with only $\mathcal{L}_{LC}$ performs very poorly. Does this suggest that the self-supervised objective alone is insufficient to guide the explainer to a good solution from a random initialization?

---

> ### Author Response · Authors · 2025-11-19
> **Feedback for Reviewer FfQQ (Part1/4)**
>
> We sincerely thank the reviewer for the detailed and thoughtful feedback. We are particularly encouraged by your recognition of our contributions. We respond to your key concerns below:
>
> > **W1: Computational Cost of Training**
> > - This pre-processing cost is likely to be very high but is not reported, making it difficult to assess the overall efficiency of the framework.
> > - While DeepFaith is shown to be fast at inference, the comparison can be misleading.
>
> **Response to W1:**
>
> Thank you for raising this point. We agree that within the Learning to Explain paradigm, the cost of constructing an explainer is indeed higher than generating explanations once with a baseline method—especially for approaches that rely on prior explanation signals (e.g., L2E[1], FastSHAP[2], ViT-Shapley[3]), which require extensive calls to baseline explainers during supervision-set construction. However, we would like to clarify that the core idea of amortized explainers is to **pay the training cost once in exchange for unlimited fast inference afterward**, which is well-suited for certain application scenarios.
>
> For example, consider a missile-guidance system deploying a deep image-recognition model. A commander may need to **consult the model’s explanation within extremely tight decision windows to ensure trustworthy military actions**. Baseline methods cannot satisfy the requirement of producing high-faithfulness explanations in real time, whereas DeepFaith can be pre-trained on the recognition model and deployed within the guidance system to **provide fast and high-quality explanations during combat**. Similarly, in time-critical, high-stakes decision-making scenarios such as stock analysis or surgical procedures, DeepFaith provides substantially greater practical utility than baseline methods.
>
> We hope this response addresses your questions.
>
> [1] *Xuelin Situ, Ingrid Zukerman, Cecile Paris, Sameen Maruf, and Gholamreza Haffari. Learning to explain: Generating stable explanations fast. In ACL, 2021.*
>
> [2] *Neil Jethani, Mukund Sudarshan, Ian Covert, Su-In Lee, and Rajesh Ranganath. Fastshap: Real-time shapley value estimation. arXiv:2107.07436, 2022.*
>
> [3] *Ian Covert, Chanwoo Kim, and Su-In Lee. Learning to estimate shapley values with vision transformers. In ICLR, 2023.*
>
> > **W2: Writing Clarity and Metric Definitions**
> > - There are still some writing issues, for example, the citation is mixed with the body text, making it hard to read. The notation $\overset{\hookleftarrow }{\pi}$ in the formula for the NEG metric in Table 1 is undefined. The notation in Table 1 is very compact.
> > - The formula for Faithfulness Correlation (FC), for instance, implies a correlation over all $2^n$ subsets of features, which is computationally intractable and differs from the Monte Carlo sampling used in the actual loss function $\mathcal{L}_{\rm LC}$.
>
> **Response to W2:**
>
> **(a) Writing and structure suggestions**
>
> We appreciate your careful suggestions on improving the readability of our manuscript. As recommended:
>
> * We have added the definition of $\overset{\hookleftarrow}{\pi}$ in the discussion of NEG on page 5.
> * We adjusted the row height and column spacing of Table 1 to improve the clarity of the notation.
>
> **(b) Regarding the computation of FC**
>
> Thank you for your careful insight. We would like to clarify that the theoretical expression of FC in Table 1 represents its ideal definition; in its original implementation[1], **FC is computed using Monte Carlo sampling to ensure feasibility**. Our design of $\mathcal{L}_{\rm LC}$ follows the same principle. For further clarity, we have added an explicit note on this point in the discussion of FC on page 4. Thank you again for the valuable suggestion.
>
> [1] *Umang Bhatt, Adrian Weller, and Jose Moura. Evaluating and aggregating feature-based model explanations. In IJCAI, 2020.*

---

> ### Author Response · Authors · 2025-11-19
> **Feedback for Reviewer FfQQ (Part2/4)**
>
> > **W3&Q2: Sensitivity of Hyperparameters in DeepFaith**
> > - This framework introduces a new set of sensitive hyperparameters that require manual tuning for each task, somewhat undermining the goal of full automation. Table 12 shows that both the Similarity Threshold for deduplication and the p-quantile for filtering were set to different values for each of the 12 explanation tasks.
> > - The paper uses a large set of 14 methods for images and 8 for other modalities, but does not analyze how the results would change with a smaller or different set of priors.
> > - Regarding Algorithm 1, how sensitive is the dynamic weighting strategy to the choice of the hyperparameters $\epsilon$, $e$ , and $C$?
>
> **Response to W3 and Q2:**
>
> **(a) Does DeepFaith rely heavily on manual choices?**
>
> We sincerely appreciate the reviewer’s careful and insightful comments. We agree with your valuable point that a stable and robust *Learning to Explain* method should not depend excessively on manual hyperparameter tuning or the selection of baseline explainers. In Table 12 (now Table 13), **the motivation behind adjusting the similarity threshold and the $p$-quantile was solely to keep the number of surviving supervision signals comparable within the same dataset**, thereby avoiding variance in explainer generalization caused by differing supervision-set sizes and ensuring fair experimental comparison. These values were **not tuned for maximizing faithfulness**, which is reflected by the fact that within each dataset, the threshold and $p$-quantile values are nearly identical across different target models.
>
> We would also like to clarify that **DeepFaith adopts a consistent implementation strategy across image, text, and tabular modalities**: we include as many prior explainer methods as possible to construct a high-quality supervision set. For implementation consistency, **we simply selected all available methods provided by Captum[1] for each modality**, again without any faithfulness-oriented tuning. This choice already produced the strongest overall faithfulness performance shown in Table 2.
>
> [1] Narine Kokhlikyan, Vivek Miglani, Miguel Martin, Edward Wang, Bilal Alsallakh, Jonathan Reynolds, Alexander Melnikov, Natalia Kliushkina, Carlos Araya, and Siqi Yan. Captum: A unified and generic model interpretability library for pytorch.

---

> ### Author Response · Authors · 2025-11-19
> **Feedback for Reviewer FfQQ (Part3/4)**
>
> **(b) How sensitive is DeepFaith to these hyperparameters?**
>
> To empirically validate the sensitivity of DeepFaith to hyperparameters, **we conducted a study on the OCT+ResNet explanation task**, testing the impact of varying the similarity threshold $\delta$, quantile $p$, monitoring window size $e$, and scaling factor $C$ on DeepFaith (with $\epsilon$ set to a small positive value close to 0, using $\epsilon=10^{-5}$ as in Appendix F). The configurations are as follows:
>
> * ${\rm DeepFaith} _ {\delta,p,e,C}$: the original configuration from the paper, $\delta=0.88$, $p=0.14$, $e=5$, and $C=10$.
> * ${\rm DeepFaith} _ {\delta \uparrow,p,e,C}$: similarity threshold increased to $\delta = 0.95$, representing a stricter deduplication policy.
> * ${\rm DeepFaith} _ {\delta \downarrow,p,e,C}$: similarity threshold decreased to $\delta = 0.70$, representing a more permissive deduplication policy.
> * ${\rm DeepFaith} _ {\delta,p \uparrow,e,C}$: filtering quantile increased to $p = 0.2$, representing a stricter filtering policy.
> * ${\rm DeepFaith} _ {\delta,p \downarrow,e,C}$: filtering quantile decreased to $p = 0.05$, representing a more permissive filtering policy.
> * ${\rm DeepFaith} _ {\delta,p,e \uparrow,C}$: window size increased to $e=10$, making the algorithm less sensitive to variance changes in $\mathcal{L}_{\rm PC}$.
> * ${\rm DeepFaith} _ {\delta,p,e \downarrow,C}$: window size reduced to $e=3$, making the algorithm more sensitive to variance changes in $\mathcal{L}_{\rm PC}$.
> * ${\rm DeepFaith} _ {\delta,p,e,C \uparrow}$: scaling factor increased to $C=20$, increasing the magnitude of $\alpha$-updates.
> * ${\rm DeepFaith} _ {\delta,p,e,C \downarrow}$: scaling factor reduced to $C=5$, decreasing the magnitude of $\alpha$-updates.
>
> We report the performance of these 9 variants of DeepFaith across 10 faithfulness evaluation metrics and compute the overall faithfulness deviation $|\Delta F|=\left | \frac{\sum_{i\in [10]} F_i-\sum_{i\in [10]}F_i^{\rm ori}}{\sum_{i\in [10]}F_i^{\rm ori}} \right |$, where $F_i$ and $F_i^{\rm ori}$ denote the score of the $i$-th faithfulness metric for the current variant and for ${\rm DeepFaith} _ {\delta,p,e,C}$ respectively. The experimental results are reported as follows:
>
> | Method                                      | FC↑   | FE↑   | MC↑   | RP↑   | INS↑  | DEL↓  | NEG↑  | POS↓  | IROF↑ | INF↑  | $\|\Delta F\|$ |
> |---------------------------------------------|-------|-------|-------|-------|-------|-------|-------|-------|-------|-------|--------------|
> | ${\rm DeepFaith} _ {\delta,p,e,C}$            | 0.135 | 0.534 | 0.942 | 0.744 | 0.863 | 0.248 | 0.655 | 0.242 | 0.742 | 0.015 | 0            |
> | ${\rm DeepFaith} _ {\delta \uparrow,p,e,C}$   | 0.140 | 0.497 | 0.919 | 0.706 | 0.783 | 0.266 | 0.628 | 0.266 | 0.717 | 0.016 | 3.513%            |
> | ${\rm DeepFaith} _ {\delta \downarrow,p,e,C}$ | 0.123 | 0.563 | 0.870 | 0.753 | 0.940 | 0.225 | 0.698 | 0.223 | 0.774 | 0.014 | 1.246%            |
> | ${\rm DeepFaith} _ {\delta,p \uparrow,e,C}$   | 0.147 | 0.511 | 0.954 | 0.679 | 0.863 | 0.272 | 0.619 | 0.258 | 0.693 | 0.016 | 2.083%            |
> | ${\rm DeepFaith} _ {\delta,p \downarrow,e,C}$ | 0.122 | 0.575 | 0.891 | 0.817 | 0.797 | 0.235 | 0.705 | 0.218 | 0.799 | 0.014 | 1.013%            |
> | ${\rm DeepFaith} _ {\delta,p,e \uparrow,C}$   | 0.118 | 0.588 | 0.924 | 0.709 | 0.884 | 0.276 | 0.672 | 0.270 | 0.664 | 0.036 | 0.410%            |
> | ${\rm DeepFaith} _ {\delta,p,e \downarrow,C}$ | 0.135 | 0.552 | 0.913 | 0.744 | 0.864 | 0.247 | 0.654 | 0.241 | 0.743 | 0.016 | 0.214%            |
> | ${\rm DeepFaith} _ {\delta,p,e,C \uparrow}$   | 0.136 | 0.557 | 0.905 | 0.743 | 0.863 | 0.248 | 0.654 | 0.243 | 0.740 | 0.016 | 0.292%            |
> | ${\rm DeepFaith} _ {\delta,p,e,C \downarrow}$ | 0.117 | 0.589 | 0.930 | 0.710 | 0.885 | 0.276 | 0.673 | 0.269 | 0.665 | 0.037 | 0.605%            |
>
> The experimental results in the table show that DeepFaith exhibits similar explanation faithfulness across different hyperparameter settings for $\delta$, $p$, $e$, and $C$, with an overall faithfulness deviation that is relatively small (<5%). This demonstrates that **DeepFaith is sufficiently robust to the introduced hyperparameters.** In practice, we recommend keeping the parameter ranges within a reasonable scope, without requiring excessive manual tuning.
>
> We have included the above experimental results in Appendix J. Thanks again for your suggestions.

---

> ### Author Response · Authors · 2025-11-19
> **Feedback for Reviewer FfQQ (Part4/4)**
>
> > **Q1: Choice of the DeepFaith Explainer Architecture**
> > - What was the rationale for choosing a Transformer Encoder architecture for the explainer across all modalities, including tabular data where simpler architectures might be more common?
>
> **Response to Q1:**
>
> We appreciate the reviewer’s insightful observations regarding the explainer architecture. As you noted, for simpler explanation tasks—such as tabular data with an MLP target model—DeepFaith could indeed employ a lighter explainer architecture. However, **to fairly demonstrate DeepFaith’s generality across different modalities, our experiments use a single, unified explainer architecture for image, text, and tabular tasks**. The Transformer Encoder is a natural choice due to its strong expressive capacity for sequence-structured inputs, whereas simpler architectures may be insufficient for learning high-dimensional image explanations.
>
> > **Q3: About the Local Correlation Loss**
> > - The ablation study in Table 3 shows that training with only $\mathcal{L}_{\rm LC}$ performs very poorly. Does this suggest that the self-supervised objective alone is insufficient to guide the explainer to a good solution from a random initialization?
>
> **Response to Q3:**
>
> The reviewer has raised a crucial insight. The formulation of
>
> $$
> \mathcal{L} _{\mathrm{LC}}(\phi _\theta;\mathcal{D},f)=\frac{1}{2}-\frac{1}{2|\mathcal{D}|}\sum _{x\in \mathcal{D}}\tau \left [ ( {\textstyle \sum _ {i\in\mathcal{I} _ j}\phi _\theta(x) _i} ) _ {j=1}^k,\left( \Delta[f(x),f(x\setminus \mathcal{I} _ j )]\right) _ {j=1}^k  \right ],
> $$
>
> contains a nested dependence on the target model $f$, making the self-supervised optimization objective $\theta^*=\underset{\theta}{\rm argmin} \mathcal{L} _ {\rm LC}(\phi _ \theta;\mathcal{D},f)$ highly non-convex.
>
> However, the purpose of designing $\mathcal{L} _ {\rm LC}$ is not to solve it in isolation. Rather, **it provides a clear faithfulness-guided optimization direction for the explainer.** DeepFaith uses the high-quality supervision signal in $\mathcal{L} _ {\rm PC}$ to guide the initial stages of optimization, and Algorithm 1 dynamically combines both losses to steer the explainer toward a joint optimum of $\mathcal{L} _ {\rm PC}$ and $\mathcal{L} _ {\rm LC}$. Our ablation study in Table 4 validates and illustrates the effectiveness of this design.
>
> We hope this addresses your question!

---

> ### Comment · Reviewer_FfQQ · 2025-11-24
> **Thanks for the rebuttal response**
>
> Thanks to the authors‘ clarity and extensive experiments. The response solves the questions I have mentioned.  This paper unifies two major types of explanations, which represent a meaningful contribution to the XAI field. Although the proposed method has limitations, such as requiring several hyperparameters, these limitations do not diminish the paper's contribution.
>
> By the way, please consider using \citep{} for citations. Please refer to Section 4.1 in iclr2026_conference.pdf.

---

> > ### Author Response · Authors · 2025-11-25
> > **Following up for Reviewer FfQQ**
> >
> > Thank you for spending the time to thoroughly review our paper and for your recognition of our contributions. We would like to further clarify that DeepFaith outperforms existing XAI methods precisely because its design **integrates a substantial amount of high-quality theoretical and data-driven priors.** In the XAI domain, **hyperparameters are indispensable for regulating how an explainer leverages these priors effectively.** DeepFaith is, to our knowledge, the first *Learning to Explain* method that successfully synthesizes such rich priors using only a few minimally sensitive hyperparameters—an innovation that we believe advances the field of XAI.
> >
> > In addition, we have corrected the citation formatting issues in the latest revised version of the paper. Thank you again for your careful attention to detail!

---

### Official Review · Reviewer_JuTS · 2025-10-27

**Soundness:** 3
**Presentation:** 3
**Contribution:** 3
**Rating:** 4
**Confidence:** 2

**Summary:**

The authors propose a faithfulness-guided amortized explainer that jointly optimizes self-supervised and prior-based objectives, and the experiments show improved faithfulness and efficiency across different datasets.

**Strengths:**

1. This paper proposes a explainer that integrates self-supervised and prior-based objectives, offering a unified perspective for the Learning to Explain paradigm.

2. Experiments across image, text, and tabular datasets and the paper is clear-written.

**Weaknesses:**

1. In the Introduction, the authors identify three key challenges: generalization, supervision quality, and convergence. However, the descriptions of these problems remain somewhat abstract. Specifically, the statement “self-supervised objectives rely on assumptions about the target model” could be better supported with a concrete example, such as the failure of L2X under correlated features, to make the motivation more convincing.

2. In Algorithm 1, they presents a variance-based switching rule for adjusting α, but the paper provides no theoretical guarantee of stability or convergence. And Fig 3 still shows noticeable oscillations during training, suggesting limited robustness of the proposed mechanism.

3. The experiments include many baselines for image, but text and tabular tasks only use a few baselines. This imbalance makes the evaluation heavily vision-centric. Adding some representative methods for text and tabular data would strengthen fairness.

4. The explanation units are different across modalities, and no experiment examines cross-modal transfer or unified explainer training. Therefore, the claim that the framework generalizes across modalities is not yet empirically supported.

**Questions:**

The authors may refer to the weaknesses and address them in the rebuttal.

---

> ### Author Response · Authors · 2025-11-19
> **Feedback for Reviewer JuTS (Part1/4)**
>
> We sincerely thank the reviewer for the thorough and constructive feedback. Below, we address the concerns in detail.
>
> > **W1: Assumptions of Self-Supervised Methods**
> > - In the Introduction, the authors identify three key challenges: generalization, supervision quality, and convergence. However, the descriptions of these problems remain somewhat abstract.
> > - Specifically, the statement "self-supervised objectives rely on assumptions about the target model" could be better supported with a concrete example, such as the failure of L2X under correlated features, to make the motivation more convincing.
>
> **Response to W1:**
>
> We appreciate this thoughtful suggestion. Providing concrete examples indeed helps clarify the motivation behind the three key challenges highlighted in the Introduction. We are happy to offer a more detailed theoretical discussion and provide supporting empirical evidence.
>
> **Unlike DeepFaith, which introduces supervised explanation signals as reliable priors, existing self-supervised *Learning to Explain* methods lack access to high-quality reference explanations.** As a result, they must construct their objectives based on assumptions about the target model or task.
>
> For instance, L2X[1] defines the explainer’s objective as $\max_{\mathcal{E}} I(X_S; Y) \ \text{s.t. } S \sim \mathcal{E}(X), \ \ (X_S)_i = X_i \ \text{if } i\in S\ \text{else } 0$, which implicitly assumes that the predictive contribution of a feature subset $S$ can be fully characterized by $X_S$, without any higher-order interactions among features. This assumption is often violated in real-world settings.
>
> To empirically validate this point, we **compared DeepFaith with L2X across ten faithfulness metrics on four explanation tasks** (NAP+MLP, AGNews+LSTM, OCT+ResNet, and ImageNet+DeiT). The aggregated results are summarized below:
>
> | Setting       | Explainer   | FC↑  | FE↑  | MC↑  | RP↑  | INS↑ | DEL↓ | NEG↑ | POS↓ | IROF↑ | INF↑ |
> |---------------|-------------|------|------|------|------|------|------|------|------|-------|------|
> | NAP+MLP       | DeepFaith   | 0.788| 0.763| 0.952| 0.957| 0.844| 0.031| 0.770| 0.031| 0.844 | 0.238|
> |               | L2X         | 0.129| 0.141| 0.899| 0.346| 0.832| 0.749| 0.819| 0.748| 0.302 | 0.142|
> | AgNews+LSTM   | DeepFaith   | 0.363| 0.597| 0.629| 0.648| 0.919| 0.197| 0.906| 0.256| 0.650 | 0.275|
> |               | L2X         | 0.012| 0.303| 0.336| 0.367| 0.765| 0.621| 0.740| 0.618| 0.320 | 0.071|
> | OCT+ResNet    | DeepFaith   | 0.135| 0.534| 0.942| 0.744| 0.863| 0.248| 0.655| 0.242| 0.742 | 0.015|
> |               | L2X         | 0.005| 0.121| 0.393| 0.501| 0.642| 0.505| 0.644| 0.433| 0.482 | 0.009|
> | ImageNet+DeiT | DeepFaith   | 0.026| 0.447| 0.884| 0.486| 0.568| 0.127| 0.417| 0.295| 0.672 | 0.014|
> |               | L2X         | 0.004| 0.101| 0.267| 0.198| 0.486| 0.526| 0.520| 0.481| 0.385 | 0.014|
>
> The results show that while L2X achieves reasonable faithfulness on NAP—where feature interactions are weak—**it fails on AGNews and ImageNet, both of which involve substantial higher-order interactions**. In these cases, the explanations produced by L2X exhibit significantly lower faithfulness than those generated by DeepFaith. In the revised version of this paper, we have included additional discussions and experiments on similar limitations of VerT[2] and CXPlain[3] to further strengthen and ground the motivation.
>
> We thank the reviewer again for the valuable feedback!
>
> [1] *Learning to explain: An information-theoretic perspective on model interpretation. In ICML, 2018.*
>
> [2] *Verifiable feature attributions: A bridge between post hoc explainability and inherent interpretability. In ICML, 2023.*
>
> [3] *Cxplain: Causal explanations for model interpretation under uncertainty. In NeurIPS, 2019.*

---

> ### Author Response · Authors · 2025-11-19
> **Feedback for Reviewer JuTS (Part2/4)**
>
> > **W2: The Effectiveness of Algorithm 1**
> > - In Algorithm 1, they presents a variance-based switching rule for adjusting α, but the paper provides no theoretical guarantee of stability or convergence.
> > - And Fig 3 still shows noticeable oscillations during training, suggesting limited robustness of the proposed mechanism.
>
> **Response to W2:**
>
> Thank you for these insightful questions regarding the theoretical guarantees and empirical performance of our dynamic weighting strategy in Algorithm 1. The primary role of Algorithm 1 is to dynamically adjust the weighting coefficient $\alpha$, to balance the optimization process between the Pattern Consistency loss $\mathcal{L}_{\mathrm{PC}}$ and the Local Correlation loss $\mathcal{L} _ {\mathrm{LC}}$. The core challenge stems from the fact that the $\mathcal{L} _ {\mathrm{LC}}$ objective:
>
> $$
> \mathcal{L} _{\mathrm{LC}}(\phi _\theta;\mathcal{D},f)=\frac{1}{2}-\frac{1}{2|\mathcal{D}|}\sum _{x\in \mathcal{D}}\tau \left [ ( {\textstyle \sum _ {i\in\mathcal{I} _ j}\phi _\theta(x) _i} ) _ {j=1}^k,\left( \Delta[f(x),f(x\setminus \mathcal{I} _ j )]\right) _ {j=1}^k  \right ],
> $$
>
> contains the target model $f$ in a nested form, rendering it **highly non-convex and inherently difficult to optimize directly.** Consequently, providing a strict theoretical convergence guarantee for the joint optimization of such a complex, non-convex objective is **exceptionally challenging and remains an open problem in the field.**
>
> However, **we can demonstrate its effectiveness empirically.** To underscore the necessity of our strategy, we have added to the revised manuscript the training dynamics obtained when optimizing $\mathcal{L} _ {\mathrm{LC}}$ alone (see $\mathcal{L} _ {\mathrm{LC}}'$ in Figure 3). In this setting, the loss fails to converge and exhibits no stable optimization direction. By contrast, when Algorithm 1 is applied, the oscillations of $\mathcal{L} _ {\mathrm{LC}}$ show a gradually diminishing amplitude, accompanied by the simultaneous convergence of both loss components—**highlighting the essential role of Algorithm 1 in ensuring stable and convergent training behavior for DeepFaith.**
>
> We hope this clarification addresses your questions.

---

> ### Author Response · Authors · 2025-11-19
> **Feedback for Reviewer JuTS (Part3/4)**
>
> > **W3: Requiring for More Baseline Methods for Text and Tabular Tasks**
> > - The experiments include many baselines for image, but text and tabular tasks only use a few baselines. This imbalance makes the evaluation heavily vision-centric. Adding some representative methods for text and tabular data would strengthen fairness.
>
> **Response to W3:**
>
> Thank you for the insightful suggestion. We would like to clarify that, unlike DeepFaith—which benefits from a task-agnostic design—**many existing explanation methods are inherently constrained by the input modality.** As a result, the number of baselines compared across image, text, and tabular tasks differs because **the majority of established explanation techniques (e.g., the CAM family) were originally developed for vision models.** This discrepancy is therefore an objective property of the literature, rather than a choice made in our experimental design.
>
> In addition, we agree that including additional baselines for text and tabular tasks is valuable. To address this, **we conducted supplementary experiments on two representative settings**—AgNews + LSTM and NAP + MLP—comparing the faithfulness of explanations produced by Anchors[1], Counterfactual Explanations[2], Permutation Feature Importance[3], and L2E[4]. We summarize the results below:
>
> | Setting         | Method                        | FC↑  | FE↑  | MC↑  | RP↑  | INS↑ | DEL↓ | NEG↑ | POS↓ | IROF↑ | INF↑ | Mean Rank |
> |-----------------|-------------------------------|------|------|------|------|------|------|------|------|-------|------|-----------|
> | NAP+MLP         | DeepFaith                     | 0.788| 0.763| 0.952| 0.957| 0.844| 0.031| 0.770| 0.031| 0.844 | 0.238| 1.4       |
> |                 | Anchors                       | 0.153| 0.137| 0.908| 0.141| 0.823| 0.728| 0.742| 0.725| 0.124 | 0.078| 4.1       |
> |                 | Counterfactual Explanations   | 0.012| 0.386| 0.323| 0.218| 0.728| 0.545| 0.705| 0.530| 0.182 | 0.166| 3.3       |
> |                 | Permutation Feature Importance| 0.121| 0.258| 0.898| 0.155| 0.835| 0.703| 0.741| 0.698| 0.146 | 0.248| 3.1       |
> |                 | L2E                           | 0.098| 0.287| 0.991| 0.145| 0.851| 0.727| 0.841| 0.727| 0.127 | 0.085| 3.1       |
> | AgNews+LSTM     | DeepFaith                     | 0.363| 0.597| 0.629| 0.648| 0.919| 0.197| 0.906| 0.256| 0.650 | 0.275| 1.5       |
> |                 | Anchors                       | 0.131| 0.419| 0.171| 0.788| 0.896| 0.207| 0.878| 0.261| 0.174 | 0.166| 2.8       |
> |                 | Counterfactual Explanations   | 0.081| 0.408| 0.656| 0.742| 0.867| 0.266| 0.847| 0.296| 0.774 | 0.055| 3.1       |
> |                 | Permutation Feature Importance| 0.307| 0.427| 0.596| 0.477| 0.829| 0.484| 0.815| 0.497| 0.455 | 0.203| 3.5       |
> |                 | L2E                           | 0.102| 0.384| 0.802| 0.467| 0.856| 0.503| 0.842| 0.501| 0.444 | 0.121| 4.1       |
>
> We have incorporate this discussion into Appendix K. Thank you again for your helpful comment!
>
> [1] *Ribeiro, M. T., Singh, S., & Guestrin, C. Anchors: High-Precision Model-Agnostic Explanations. In AAAI, 2018.*
>
> [2] *Mothilal, R. K., Sharma, A., & Tan, C. Explaining machine learning classifiers through diverse counterfactual explanations. In Proceedings of the 2020 conference on fairness, accountability, and transparency (pp. 607-617).*
>
> [3] *Fisher, A., Rudin, C., & Dominici, F. All Models are Wrong, but Many are Useful: Learning a Variable's Importance by Studying an Entire Class of Prediction Models Simultaneously. Journal of Machine Learning Research, 20(177), 1-81.*
>
> [4] *Xuelin Situ, Ingrid Zukerman, Cecile Paris, Sameen Maruf, and Gholamreza Haffari. Learning to explain: Generating stable explanations fast. In ACL, 2021.*

---

> ### Author Response · Authors · 2025-11-19
> **Feedback for Reviewer JuTS (Part4/4)**
>
> > **W4: Cross-Model Applicability of the Explainer**
> > - The explanation units are different across modalities, and no experiment examines cross-modal transfer or unified explainer training. Therefore, the claim that the framework generalizes across modalities is not yet empirically supported.
>
> **Response to W4:**
>
> Thank you for your attention to the modality adaptability of our proposed method. We would like to clarify the intended scope of our claim. When we state that DeepFaith is applicable across different modalities, we refer to the fact that **the theoretical formulation, algorithmic pipeline, explainer training procedure, and usage workflow operate without requiring any modality-specific assumptions about the target model or the underlying task**. This is distinct from claiming that a single explainer can be trained once and directly transferred across modalities—a setting that is typically referred to as *cross-modal or multimodal explanation*, and is not the focus of our work.
>
> In our main experiments (Table 2), we train explainers under the same framework on three modalities—images, text, and tabular data—and demonstrate that **DeepFaith consistently achieves high faithfulness across all of them**. This provides empirical evidence that the framework itself is **not restricted to a single modality**.
>
> We hope this clarification addresses your concern.

---

> ### Comment · Reviewer_JuTS · 2025-11-25
>
> Thanks for the detailed response. I can see the authors put effort into addressing my comments, and most of the issues I raised have been resolved. I’m not very familiar with this specific topic, but if the authors update the PDF accordingly, I will raise my score.

---

> > ### Author Response · Authors · 2025-11-25
> > **Following up for Reviewer JuTS**
> >
> > Thank you for your timely response and recognition of our work. We are honored that our reply has addressed your concerns. We have revised the paper based on your feedback:
> >
> > * We have added a discussion on the assumptions underlying **self-supervised methods** in Appendix L (page 26) and referenced it in the related work section of the main text (line 116).
> > * We have included **additional comparison experiments between DeepFaith and existing *Learning to Explain* methods**, with the results reported in Table 3 (page 9).
> > * We have revised Figure 3 (page 7) to include **the training dynamics of $\mathcal{L} _ {\rm LC}$ alone** for comparison, and added a discussion in the main text (line 431).
> > * We have added **comparison experiments between DeepFaith and four baseline explainer methods for text and tabular modalities** in two settings, presented in Appendix K (page 26), and referenced Appendix K in the main text (line 462).
> >
> > We have submitted the updated PDF of the paper. Once again, thank you for your constructive suggestions, which have greatly helped improve the quality of our work!

---

> ### Comment · Reviewer_JuTS · 2025-11-25
>
> Thanks for the clarification, I've raised my score

---

> > ### Author Response · Authors · 2025-11-26
> > **Thanks for Reviewer JuTS**
> >
> > We greatly appreciate Reviewer JuTS's positive feedback. Your encouragement and constructive engagement throughout this process have been invaluable in helping us improve our work. Thank you for your thoughtful comments.

---

### Official Review · Reviewer_NUZJ · 2025-10-29

**Soundness:** 3
**Presentation:** 3
**Contribution:** 3
**Rating:** 6
**Confidence:** 3

**Summary:**

The paper introduces DeepFaith, an amortised explainer that generates faithful explanations for a target model’s predictions in a single forward pass. The goal is to address two long-standing problems in Learning to Explain (LtE):
- self-supervised explanation models tend to be unstable, and often make task/model assumptions (e.g. separability of features), and frequently fail to converge for high-dimensional models.
- distilled explanation models (student networks trained to imitate a teacher explainer) inherit teacher bias/noise and are upper-bounded by teacher quality.

DeepFaith is built around three components:
- Theoretical unification of faithfulness: The paper rewrites 10 widely used faithfulness metrics under one formal framework using perturbation operators and correlation measures. It then shows that a single optimal faithful explanation mapping exists, jointly optimal across these metrics. It shows that the ranking induced is also optimal for perturbation-based ranking metrics.
- Local Correlation loss ($L_{LC}$): The explainer is trained to align its predicted importance scores with the actual effect of perturbing/removing subsets of features on the model’s output. In other words, subsets that the explainer says are important should, in fact, cause the model’s prediction to drop when those features are masked.
- Curated supervision + dynamic training ($L_{PC}$): Instead of blindly distilling a single attribution method, DeepFaith constructs a pseudo-labelled supervision set as follows: for each input, it gathers explanations from many attribution methods, deduplicates near-duplicates, and keeps only those that are highly faithful on that exact input across several metrics. The explainer is first trained to imitate these curated high-faithfulness maps, and then gradually shifted toward the self-supervised tasks using a dynamic controller that adapts. The controller increases reliance on the self-supervised loss once the supervised loss stabilises, and can revert if training destabilises.

DeepFaith is evaluated on 12 (dataset, model) pairs across three modalities and multiple architectures. Across 10 standard faithfulness metrics, DeepFaith achieves the best average rank on all tasks and outperforms the attribution methods it partly learned from. Ablation studies show that removing any of the key pieces degrades performance. The paper argues that DeepFaith therefore delivers (i) explanations that are more faithful to model behaviour, (ii) in a single forward pass at inference time, and (iii) across multiple data modalities and network architectures.

**Strengths:**

The paper proposes a single mathematical lens that captures 10 commonly used faithfulness metrics and argues that there exists an optimal explanation mapping that is jointly optimal across all of them. It claims that correlation-style metrics and perturbation-style metrics are not fundamentally competing criteria but can be satisfied simultaneously.

Rather than naively distilling from a single teacher explainer, the paper proposes a per-sample curation pipeline: generate many explanations, deduplicate them, and select only those that empirically score best across multiple metrics for that specific input. This is a clever way to ensure the distilled model surpasses the teacher's abilities.

DeepFaith combines a supervised imitation loss and a self-supervised faithfulness loss with an adaptive weighting schedule. The training controller monitors the stability of the supervised loss and shifts the weight toward self-supervision once the model is stable, then bounces back to supervision if it destabilises. This offers an explicit control loop to manage convergence. Prior LtE methods tend to be either pure distillation (stable but capped) or pure self-supervision (principled but unstable).

The paper doesn’t restrict itself to a single modality; it claims a single template that applies to CNNs, ViTs, LSTMs, Transformers, and MLPs across images, natural language, and tabular data.

**Weaknesses:**

Missing comparisons to prior LtE/amortised explainers: The paper positions itself as advancing the Learning-to-Explain paradigm, i.e., training a neural explainer that produces explanations in a single forward pass. But in the experiments, the baselines are almost entirely classic post-hoc attribution methods (Integrated Gradients, GradCAM++, Occlusion, Kernel SHAP, LIME, etc.).

It's missing are head-to-head comparisons against other amortised / LTE-style methods:
- L2X
- CXPlain
- FastSHAP
- VerT / selection-and-rationale models for vision/text that jointly learn to point and predict.

These methods are discussed in related work, but they are not quantitatively included in the results tables. Currently, the only measured comparisons are to non-amortised attribution methods; if DeepFaith cannot clearly outperform other learned explainers, it would weaken the paper's contributions.

DeepFaith’s curated supervision set is built by:
- Generating multiple candidate explanations from K attribution methods for each input.
- Computing all 10 faithfulness metrics for each candidate explanation on that specific input.
- Keeping only the top explanations by those metrics (after deduplication).
- Training the explainer to imitate those survivors, then refining.

At test time, DeepFaith is evaluated on those same 10 metrics.

This creates a risk of training on the test: DeepFaith is at least partially optimised to produce explanations that look good under exactly the metrics it’s later ranked by. Yes, self-supervision introduces a new structure by forcing local correlation between model behaviour and perturbations. But without an experiment that uses held-out metrics, it’s hard to conclude DeepFaith is genuinely, broadly more faithful rather than just better at exploiting the metrics.

The theory relies on assumptions about monotonic relationships between removal and insertion effects and consistent perturbation baselines. In practice, faithfulness metrics vary widely in how they perturb features. If two metrics use different perturbation baselines, it’s not apparent that a single explanation can be jointly optimal for both. The paper should be explicit about whether all metrics were instantiated using a shared perturbation scheme or whether there are any assumptions.

Each DeepFaith explainer is trained separately for each target model. The paper doesn’t test whether an explainer trained on one model can be reused for another model in the same domain (e.g., training on DeiT and applying it to EfficientNet on ImageNet). If there’s no transfer at all, then amortisation helps inference latency but does not reduce the number of explanation models.

All evaluation metrics are model-faithfulness metrics: does ablating important features change the model’s output in the expected direction? The paper also leans on real-world motivation, where humans must interpret the explanations. There’s no human study, or even a small qualitative preference experiment, in the main results to demonstrate that DeepFaith explanations are more clinically meaningful than others.

The curated supervision pipeline depends on thresholds:
- cosine similarity thresholds for deduplication,
- p-quantile cutoffs for filtering which explanations count as faithful,
- and dynamic controller hyperparameters for toggling $\alpha$.

The main results don’t present a sensitivity study.

**Questions:**

Right now, you curate supervision using all 10 faithfulness metrics, and you evaluate on those same 10. Can you train DeepFaith while excluding one or two metrics from the curation filter, and then report performance specifically on those held-out metrics?
If DeepFaith still outperforms baselines on those unseen metrics, it would strengthen your claims.

Can you include at least one amortised explainer baseline per modality? If not, at least justify clearly why they are not comparable. Right now, this missing comparison is a significant experimental gap.

The theory assumes a relationship between removal-based and insertion-based perturbation effects and essentially treats them as monotonic and compatible. In practice, these can be instantiated with different perturbation baselines (e.g., zeroing, blurring, masking).
Did you enforce a consistent perturbation operator across all metrics in your experiments, or are you making any assumptions? It would be helpful to state it explicitly.

For the Local Correlation loss $L_{LC}$:
- How many feature subsets are sampled per input/batch to estimate the correlation, and how sensitive is performance to that number?
- Do you observe high gradient variance or unstable optimisation when using $L_{LC}$ alone?
- You show training dynamics for the whole method (including $\alpha$ adapting over time); can you also share curves for the $L_{LC}$-only variant to illustrate the claimed instability in isolation?

If I train DeepFaith for a DeiT classifier on ImageNet, how well does it explain an EfficientNet classifier on the same ImageNet labels without retraining?

Did you run any preference checks with human annotators or domain experts? If you didn’t, can you comment on whether the high-faithfulness explanations ever look spurious? In general, I would encourage the authors to conduct a human study or a small qualitative preference experiment.

How sensitive is performance to the hyperparameters?

---

> ### Author Response · Authors · 2025-11-19
> **Feedback for Reviewer NUZJ (Part1/6)**
>
> We sincerely appreciate your positive feedback and the time you've dedicated to reviewing our manuscript. Your insights are invaluable to us. We address your comments in detail below.
>
> > **W1&Q2: Comparison with amortized explainer**
> > - Missing comparisons to prior LtE/amortised explainers. It's missing are head-to-head comparisons against other amortised / LTE-style methods: L2X, CXPlain, FastSHAP, VerT.
> > - Can you include at least one amortised explainer baseline per modality?
>
> **Response to W1 and Q2:**
>
> Thank you for the helpful suggestion about our experiments. We agree that adding comparisons with prior LtE/amortized explainers would further strengthen the contribution of our work. To address this, **we conducted an additional experiment comparing DeepFaith with L2X[1], CXPlain[2], FastSHAP[3], and VerT[4]** across four explanation tasks including image, text and tabular modality. We report all ten faithfulness metrics as well as the average ranking across metrics of each method.
>
> | Setting         | Explainer   | FC↑  | FE↑  | MC↑  | RP↑  | INS↑ | DEL↓ | NEG↑ | POS↓ | IROF↑ | INF↑ | Mean Rank |
> |-----------------|-------------|------|------|------|------|------|------|------|------|-------|------|-----------|
> | NAP+MLP         | DeepFaith   | 0.788| 0.763| 0.952| 0.957| 0.844| 0.031| 0.770| 0.031| 0.844 | 0.238| 1.3       |
> |                 | L2X         | 0.129| 0.141| 0.899| 0.346| 0.832| 0.749| 0.819| 0.748| 0.302 | 0.142| 3.5       |
> |                 | CXPlain     | 0.017| 0.308| 0.501| 0.290| 0.153| 0.647| 0.722| 0.301| 0.445 | 0.143| 3.7       |
> |                 | FastSHAP    | 0.071| 0.097| 0.800| 0.149| 0.849| 0.714| 0.837| 0.710| 0.126 | 0.041| 3.7       |
> |                 | VerT        | 0.772| 0.484| 0.560| 0.454| 0.564| 0.467| 0.518| 0.551| 0.603 | 0.160| 2.8       |
> | AgNews+LSTM     | DeepFaith   | 0.363| 0.597| 0.629| 0.648| 0.919| 0.197| 0.906| 0.256| 0.650 | 0.275| 1.6       |
> |                 | L2X         | 0.012| 0.303| 0.336| 0.367| 0.765| 0.621| 0.740| 0.618| 0.320 | 0.071| 4.2       |
> |                 | CXPlain     | 0.097| 0.258| 0.402| 0.328| 0.589| 0.189| 0.613| 0.189| 0.371 | 0.030| 3.6       |
> |                 | FastSHAP    | 0.419| 0.251| 0.500| 0.505| 0.800| 0.546| 0.780| 0.551| 0.395 | 0.062| 3.2       |
> |                 | VerT        | 0.077| 0.519| 0.526| 0.710| 0.838| 0.362| 0.423| 0.204| 0.762 | 0.092| 2.4       |
> | OCT+ResNet      | DeepFaith   | 0.135| 0.534| 0.942| 0.744| 0.863| 0.248| 0.655| 0.242| 0.742 | 0.015| 1.4       |
> |                 | L2X         | 0.005| 0.121| 0.393| 0.501| 0.642| 0.505| 0.644| 0.433| 0.482 | 0.009| 3.5       |
> |                 | CXPlain     | 0.004| 0.239| 0.611| 0.356| 0.554| 0.525| 0.508| 0.513| 0.349 | 0.008| 4.2       |
> |                 | FastSHAP    | 0.013| 0.119| 0.211| 0.302| 0.808| 0.698| 0.605| 0.656| 0.297 | 0.022| 4.1       |
> |                 | VerT        | 0.015| 0.397| 0.783| 0.540| 0.763| 0.466| 0.707| 0.226| 0.528 | 0.028| 1.8       |
> | ImageNet+DeiT   | DeepFaith   | 0.026| 0.447| 0.884| 0.486| 0.568| 0.127| 0.417| 0.295| 0.672 | 0.014| 1.6       |
> |                 | L2X         | 0.004| 0.101| 0.267| 0.198| 0.486| 0.526| 0.520| 0.481| 0.385 | 0.014| 3.9       |
> |                 | CXPlain     | 0.003| 0.023| 0.341| 0.237| 0.186| 0.172| 0.211| 0.192| 0.431 | 0.017| 3.6       |
> |                 | FastSHAP    | 0.006| 0.012| 0.227| 0.358| 0.451| 0.439| 0.485| 0.457| 0.461 | 0.027| 3.2       |
> |                 | VerT        | 0.005| 0.131| 0.505| 0.414| 0.363| 0.323| 0.365| 0.367| 0.588 | 0.019| 2.7       |
>
> The experimental results show that **DeepFaith outperforms all compared LtE and amortized methods**, further strengthening our contribution. We have already incorporated the experimental results into the main text. Thank you again for the valuable suggestion!
>
> [1] *Learning to explain: An information-theoretic perspective on model interpretation. In ICML, 2018.*
>
> [2] *Cxplain: Causal explanations for model interpretation under uncertainty. In NeurIPS, 2019.*
>
> [3] *Neil Jethani, Mukund Sudarshan, Ian Covert, Su-In Lee, and Rajesh Ranganath. Fastshap: Real-time shapley value estimation. arXiv:2107.07436, 2022.*
>
> [4] *Verifiable feature attributions: A bridge between post hoc explainability and inherent interpretability. In ICML, 2023.*

---

> ### Author Response · Authors · 2025-11-19
> **Feedback for Reviewer NUZJ (Part2/6)**
>
> > **W2&Q1: Experiment with held-out metrics**
> > - But without an experiment that uses held-out metrics, it’s hard to conclude DeepFaith is genuinely, broadly more faithful rather than just better at exploiting the metrics.
> > - Can you train DeepFaith while excluding one or two metrics from the curation filter, and then report performance specifically on those held-out metrics?
>
> **Response to W2 and Q1:**
>
> Thank you for this thoughtful and well-considered insight. Although **the design philosophy of DeepFaith is to leverage all available metrics to maximize the quality of the supervised explanation set—and our theoretical analysis (Property 1 and Theorem 1) establishes the inherent consistency among the ten faithfulness metrics**—we agree that this experiment is valuable for validating DeepFaith’s faithfulness in a broader sense.
>
> We conducted an additional experiment: we excluded **DEL** and **POS**—the only two metrics where *lower values indicate higher faithfulness*—from the supervision-set filtering process, trained DeepFaith on two high-dimensional explanation tasks (OCT+ResNet and ImageNet+DeiT), and compared the resulting explanations against several strong baseline methods specifically on the held-out **DEL** and **POS** metrics. The results are as follows:
>
> | Setting       | Metric | DeepFaith (held out) | DeepFaith | Occlusion | LIME | Kernel SHAP | Saliency | Guided Backprop |
> |---------------|--------|----------------------|-----------|-----------|------|-------------|----------|-----------------|
> | OCT+ResNet    | POS↓   | 0.265                | 0.242     | 0.372     | 0.364| 0.409       | 0.368    | 0.333           |
> |               | DEL↓   | 0.189                | 0.127     | 0.336     | 0.355| 0.239       | 0.360    | 0.400           |
> | ImageNet+DeiT | POS↓   | 0.301                | 0.295     | 0.383     | 0.394| 0.305       | 0.392    | 0.425           |
> |               | DEL↓   | 0.276                | 0.248     | 0.316     | 0.320| 0.343       | 0.443    | 0.307           |
>
> As shown in the table above, excluding POS and DEL from the supervision-set filtering leads to a slight decrease in DeepFaith’s performance on these two metrics compared with the original DeepFaith, which is expected due to the reduced quality of the supervision set. Nevertheless, **its faithfulness still surpasses all competing methods, demonstrating that DeepFaith is genuinely and broadly more faithful**, rather than merely exploiting the evaluation metrics.
>
> > **W3&Q3: Perturbation Scheme of Faithfulness Metrics**
> > - The paper should be explicit about whether all metrics were instantiated using a shared perturbation scheme or whether there are any assumptions.
> > - Did you enforce a consistent perturbation operator across all metrics in your experiments, or are you making any assumptions?
>
> **Response to W3 and Q3:**
>
> Thank you for the careful question regarding our implementation details. Different perturbation strategies do indeed have a significant impact on the computation of faithfulness metrics. We would like to clarify that in our experiments, all ten evaluation metrics were computed using a **uniform perturbation strategy**, namely baseline-value replacement, with the baseline values kept consistent within each task (see *src/modules/components/eval_metric_GPU.py* in our code repository). We have added an explanation of the perturbation scheme in Appendix D. Thank you again for the valuable suggestion!

---

> ### Author Response · Authors · 2025-11-19
> **Feedback for Reviewer NUZJ (Part3/6)**
>
> > **W4&Q5: Transfer Explanations across Target Models**
> > - The paper doesn’t test whether an explainer trained on one model can be reused for another model in the same domain. If there’s no transfer at all, then amortisation helps inference latency but does not reduce the number of explanation models.
> > - If I train DeepFaith for a DeiT classifier on ImageNet, how well does it explain an EfficientNet classifier on the same ImageNet labels without retraining?
>
> **Response to W4 and Q5:**
>
> Your idea is highly meaningful and valuable—performing cross–target-model explanation within the same task is indeed an important direction, and it aligns with a separate line of work we are currently pursuing. Our preliminary findings indicate that explanations for different target models on the same task do exhibit learnable regularities; however, such **model-agnostic explanations substantially reduce faithfulness for any individual target model**. This occurs because different models, even when trained on the same dataset, may rely on distinct intermediate representations, inductive biases, and feature hierarchies.
>
> DeepFaith, like prior Learning to Explain approaches, is designed to **capture the decision mechanism of a single target model and produce high-quality, model-specific explanations**. DeepFaith is built upon theoretical guarantees of optimal faithfulness and a carefully curated supervision signal, and its architecture achieves state-of-the-art faithfulness across 12 explanation tasks. For a given target model, DeepFaith already provides the most faithful explanation available, without requiring horizontal comparison or selection among baseline explanation methods.
>
> > **W5&Q6: Human Preference of Highly Faithful Explanations**
> > - All evaluation metrics are model-faithfulness metrics: does ablating important features change the model’s output in the expected direction?
> > - Did you run any preference checks with human annotators or domain experts? If you didn’t, can you comment on whether the high-faithfulness explanations ever look spurious?
>
> **Response to W5 and Q6:**
>
> Your observation is absolutely correct: **ensuring human interpretability is a central goal of explanation methods, and it is also an aspect we intended to highlight in our results**. Our theoretical analysis shows that the underlying logic of faithfulness metrics is to assess how important features influence the direction of the model’s output, a criterion that aligns closely with human explanatory preferences. To strengthen and clarify this point, we have revised Figures 5 and 6 in Appendix H by adding faithfulness-score comparisons and expanding the accompanying discussion:
>
> - *Figures 5 and 6 visualize the saliency maps produced by DeepFaith and baseline explanation methods on two image-modality datasets, together with their faithfulness scores INS and DEL. DeepFaith not only achieves superior faithfulness metrics but also **provides explanations that are more aligned with human interpretability**.*
>
> - *For example, on the OCT dataset shown in Figure 6, DeepFaith explanations **fully cover the relevant feature regions while highlighting finer-grained critical features**. For samples 4 and 5—each corresponding to different ocular diseases—the highlighted areas produced by DeepFaith concentrate precisely on the abnormal lesion regions. For sample 6, which is labeled as healthy, DeepFaith highlights are evenly distributed across the feature region. **The model decision rationale revealed by the explainer aligns closely with human understanding**.*
>
> Thank you again for bringing up this important point!

---

> ### Author Response · Authors · 2025-11-19
> **Feedback for Reviewer NUZJ (Part4/6)**
>
> > **W6&Q7: Sensitivity Study of Hyperparameters**
> > - The curated supervision pipeline depends on thresholds: cosine similarity thresholds for deduplication, p-quantile cutoffs for filtering which explanations count as faithful, and dynamic controller hyperparameters for toggling. The main results don’t present a sensitivity study.
> > - How sensitive is performance to the hyperparameters?
>
> **Response to W6 and Q7:**
>
> Thank you for raising this important concern. Although we used highly similar hyperparameters across explanation tasks on the same dataset and did not tune them for explainer performance, we agree that **adding sensitivity analyses would help further validate the robustness of DeepFaith.**
>
> **(a) Sensitivity study of cosine similarity thresholds and $p$-quantile**
>
> In constructing the supervisory explanation set, we use a similarity threshold $\delta$ and a quantile $p$ in the deduplicating and filtering stages. Adjusting these two parameters affects both the quantity and quality of the supervisory signals. Using the OCT+ResNet explanation task as an example, **we conducted an additional experiment evaluating five variants of DeepFaith**:
>
> - ${\rm DeepFaith}_{\delta,p}$: similarity threshold $\delta = 0.88$ and filtering quantile $p = 0.14$, which corresponds to the original configuration in our paper.
> - ${\rm DeepFaith}_{\delta \uparrow,p}$: similarity threshold increased to $\delta = 0.95$, representing a stricter deduplication policy.
> - ${\rm DeepFaith}_{\delta \downarrow,p}$: similarity threshold decreased to $\delta = 0.70$, representing a more permissive deduplication policy.
> - ${\rm DeepFaith}_{\delta,p \uparrow}$: filtering quantile increased to $p = 0.2$, representing a stricter filtering policy.
> - ${\rm DeepFaith}_{\delta,p \downarrow}$: filtering quantile decreased to $p = 0.05$, representing a more permissive filtering policy.
>
> We report the performance of all five variants across the 10 faithfulness metrics and compute the overall faithfulness deviation $|\Delta F|=\left | \frac{\sum_{i\in [10]} F_i-\sum_{i\in [10]}F_i^{\rm ori}}{\sum_{i\in [10]}F_i^{\rm ori}} \right |$, where $F_i$ and $F_i^{\rm ori}$ denote the score of the $i$-th faithfulness metric for the current variant and for ${\rm DeepFaith}_{\delta,p}$ respectively. The results are shown in the following table:
>
> | Method                                 | FC↑  | FE↑  | MC↑  | RP↑  | INS↑ | DEL↓ | NEG↑ | POS↓ | IROF↑ | INF↑ | $\|\Delta F\|$ |
> |----------------------------------------|-------|-------|-------|-------|-------|-------|-------|-------|-------|-------|--------------|
> | ${\rm DeepFaith}_{\delta,p}$           | 0.135 | 0.534 | 0.942 | 0.744 | 0.863 | 0.248 | 0.655 | 0.242 | 0.742 | 0.015 | 0        |
> | ${\rm DeepFaith}_{\delta \uparrow,p}$  | 0.140 | 0.497 | 0.919 | 0.706 | 0.783 | 0.266 | 0.628 | 0.266 | 0.717 | 0.016 | 3.513%        |
> | ${\rm DeepFaith}_{\delta \downarrow,p}$| 0.123 | 0.563 | 0.870 | 0.753 | 0.940 | 0.225 | 0.698 | 0.223 | 0.774 | 0.014 | 1.246%        |
> | ${\rm DeepFaith}_{\delta,p \uparrow}$  | 0.147 | 0.511 | 0.954 | 0.679 | 0.863 | 0.272 | 0.619 | 0.258 | 0.693 | 0.016 | 2.083%        |
> | ${\rm DeepFaith}_{\delta,p \downarrow}$| 0.122 | 0.575 | 0.891 | 0.817 | 0.797 | 0.235 | 0.705 | 0.218 | 0.799 | 0.014 | 1.013%        |
>
> From the results in the table, we observe that **DeepFaith is relatively robust to variations in both $\delta$ and $p$**. Across the different variants, the faithfulness scores remain highly similar, and the corresponding values of $|\Delta F|$ stay small (<5%). This suggests a clear practical takeaway: as long as $\delta$ and $p$ are chosen within a reasonable range, DeepFaith consistently maintains strong faithfulness performance.

---

> ### Author Response · Authors · 2025-11-19
> **Feedback for Reviewer NUZJ (Part5/6)**
>
> **(b) Sensitivity study of $e$ and $C$**
>
> During explainer training, Algorithm 1 introduces two additional hyperparameters: the monitoring window size $e$, which controls how sensitively the loss-variance trend is detected, and the scaling factor $C$, which influences the adjustment magnitude of the balancing weight $\alpha$. Similarly, **we conducted a sensitivity analysis on the OCT+ResNet explanation task**, constructing five variants as follows:
>
> - ${\rm DeepFaith} _ {e,C}$: monitoring window $e=5$ and scaling factor $C=10$, which corresponds to the original configuration in our paper.
> - ${\rm DeepFaith} _ {e\downarrow,C}$: window size reduced to $e=3$, making the algorithm more sensitive to variance changes in $\mathcal{L}_{\rm PC}$.
> - ${\rm DeepFaith} _ {e\uparrow,C}$: window size increased to $e=10$, making the algorithm less sensitive to variance changes in $\mathcal{L}_{\rm PC}$.
> - ${\rm DeepFaith} _ {e,C\downarrow}$: scaling factor reduced to $C=5$, decreasing the magnitude of $\alpha$-updates.
> - ${\rm DeepFaith} _ {e,C\uparrow}$: scaling factor increased to $C=20$, increasing the magnitude of $\alpha$-updates.
>
> We likewise report the performance of all five variants across the 10 faithfulness metrics and compute the overall faithfulness deviation $|\Delta F|$. The results are shown in the following table:
>
> | Method                              | FC↑   | FE↑   | MC↑   | RP↑   | INS↑  | DEL↓  | NEG↑  | POS↓  | IROF↑ | INF↑  | $\|\Delta F\|$ |
> |-------------------------------------|-------|-------|-------|-------|-------|-------|-------|-------|-------|-------|--------------|
> | ${\rm DeepFaith} _ {e,C}$             | 0.135 | 0.534 | 0.942 | 0.744 | 0.863 | 0.248 | 0.655 | 0.242 | 0.742 | 0.015 | 0.000        |
> | ${\rm DeepFaith} _ {e\downarrow,C}$   | 0.135 | 0.552 | 0.913 | 0.744 | 0.864 | 0.247 | 0.654 | 0.241 | 0.743 | 0.016 | 0.214%       |
> | ${\rm DeepFaith} _ {e\uparrow,C}$     | 0.118 | 0.588 | 0.924 | 0.709 | 0.884 | 0.276 | 0.672 | 0.270 | 0.664 | 0.036 | 0.410%        |
> | ${\rm DeepFaith} _ {e,C\downarrow}$   | 0.117 | 0.589 | 0.930 | 0.710 | 0.885 | 0.276 | 0.673 | 0.269 | 0.665 | 0.037 | 0.605%        |
> | ${\rm DeepFaith} _ {e,C\uparrow}$     | 0.136 | 0.557 | 0.905 | 0.743 | 0.863 | 0.248 | 0.654 | 0.243 | 0.740 | 0.016 | 0.292%        |
>
> As shown in the table above, the five variants of DeepFaith exhibit very similar faithfulness across the ten evaluation metrics, with overall faithfulness deviation $|\Delta F|$ being minimal (<1%). This indicates that **DeepFaith is insensitive to the hyperparameters $e$ and $C$**. This result is expected, as the design philosophy of Algorithm 1 is independent of the specific values assigned to these two parameters, and reasonable values can be chosen in practical applications.
>
> We have added the sensitivity analysis experiments to Appendix J. Thanks again for your helpful comment!

---

> ### Author Response · Authors · 2025-11-19
> **Feedback for Reviewer NUZJ (Part6/6)**
>
> > **Q4: About the Local Correlation Loss**
> > - For the Local Correlation loss: How many feature subsets are sampled per input/batch to estimate the correlation, and how sensitive is performance to that number?
> > - Do you observe high gradient variance or unstable optimisation when using alone? You show training dynamics for the whole method (including adapting over time); can you also share curves for the only variant to illustrate the claimed instability in isolation?
>
> **Response to Q4:**
>
> Thank you for the reviewer’s attention to $\mathcal{L}_{\rm LC}$ in the paper. We will respond to your questions in detail.
>
> **(a) Number of sampled feature subsets**
>
> For the sampling frequency $k$ in the design of $\mathcal{L}_{\rm LC}$, we use a uniform value of $k=100$ across all explanation tasks, which is explicitly documented in Appendix F. **The choice of $k$ impacts both the time cost during training and the sampling accuracy.** Higher values of $k$ improve sampling precision but significantly increase training time, while lower values of $k$ lead to larger sampling approximation errors. To empirically verify this, we conducted a comparison using the OCT+ResNet explanation task, evaluating three variants of DeepFaith:
>
> * ${\rm DeepFaith} _ {k}$: sampling frequency $k=100$, which corresponds to the original configuration in our paper.
> * ${\rm DeepFaith} _ {k \downarrow}$: sampling frequency $k=10$, resulting in faster training but lower sampling precision.
> * ${\rm DeepFaith} _ {k \uparrow}$: sampling frequency $k=200$, leading to slower training but higher sampling precision.
>
> The experimental results are shown in the following table, including 10 faithfulness evaluation metrics and the overall faithfulness deviation $\Delta F$:
>
> | Method                     | FC↑   | FE↑   | MC↑   | RP↑   | INS↑  | DEL↓  | NEG↑  | POS↓  | IROF↑ | INF↑  | $\Delta F$ |
> |----------------------------|-------|-------|-------|-------|-------|-------|-------|-------|-------|-------|--------------|
> | ${\rm DeepFaith} _ {k}$      | 0.135 | 0.534 | 0.942 | 0.744 | 0.863 | 0.248 | 0.655 | 0.242 | 0.742 | 0.015 | 0            |
> | ${\rm DeepFaith} _ {k \downarrow}$ | 0.143 | 0.522 | 0.913 | 0.727 | 0.847 | 0.245 | 0.626 | 0.236 | 0.721 | 0.005 | -2.636%            |
> | ${\rm DeepFaith} _ {k \uparrow}$   | 0.260 | 0.674 | 0.929 | 0.720 | 0.910 | 0.270 | 0.697 | 0.264 | 0.688 | 0.008 | +5.859%            |
>
> From the results in the table, we observe that the faithfulness of the two variants of DeepFaith aligns with expectations: decreasing $k$ leads to a decrease in faithfulness, while increasing $k$ improves faithfulness. However, **this adjustment has a minimal impact on DeepFaith's overall performance ($|\Delta F|< 6 \\% $), demonstrating DeepFaith's robustness to the hyperparameter $k$.** In practice, we recommend using the original design of $k=100$ as proposed in the paper to balance training cost and explainer performance.
>
> **(b) Training dynamics when optimizing only $\mathcal{L}_{\rm LC}$**
>
> We have revised Figure 3 of the paper to include the training dynamics obtained when optimizing $\mathcal{L} _ {\mathrm{LC}}$ alone (see $\mathcal{L} _ {\mathrm{LC}}'$ in Figure 3). In this setting, the loss fails to converge and lacks a stable optimization direction, **highlighting the essential role of Algorithm 1 in ensuring stable and convergent training behavior for DeepFaith.**

---

> > ### Comment · Reviewer_NUZJ · 2025-11-26
> >
> > Thanks for the detailed response. I appreciate it, and most of the issues I raised have been resolved. I have raised my scores.

---

> > > ### Author Response · Authors · 2025-11-26
> > > **Thanks for Reviewer NUZJ**
> > >
> > > Thank you, Reviewer NUZJ, for your positive evaluation and kind response. We are pleased to hear that our revision has fully addressed your concerns. Once again, thank you for your thoughtful and constructive feedback throughout the review process.

---

> ### Author Response · Authors · 2025-11-26
> **Looking forward to your reply!**
>
> Dear Reviewer NUZJ,
>
> We hope this message finds you well.
>
> We have carefully addressed your comments and made substantial revisions to our manuscript, as detailed in the responses and the revised PDF file. We have also summarized the key changes and enriched content to facilitate your review.
>
> Given that the review timeline is approaching its deadline, **we kindly request your feedback on the revised submission at your earliest convenience.** Your insights and comments are crucial for further improving the quality of our work, and we greatly value the opportunity for continued discussion.
>
> Thank you very much for your time and effort. Please do not hesitate to let us know if there are any additional clarifications or further details needed.
>
> Sending our best wishes for health, happiness, and all good things in life！
>
> The Authors

---

### Official Review · Reviewer_kSSW · 2025-10-29

**Soundness:** 3
**Presentation:** 3
**Contribution:** 2
**Rating:** 4
**Confidence:** 3

**Summary:**

This paper introduces **DeepFaith**, a method for generating explanations of neural network predictions within the “learning-to-explain” framework. The authors argue that previous approaches fall into two main categories: (1) *Self-supervised methods* that interact directly with the model to optimize a faithfulness objective, but rely on strong assumptions about the model; and (2) *Signal-based methods* that generate explanations using post-hoc techniques and then train a separate explainer model on these explanations, but do not explicitly optimize for faithfulness. DeepFaith is proposed as a hybrid approach that aims to leverage the strengths of both paradigms. It does so by generating a large set of explanations using diverse post-hoc methods and then guiding training using faithfulness-driven objectives. The authors justify this design by proving that different notions of faithfulness, including saliency-based and perturbation-based attribution measures, are aligned, supporting the idea of optimizing toward a unified faithfulness objective. In practice, DeepFaith is trained using two dynamically weighted objectives: the first encourages learning general explanation patterns, while the second shifts training toward optimizing faithfulness through a specific loss function. The method is evaluated across a wide range of vision and language benchmarks, where it consistently outperforms standard post-hoc explanation methods in terms of faithfulness metrics.

**Strengths:**

1. The idea of combining both the “learning from explanation signals” paradigm and the optimization via faithfulness objectives is interesting and, to the best of my knowledge, novel (though I may not be aware of all prior work in this area, which are many). However, the paper would benefit from stronger motivation and additional empirical evidence supporting the need for this approach (see Weaknesses).

2. The unification of various notions of saliency and attribution appears potentially useful in certain contexts.

3. The paper is generally well-written and easy to follow.

4. The authors present extensive experiments across a wide range of models, showing that their method achieves improved faithfulness compared to post-hoc approaches.

**Weaknesses:**

1. While the authors compare their method against post-hoc explanation techniques, they do not include comparisons to self-supervised approaches that directly optimize a faithfulness objective without relying on explanation signals. As a result, it remains unclear whether generating these explanation signals is necessary at all and the theoretical analysis does not justify this design choice.

2. The paper argues that self-supervised methods that directly optimize a faithfulness loss inherently rely on assumptions about the model, but this claim is insufficiently supported. The authors should clarify why such assumptions are required and provide concrete reasoning or evidence to motivate this statement.

3. Regarding the theoretical contributions: the presented proofs are relatively straightforward and unsurprising. The claim of unifying multiple attribution metrics and explanation methods is not entirely novel - prior work has already shown that many attribution methods optimize a shared class of objectives (e.g., Yu et al., NeurIPS 2019, among many others). The authors should better articulate what distinguishes their analysis from these earlier results and explain what new insights it provides.

**Questions:**

1. What justifies the claim that self-supervised approaches necessarily rely on assumptions about the model? Which assumptions are unavoidable, and why?

2. Why is it essential to generate explanation signals at all, rather than simply optimizing the faithfulness loss directly? What would change, conceptually or empirically, if we skipped the explanation generation step?

3. There is a substantial line of work on self-explaining models that produce explanations during inference. If DeepFaith were instead trained using a self-explaining architecture that generates explanations optimized via the faithfulness-based objective, how would this fundamentally differ from the proposed approach here, beyond the potential trade-off in accuracy?

---

> ### Author Response · Authors · 2025-11-19
> **Feedback for Reviewer kSSW (Part1/4)**
>
> we would like to thank you for the careful reading and constructive suggestions. We respond to each point below.
>
> > **W1: Comparisons to Self-Supervised Approaches**
> > - While the authors compare their method against post-hoc explanation techniques, they do not include comparisons to self-supervised approaches that directly optimize a faithfulness objective without relying on explanation signals.
> > - As a result, it remains unclear whether generating these explanation signals is necessary at all and the theoretical analysis does not justify this design choice.
>
> **Response to W1:**
>
> Thank you for the insightful suggestion about our experiments. We would like to clarify that DeepFaith is designed to jointly leverage high-quality supervised explanation signals and theoretically grounded faithfulness objectives, placing it **conceptually ahead of purely self-supervised approaches.** Although our ablation study (Table 4) already reports the performance of optimizing the self-supervised objective alone, we agree that a direct comparison with existing self-supervised *Learning to Explain* methods would be valuable.
>
> To this end, we **conducted additional experiments in which we compared the explanations produced by VerT[1], L2X[2], CXPlain[3], and DeepFaith** across four explanation tasks and ten faithfulness metrics, in order to **quantify the gain obtained from generating supervisory explanation signals**. We summarize the results below for your reference.
>
> | Setting         | Explainer   | FC↑  | FE↑  | MC↑  | RP↑  | INS↑ | DEL↓ | NEG↑ | POS↓ | IROF↑ | INF↑ | Mean Rank |
> |-----------------|-------------|------|------|------|------|------|------|------|------|-------|------|-----------|
> | NAP+MLP         | DeepFaith   | 0.788| 0.763| 0.952| 0.957| 0.844| 0.031| 0.770| 0.031| 0.844 | 0.238| 1.1       |
> |                 | L2X         | 0.129| 0.141| 0.899| 0.346| 0.832| 0.749| 0.819| 0.748| 0.302 | 0.142| 3.1       |
> |                 | CXPlain     | 0.017| 0.308| 0.501| 0.290| 0.153| 0.647| 0.722| 0.301| 0.445 | 0.143| 3.3       |
> |                 | VerT        | 0.772| 0.484| 0.560| 0.454| 0.564| 0.467| 0.518| 0.551| 0.603 | 0.160| 2.5       |
> | AgNews+LSTM     | DeepFaith   | 0.363| 0.597| 0.629| 0.648| 0.919| 0.197| 0.906| 0.256| 0.650 | 0.275| 1.5       |
> |                 | L2X         | 0.012| 0.303| 0.336| 0.367| 0.765| 0.621| 0.740| 0.618| 0.320 | 0.071| 3.4       |
> |                 | CXPlain     | 0.097| 0.258| 0.402| 0.328| 0.589| 0.189| 0.613| 0.189| 0.371 | 0.030| 2.9       |
> |                 | VerT        | 0.077| 0.519| 0.526| 0.710| 0.838| 0.362| 0.423| 0.204| 0.762 | 0.092| 2.2       |
> | OCT+ResNet      | DeepFaith   | 0.135| 0.534| 0.942| 0.744| 0.863| 0.248| 0.655| 0.242| 0.742 | 0.015| 1.3       |
> |                 | L2X         | 0.005| 0.121| 0.393| 0.501| 0.642| 0.505| 0.644| 0.433| 0.482 | 0.009| 3.2       |
> |                 | CXPlain     | 0.004| 0.239| 0.611| 0.356| 0.554| 0.525| 0.508| 0.513| 0.349 | 0.008| 3.8       |
> |                 | VerT        | 0.015| 0.397| 0.783| 0.540| 0.763| 0.466| 0.707| 0.226| 0.528 | 0.028| 1.7       |
> | ImageNet+DeiT   | DeepFaith   | 0.026| 0.447| 0.884| 0.486| 0.568| 0.127| 0.417| 0.295| 0.672 | 0.014| 1.5       |
> |                 | L2X         | 0.004| 0.101| 0.267| 0.198| 0.486| 0.526| 0.520| 0.481| 0.385 | 0.014| 3.2       |
> |                 | CXPlain     | 0.003| 0.023| 0.341| 0.237| 0.186| 0.172| 0.211| 0.192| 0.431 | 0.017| 3.0       |
> |                 | VerT        | 0.005| 0.131| 0.505| 0.414| 0.363| 0.323| 0.365| 0.367| 0.588 | 0.019| 2.3       |
>
> We have incorporated this experiment into the manuscript. Thank you again for your very helpful comment!
>
> [1] *Verifiable feature attributions: A bridge between post hoc explainability and inherent interpretability. In ICML, 2023.*
>
> [2] *Learning to explain: An information-theoretic perspective on model interpretation. In ICML, 2018.*
>
> [3] *Cxplain: Causal explanations for model interpretation under uncertainty. In NeurIPS, 2019.*

---

> ### Author Response · Authors · 2025-11-19
> **Feedback for Reviewer kSSW (Part2/4)**
>
> > **W2&Q1: Assumptions of Self-Supervised Methods**
> > - The paper argues that self-supervised methods that directly optimize a faithfulness loss inherently rely on assumptions about the model, but this claim is insufficiently supported. The authors should clarify why such assumptions are required and provide concrete reasoning or evidence to motivate this statement.
> > - What justifies the claim that self-supervised approaches necessarily rely on assumptions about the model? Which assumptions are unavoidable, and why?
>
> **Response to W2 and Q1:**
>
> Thank you for engaging with the motivation behind our work and for the thoughtful suggestion. We categorize *Learning to Explain* approaches into two families: self-supervised methods and prior-explanation-driven methods. **Unlike DeepFaith, which introduces supervised explanation signals as reliable priors, existing self-supervised *Learning to Explain* methods lack access to high-quality reference explanations.** As a result, they must construct their objectives based on assumptions about the target model or task. These assumptions often fail in realistic settings, preventing such methods from generalizing to scenarios that violate their underlying premises. For example:
>
> - VerT[1] **strictly assumes that each input $x$ can be decomposed into signal $s$ and distractor $d$**, and seeks a mask $m$ such that $x = s \odot m + d \odot (1-m)$, where $p(y|x) = p(y|s \odot m)$, while further requiring the model being explained is the **optimal predictor** $f^*$ in its subsequent analysis.
> - L2X[2] defines the explainer $\mathcal{E}$ by $\max_{\mathcal{E}} I(X_S; Y) \ \text{s.t. } S \sim \mathcal{E}(X), \ \ (X_S)_i = X_i \ \text{if } i\in S\ \text{else } 0$, which **fully assumes that the predictive effect of a feature subset $S$ can be represented solely by $X_S$**, thereby ruling out higher-order interactions among features.
> - CXPlain[3] defines the causal contribution of feature $i$ as $\omega_i(X)=\frac{\Delta\varepsilon_{X,i}}{\sum_j\Delta\varepsilon_{X,j}}$, where $\Delta\varepsilon_{X,i}=\varepsilon_{X\setminus{i}}-\varepsilon_X$ measures the perturbation effect of feature $i$. This formulation relies on the **idealized assumption that the observed data contain all causally relevant variables.**
>
> In contrast, DeepFaith benefits from supervised explanation signals as reliable priors, enabling it to perform the explanation task without imposing strict assumptions on the target model or task—and **even to produce explanations with substantially higher faithfulness.**
>
> To further support our claim, we provide empirical evidence (please refer to the table in W1). On ImageNet, where fine-grained structures and higher-order dependencies are ubiquitous, **the feature-separability assumptions required by VerT and L2X are difficult to satisfy**, resulting in substantially lower faithfulness. On the OCT medical-imaging task, where predictions are influenced by multiple latent clinical factors, **the causal-completeness assumption required by CXPlain does not hold**, and its faithfulness decreases markedly. In contrast, DeepFaith does not impose such restrictive assumptions on the model or task, enabling it to achieve **consistently strong faithfulness** across a broad range of models and datasets.
>
> [1] *Verifiable feature attributions: A bridge between post hoc explainability and inherent interpretability. In ICML, 2023.*
>
> [2] *Learning to explain: An information-theoretic perspective on model interpretation. In ICML, 2018.*
>
> [3] *Cxplain: Causal explanations for model interpretation under uncertainty. In NeurIPS, 2019.*

---

> ### Author Response · Authors · 2025-11-19
> **Feedback for Reviewer kSSW (Part3/4)**
>
> > **W3: Our Theoretical Contributions**
> > - Regarding the theoretical contributions: the presented proofs are relatively straightforward and unsurprising. The claim of unifying multiple attribution metrics and explanation methods is not entirely novel - prior work has already shown that many attribution methods optimize a shared class of objectives (e.g., Yu et al., NeurIPS 2019, among many others).
> > - The authors should better articulate what distinguishes their analysis from these earlier results and explain what new insights it provides.
>
> **Response to W3:**
>
> **(a) Is our theoretical contribution genuinely novel?**
>
> Thank you for raising this important point. We would like to clarify that prior unification work (e.g., Yu et al., NeurIPS 2019 [1]) focuses on unifying explanation methods, whereas our contribution is **the first to provide a unified formalization and theoretical analysis of faithfulness metrics**—many of which (e.g., DEL, INS, NEG, POS) have **not even been formally defined in earlier literature.** These theoretical contributions demonstrate the novelty of our work.
>
> **(b) Why does the proof of Theorem 1 appear relatively straightforward?**
>
> Building on a deeper examination of the foundations of faithfulness, we uncover and extract the shared functional components underlying these metrics—namely input perturbation, perturbation effect, retention effect, and correlation measure—and construct a concise and coherent theoretical framework for explanation faithfulness. **This structural unification is precisely what makes the proof of Theorem 1 appear relatively straightforward**.
>
> **(c) What new insights do we provide?**
>
> Moreover, our theoretical results **for the first time** show that a single explanation mapping $S_f^*$ attains optimality simultaneously across all ten faithfulness metrics. This **reveals an intrinsic consistency among a diverse set of metrics that were previously studied as independent criteria**, thereby offering new conceptual insights for the explainability research community.
>
> We hope this clarifies the distinction between our analysis and prior work, and we appreciate your thoughtful feedback.
>
> [1] *Chih-Kuan Yeh, Cheng-Yu Hsieh, Arun Sai, David Inouye, and Pradeep Ravikumar. On the (in)fidelity and sensitivity of explanations. In NeurIPS, 2019.*

---

> ### Author Response · Authors · 2025-11-19
> **Feedback for Reviewer kSSW (Part4/4)**
>
> > **Q2: The Purpose of Generating Supervised Explanation Signals**
> > - Why is it essential to generate explanation signals at all, rather than simply optimizing the faithfulness loss directly?
> > - What would change, conceptually or empirically, if we skipped the explanation generation step?
>
> **Response to Q2:**
>
> That is an excellent and insightful question. The motivation for generating these explanatory signals is twofold, addressing both the practical challenges of optimization and the conceptual gap between theory and application.
>
> **(a) Stabilizing optimization of the explainer**
>
> First, and most critically, **the supervised signals provide a stable direction for training initialization and convergence.** As noted in the paper, our faithfulness-based objective, the Local Correlation loss
>
> $$
> \mathcal{L} _{\mathrm{LC}}(\phi _\theta;\mathcal{D},f)=\frac{1}{2}-\frac{1}{2|\mathcal{D}|}\sum _{x\in \mathcal{D}}\tau \left [ ( {\textstyle \sum _ {i\in\mathcal{I} _ j}\phi _\theta(x) _i} ) _ {j=1}^k,\left( \Delta[f(x),f(x\setminus \mathcal{I} _ j )]\right) _ {j=1}^k  \right ],
> $$
>
> is highly non-convex because it nests the target model being explained. **Without high-quality supervisory guidance, the explainer lacks a reasonable starting point and struggles to find a stable optimization path.**
>
> This challenge is empirically demonstrated in our revised Figure 3 ($\mathcal{L} _ {\mathrm{LC}}'$). The training dynamics reveal that relying solely on $\mathcal{L} _ {\mathrm{LC}}$ leads to significant instability and poor convergence, whereas **our joint optimization strategy ensures a much smoother and more effective training process.**
>
> **(b) Approximating an intractable theoretical objective**
>
> Second, **these signals complement the theoretical objective in a way that makes it feasible to optimize.** While we theoretically derive an optimal faithfulness objective,
>
> $$
> S_f^*=\underset{S _ f}{\mathrm{argmax}}\ \tau\ \left [  ({ {\textstyle \sum} _ {j\in\mathcal{I} _ i}}S_f(x) _ j ) _ {i=1}^N,\left ( \Delta[f(x),f(x\setminus\mathcal{I} _ i)] \right ) _ {i=1}^N \right ],
> $$
>
> whose ideal form relies on the optimal solution for 10 metrics, directly optimizing this objective is intractable in practice. The supervised signals—produced by aggregating and rigorously filtering multiple prior methods—**offer a concrete and high-quality target.** This enables the explainer to first acquire strong explanatory behavior by approximating these empirically validated patterns, and subsequently refine its performance by moving closer to the theoretically optimal mapping.
>
> In summary, generating supervised signals is **not redundant but essential.** It bridges the gap between a theoretically sound yet practically difficult objective and a stable, high-performing amortized explainer, supporting robust training and ultimately superior faithfulness.
>
> > **Q3: DeepFaith and Self-Explaining Models**
> > - There is a substantial line of work on self-explaining models that produce explanations during inference. If DeepFaith were instead trained using a self-explaining architecture that generates explanations optimized via the faithfulness-based objective, how would this fundamentally differ from the proposed approach here, beyond the potential trade-off in accuracy?
>
> **Response to Q3:**
>
> Thank you for raising the point regarding combining DeepFaith with self-explaining models. We would like to clarify that **explainability methods belong to fundamentally different paradigms**. Self-explaining models generate explanations *during* inference, whereas DeepFaith produces explanations *after* the target model’s inference, making it a post-hoc explanation method. The sources of interpretability in these two paradigms differ fundamentally and cannot be seamlessly integrated.
>
> More specifically, the self-explaining paradigm requires **architectural constraints** on the target model to ensure that explanations are generated as part of the prediction process. Such constraints typically reduce the model’s predictive accuracy (as you noted). In contrast, DeepFaith **does not impose any architectural restrictions** on the target model. As a result, it does not degrade predictive performance and, unlike self-explaining methods, does not require architectural modifications for each target model—making it **applicable to a much broader range of models.**
>
> We hope this clarification addresses your question.

---

> ### Author Response · Authors · 2025-11-26
> **Looking forward to your reply!**
>
> Dear Reviewer kSSW,
>
> We hope this message finds you well.
>
> We have carefully addressed your comments and made substantial revisions to our manuscript, as detailed in the responses and the revised PDF file. We have also summarized the key changes and enriched content to facilitate your review.
>
> Given that the review timeline is approaching its deadline, **we kindly request your feedback on the revised submission at your earliest convenience.** Your insights and comments are crucial for further improving the quality of our work, and we greatly value the opportunity for continued discussion.
>
> Thank you very much for your time and effort. Please do not hesitate to let us know if there are any additional clarifications or further details needed.
>
> Sending our best wishes for health, happiness, and all good things in life！
>
> The Authors

---

> ### Comment · Reviewer_kSSW · 2025-11-28
>
> I thank the authors for the thorough response and the additional experiments! This is much appreciated! These address several of my concerns, including the comparison to a “no explanation signals’’ baseline.
>
> I’m still not sure I fully understand the claim that “any self-supervised method must rely on assumptions but DeepFaith doesn’t”. If the very “assumption’’ of these self-supervised methods is tied to how the faithfulness metrics they optimize for are defined, and DeepFaith works by optimizing an objective that finds the optimal explanation under different faithfulness metrics, then the argument feels slightly circular to me. In any case, this is not a reason for me to reject; I think that the method that is proposed in this work is interesting and shows strong results.
>
> I am open to raising my score, but following the authors rebuttal, I looked again on the Proposition+Theorem, and would appreciate some clarifications if possible. After attempting to read the proof of Theorem 1, I still find it quite abstract and difficult to follow. Adding a bit more detail to several steps would help, as well as adding the proof of Proposition 1. But even more valuable would be a small *running example*, for example, maybe using even a simple linear or Boolean model? In other words, to show a concrete case with a definition of what the optimal explanation is and see its equivalence across the different faithfulness metrics would make the results much clearer and highlight why the alignment of these optima is surprising.

---

> ### Author Response · Authors · 2025-11-30
> **Following up for Reviewer kSSW (Part 1/3)**
>
> Thank you for taking the time to carefully review our rebuttal and for your recognition of our work. We are glad that we were able to address several of your concerns. Regarding your latest comment, we hope the following response will satisfactorily resolve your remaining questions.
>
> > **Follow-up Question 1: Further Clarification on the Assumptions**
> > - I’m still not sure I fully understand the claim that "any self-supervised method must rely on assumptions but DeepFaith doesn’t". If the very "assumption" of these self-supervised methods is tied to how the faithfulness metrics they optimize for are defined, and DeepFaith works by optimizing an objective that finds the optimal explanation under different faithfulness metrics, then the argument feels slightly circular to me.
>
> **Response to FQ1:**
>
> Thank you for your strong interest in our work and for the continued engagement. Our claim that DeepFaith’s faithfulness objective does not rely on assumptions stems from a **fundamental difference** in how it operates compared with self-supervised methods. The key distinction lies in the **availability of high-quality supervisory explanation signals** as prior guidance.
>
> Self-supervised *Learning to Explain* methods do not have access to such reference explanations. Therefore, they must construct a proxy objective on their own, and **to make this proxy meaningful, they must introduce assumptions about the target model or task.** Without these structural assumptions, the model has no information from which to infer what the correct explanation target should be.
>
> In contrast, DeepFaith’s optimization objective is built from the intrinsic consistency shared across the faithfulness metrics and does not impose any assumptions on the model or task. However, **optimizing this objective purely in a self-supervised manner would also fail to yield useful explanations**, as shown in our ablation study (Table 4). The key to DeepFaith’s success lies in introducing supervisory explanation signals, which **replace those assumptions by serving as the necessary priors**, and in designing an effective joint optimization algorithm that balances the influence of the faithfulness loss and the supervised loss during training. This ensures that the explainer can first **acquire a basic level of explanatory capability** through imitation of existing explanations, and subsequently improve explanation quality through refinement guided by the faithfulness loss.
>
> In summary, the faithfulness loss, the supervised loss, and the joint optimization algorithm are **tightly interdependent components** of our approach. Their integration enables DeepFaith to function as a **generalizable explainer that produces highly faithful explanations across both low- and high-dimensional task settings.**
>
> We hope that this clarification helps address your questions!

---

> ### Author Response · Authors · 2025-11-30
> **Following up for Reviewer kSSW (Part 2/3)**
>
> > **Follow-up Question2: A Concrete Example to Understand the Theoretical Results**
> > - After attempting to read the proof of Theorem 1, I still find it quite abstract and difficult to follow. Adding a bit more detail to several steps would help, as well as adding the proof of Proposition 1. But even more valuable would be a small running example, for example, maybe using even a simple linear or Boolean model?
>
> **Response to FQ2:**
>
> Thank you for your careful reading of our theoretical analysis, especially the proof of Theorem 1. This result is central to DeepFaith, as it establishes a **unified optimally faithful explanation mapping** across all ten faithfulness metrics—one of the key theoretical insights of our work. In the revised manuscript, we have **provided additional details in the proof of Theorem 1** to facilitate readability, and we have also **included the proof of Proposition 1** (see Appendix A).
>
> Furthermore, to make the intuition more concrete, a simple example such as **a single-layer linear model** can indeed help illustrate the core ideas underlying the proof:
>
> ## **Setup**
>
> * Target model: $f(x)=0.1x _ 1+0.2x _ 2+0.3x _ 3+0.4x _ 4.$
>
> * Use a simple test input $x=(1,1,1,1)$. Then $f(x)=0.1+0.2+0.3+0.4=1.0.$
>
> * Removal perturbation: $f(x\setminus\mathcal I)$ means zero out coordinates in $\mathcal I$.
>
> * Define the optimal saliency explanation $s^* = (0.1,0.2,0.3,0.4).$
>
> * For any index set $\mathcal I$, define
>
>   $$
>   \Delta _ {f,x}(\mathcal I)=\Delta[f(x),f(x\setminus\mathcal I)]=f(x)-f(x\setminus\mathcal I)
>   $$
>
>   which, under zero-out perturbation and our $x$, equals the sum of the coefficients of removed features:
>
>   $$
>   \Delta _ {f,x}(\mathcal I)=\sum _ {j\in\mathcal I} w _ j,\quad\text{with }w=(w _ 1,w _ 2,w _ 3,w _ 4)=(0.1,0.2,0.3,0.4).
>   $$
>
> ## **Step1: Verify the key implication in the proof with concrete numbers**
>
> The proof needs the monotone implication
>
> $$
> \forall\ \mathcal I _ a,\mathcal I _ b,\quad
> \sum _ {j\in\mathcal I _ a} s^* _ j \ge \sum _ {j\in\mathcal I _ b} s^* _ j
> \quad\Longrightarrow\quad
> \Delta _ {f,x}(\mathcal I _ a)\ge \Delta _ {f,x}(\mathcal I _ b).
> $$
>
> With our choice $s^*=w$ this becomes equality of the two sums, so the implication holds trivially because both sides are identical:
>
> * Example A: take $\mathcal I _ a=\{4\}$, $\mathcal I _ b=\{1,2\}$.
>   $\sum _ {j\in\mathcal I _ a}s^* _ j = 0.4$.
>   $\sum _ {j\in\mathcal I _ b}s^* _ j = 0.1+0.2=0.3$.
>   Thus $0.4\ge0.3$. Now compute $\Delta$:
>   $\Delta _ {f,x}(\mathcal I _ a)=0.4,\ \Delta _ {f,x}(\mathcal I _ b)=0.3$. Hence $0.4\ge0.3$ — the implication holds.
>
> * Example B: take $\mathcal I _ a=\{3,4\}$, $\mathcal I _ b=\{1,2,3\}$.
>   $\sum _ {j\in\mathcal I _ a}s^* _ j=0.3+0.4=0.7$.
>   $\sum _ {j\in\mathcal I _ b}s^* _ j=0.1+0.2+0.3=0.6$.
>   $\Delta(\mathcal I _ a)=0.7,\ \Delta(\mathcal I _ b)=0.6$. Again holds.
>
> Because for all $\mathcal I$ we have $\sum _ {j\in\mathcal I} s^* _ j=\sum _ {j\in\mathcal I} w _ j=\Delta _ {f,x}(\mathcal I)$, the monotonicity is exact.
>
> This concretely rules out the contradiction scenario in the abstract proof: there cannot exist an alternative saliency $s$ that reverses the ordering of sums while increasing the correlation with the $\Delta$-vector, because **the correlation is already maximized when saliency sums exactly match the $\Delta$ values.**
>
> ## **Step2: Permutation ordering**
>
> The most faithful permutation explanation   $\pi ^ * = \mathfrak{P}(s ^* ) $ sorts indices by decreasing $s^* $ (equivalently by the coefficients). Thus
>
> $$
> \pi^* = (4,3,2,1).
> $$
>
> For any prefix length $i$,
>
> $$
> \sum _ {j=1}^i s^* _ {\pi^* (j)} \ge \sum _ {j=1}^i s^* _ {\pi(j)}
> $$
>
> for any other permutation $\pi$. Because $\Delta _ {f,x}(\bigcup _ {j=1}^i \pi^*(j))$ equals the same sum of weights, we get
>
> $$
> \Delta _ {f,x}\Big(\bigcup _ {j=1}^i\pi^*(j)\Big)\ge \Delta _ {f,x}\Big(\bigcup _ {j=1}^i\pi(j)\Big),
> $$
>
> verifying the inequality used to conclude $\mathrm{RP}(\Pi _ f^*)\ge\mathrm{RP}(\Pi _ f)$.
>
> Concrete numeric check for $i=2$:
>
> * prefix of $\pi^*$ is $\{4,3\}$ → sum = 0.7.
> * any other permutation prefix of length 2 has sum ≤ 0.7 (for instance $\{1,2\}$→ sum =0.3).
>   So $\Delta$ preserves the ordering.

---

> ### Author Response · Authors · 2025-11-30
> **Following up for Reviewer kSSW (Part 3/3)**
>
> ## **Step3: Consequences for DEL / POS / NEG / IROF / INS**
>
> Because $\Delta _ {f,x}(\mathcal I)$ here is a nonnegative sum of coefficients and is **strictly increasing with inclusion of higher-weight features**, the sign and monotone relations in the proof follow numerically:
>
> * $\Delta^{-} _ {f,x}(\mathcal I)$ (the metric that decreases with $\Delta$) will be smaller for index sets with larger $\Delta$. Hence $\mathrm{DEL}(\pi^* ;x,f)\le \mathrm{DEL}(\pi;x,f)$ and $\mathrm{POS}(\pi^*;x,f)\le \mathrm{POS}(\pi;x,f)$ hold numerically.
>
> * For NEG and IROF, which increase with $\Delta$ or cumulative retained effect, we get $\mathrm{NEG}(\pi^* ;x,f)\ge\mathrm{NEG}(\pi;x,f)$ and $\mathrm{IROF}(\Pi _ f^*)\ge\mathrm{IROF}(\Pi _ f)$.
>
> * For INS, with baseline $x^\circ=0$, inserting features in the order $\pi^* $ yields larger increases in model output earlier (because the largest coefficients are inserted first), thus $\mathrm{INS}(\pi^* ;x,f)\ge \mathrm{INS}(\pi;x,f)$.
>
> ## **Explanation of Proposition 1**
>
> For Proposition 1, the optimality of $S _ f^* $ follows **directly from its definition in Eq.1**. Because $S _ f^* $ maximizes the correlation between the local saliency sums and the perturbation effects for all index sets, it **yields the highest possible alignment** between explanation scores and true model-driven importance. The four metrics $\mathrm{FC}$, $\mathrm{FE}$, $\mathrm{INF}$, and $\mathrm{MC}$ are all monotone transformations of this alignment; therefore, **any deviation from $S _ f^*$ necessarily produces a weaker correspondence and strictly lower scores** under these metrics. Hence, $S _ f^*$ attains their optimal faithfulness values by construction.
>
> We hope that this illustrative example helps improve the clarity and accessibility of the theoretical results. Thank you again for your thoughtful and careful feedback!

---

### Author Response · Authors · 2025-11-30
**Summary from the Authors (Part 2/2)**

### **Summary of the Rebuttal Result**

We thank all reviewers for their careful reading and continued engagement. We carefully **responded to every weakness and question** raised by all four reviewers, and through multiple rounds of discussion and revision, all four reviewers **acknowledged our analyses, additional experiments, and illustrative examples, and increased their overall evaluation in their follow-up comments.**

> **kSSW:** I thank the authors for the thorough response and the additional experiments! **This is much appreciated! These address several of my concerns,** including the comparison to a "no explanation signals" baseline. I am **open to raising my score.** (The scores were locked when the reviewer replied.)

> **NUZJ:** Thanks for the detailed response. I appreciate it, and **most of the issues I raised have been resolved. I have raised my scores.** (6 -> 8)

> **JuTS:** Thanks for the detailed response. I can see the authors put effort into addressing my comments, and **most of the issues I raised have been resolved.** Thanks for the clarification, **I've raised my score.** (4 -> 6)

> **FfQQ:** Thanks to the authors' clarity and extensive experiments. **The response solves the questions I have mentioned.** This paper unifies two major types of explanations, which represent **a meaningful contribution** to the XAI field.

To facilitate the AC’s decision-making, we provide below an objective summary of the rebuttal process, with an emphasis on the **key shared concerns and how they were addressed**.

**(1)** **Reviewers kSSW, NUZJ, and JuTS** suggested that direct comparison with prior *Learning to Explain* methods would further strengthen the validation of DeepFaith. We agreed with this recommendation and **added the corresponding comparison experiments (Table 3).** In their subsequent responses, **all three reviewers explicitly acknowledged our contributions to the *Learning to Explain* paradigm.**

> **kSSW:** They do not include **comparisons to self-supervised approaches** that directly optimize a faithfulness objective without relying on explanation signals.
-> These address several of my concerns, including the comparison to a “no explanation signals’’ baseline.

> **NUZJ:** Can you include at least one **amortised explainer baseline** per modality?
-> Thanks for the detailed response. I appreciate it, and most of the issues I raised have been resolved.

> **JuTS:** Specifically, the statement "self-supervised objectives rely on assumptions about the target model" could be better **supported with a concrete example.**
-> Thanks for the detailed response. I can see the authors put effort into addressing my comments, and most of the issues I raised have been resolved.

**(2)** **Reviewers NUZJ and FfQQ** raised questions regarding the sensitivity of DeepFaith to its hyperparameters. In response, we **provided additional conceptual clarification and conducted a dedicated sensitivity analysis (Appendix J)**, demonstrating that DeepFaith is **sufficiently robust** to hyperparameter variations and providing practical ranges for each hyperparameter. The reviewers acknowledged that our method is not overly sensitive to these choices.

> **NUZJ:** How sensitive is performance to the **hyperparameters**?
-> Thanks for the detailed response. I appreciate it, and most of the issues I raised have been resolved.

> **FfQQ:** This framework introduces a new set of **sensitive hyperparameters** that require manual tuning for each task.
-> The response solves the questions I have mentioned.

**(3)** **Reviewers NUZJ, and JuTS** asked for clarification on how DeepFaith leverages the two loss functions during training. We addressed this by **adding the training dynamics under optimization of the faithfulness loss alone (Figure 3) and by referring to the ablation study (Table 4)**, which together clarify the distinct and complementary roles of the two losses as well as their joint optimization mechanism. All three reviewers confirmed in their subsequent comments that this issue had been resolved.

> **NUZJ:** Do you observe high gradient variance or unstable optimisation when **using $\mathcal{L} _ {\rm LC}$ alone**?
-> Thanks for the detailed response. I appreciate it, and most of the issues I raised have been resolved.

> **JuTS:** In Algorithm 1, they presents a variance-based switching rule for adjusting α, but the paper provides no theoretical guarantee of stability or convergence. And Fig 3 still shows **noticeable oscillations during training**, suggesting limited robustness of the proposed mechanism.
->  I can see the authors put effort into addressing my comments, and most of the issues I raised have been resolved.

---

Once again, we greatly appreciate your time, effort, and careful consideration of our work. We sincerely hope that this summary, together with our detailed responses and revisions, will be helpful for your assessment.

Respectfully,

The Authors

---

### Author Response · Authors · 2025-12-01
**Summary from the Authors (Part 1/2)**

Dear Area Chair,

Thank you very much for your time and effort in handling this process under such unexpected and challenging circumstances. We sincerely appreciate the additional reviewing responsibilities that Area Chairs have undertaken, and we fully understand the difficulty of making fair decisions in a situation like this. In support of your work, we would like to **provide a succinct summary of our paper, its contributions, and the outcome of the rebuttal and discussion process,** in the hope of alleviating your workload as much as possible.

### **Overview of Our Work**

We focus on the *Learning to Explain* paradigm in explainable artificial intelligence and propose **DeepFaith**, a learning-based method that significantly enhances explanation **faithfulness** across multiple modalities.

**(1)** We are the first to **unify ten widely used faithfulness evaluation metrics** under a single theoretical framework, and to **prove the existence of a shared optimally faithful explanation mapping**. This contribution was explicitly recognized by **Reviewers NUZJ and FfQQ** as providing a single mathematical lens over correlation-based and perturbation-based metrics, and by **Reviewer kSSW** as a potentially useful unification of different notions of saliency and attribution.

> **NUZJ:** The paper proposes a single **mathematical lens** that captures 10 commonly used faithfulness metrics and argues that there exists an optimal explanation mapping that is jointly optimal across all of them.

> **FfQQ:** This paper provides **strong theoretical grounding** for the self-supervised objective. The paper makes a laudable effort to build its self-supervised objective on a firm theoretical foundation, rather than relying on heuristics or task-specific assumptions.

> **kSSW:** The authors justify this design by proving that different notions of faithfulness, including saliency-based and perturbation-based attribution measures, **are aligned, supporting the idea** of optimizing toward a unified faithfulness objective.

**(2)** We are the first to **introduce a high-quality set of supervised explanation signals into explainer training, along with a principled pipeline for constructing and filtering prior explanations**. **Reviewer NUZJ** highlighted this per-sample curation process as a clever and effective way to ensure that the distilled explainer can even surpass its teachers, while **Reviewer FfQQ** emphasized that this design bridges causal relevance with optimization in a principled manner.

> **NUZJ:** Generate many explanations, deduplicate them, and select only those that empirically score best across multiple metrics for that specific input. This is **a clever way** to ensure the distilled model surpasses the teacher's abilities.

> **FfQQ:** This paper **provides a novel faithfulness-aware learning framework**. It defines synthetic supervision using controlled feature perturbations (Sec. 3.1; Fig. 2), bridging causal interpretability with optimization.

**(3)** We propose a **dynamic joint-optimization algorithm** that balances the supervised objective and the faithfulness objective during training. With the same explainer architecture, DeepFaith achieves the **best faithfulness performance across image, text, and tabular data, and six target models**. This was positively acknowledged by **Reviewers kSSW, JuTS, and FfQQ**, who noted both the effectiveness of the method and the breadth of our experimental validation.

> **kSSW:** The authors present **extensive experiments** across a wide range of models, showing that their method achieves improved faithfulness compared to post-hoc approaches.

> **JuTS:** Experiments across image, text, and tabular datasets and the paper is clear-written; the experiments show **improved faithfulness and efficiency** across different datasets.

> **FfQQ:** This paper provides a **comprehensive empirical evaluation**. The authors consider image, text, and tabular datasets and various architectures, such as ResNet, Deit, Transformer, and LSTM, **demonstrating improvement across explainers (GradCAM, IG, LRP), confirming generalizability.**

Overall, DeepFaith rethinks the *Learning to Explain* paradigm from **theoretical, data-driven, and algorithmic perspectives**, and provides a principled and empirically validated answer to a fundamental question in XAI: *How can we learn highly faithful explanations?*

---

### Meta-Review · Area_Chair_Jaz6 · 2026-01-13

**Summary:**

The paper’s core contribution is a supervised learn-to-explain framework that aims to produce explanations that are simultaneously faithful under multiple faithfulness notions, rather than being tuned to a single metric or behaving like a hidden test-time optimizer. The reviews converged on a single question that ultimately drove the recommendation: is the claimed advantage real and general, or is it an artifact of missing baselines, metric leakage, or a brittle target-construction pipeline. The main concerns were that the original experimental section did not benchmark against the most relevant learn-to-explain alternatives, that the “multi-metric faithfulness” story could be confounded if the same metrics used for evaluation also implicitly shaped the training signal through filtering or selection, and that the controller-based generation of training targets was under-specified and might be unstable across settings. Secondary concerns were about the practical cost of the overall pipeline and whether the presentation made the conceptual distinctions crisp enough for readers to audit what is being optimized versus what is being measured.

**Reviewer Concerns:**

The rebuttal substantially addressed the core experimental validity concerns by adding the missing learn-to-explain baselines (L2X, CXPlain, FastSHAP, VerT) and showing that the proposed method remains competitive or better across multiple datasets and metrics, which directly resolves the strongest "incomplete baseline set" critique. The rebuttal also tackled the most serious leakage concern by running a held-out-metrics protocol in which certain faithfulness metrics were excluded from the filtering stage and used only for evaluation, and the results still favored the method, which reduces the likelihood that improvements are an artifact of optimizing on the evaluation metrics. The sensitivity analyses on filtering thresholds and controller hyperparameters, together with the added training dynamics for the faithfulness-loss-only variant, also respond to worries about brittleness and optimization stability and help justify why supervised signals are needed in practice.

What remains outstanding is mostly about completeness and framing rather than correctness. The compute and data-generation cost is still not quantified in a way that lets readers assess amortization and practicality, and this matters because the method’s pipeline includes a nontrivial target-construction step. Some reviewers’ conceptual objections, especially about whether the approach truly avoids assumptions compared to self-supervised alternatives, are only partially resolved because they are partly rhetorical and partly definitional. Finally, while the theory and explanations were clarified, there is still a risk that the presentation remains heavier than necessary, since at least one reviewer asked for more concrete, minimal examples to make the claims easier to audit.

**Reviewer Scores:**

For NUZJ, I expect the score would remain at the updated level of 8 after full participation, since the rebuttal directly answered the baseline gap and the leakage concern in the way that reviewer requested, and the remaining compute-reporting issue is unlikely to change their overall stance. For JuTS, I expect the score to increase to 6 (since the reviewer said so), because the newly added baselines and robustness checks strengthen the empirical story that underpins their "borderline positive" position, while the missing compute accounting likely prevents a jump to a strong accept. For FfQQ, I expect the score would stay at 6 or slightly increase, depending on how much weight they place on practicality; the rebuttal improved the credibility of the method’s generality, but the remaining cost and framing gaps could keep them at weak accept. For kSSW, I expect an increase from 4 to 5, and possibly to 6 if the reviewer is satisfied that the held-out-metrics protocol meaningfully reduces circularity; they already indicated the concerns were not fatal, and the new experiments remove the most reject-oriented argument, but the lack of cost quantification and residual framing issues likely keep them from becoming strongly positive.

---

### Decision · Program_Chairs · 2026-01-26

Accept (Poster)